# Improved Analysis of the Accelerated Noisy Power Method with Applications to Decentralized PCA

Pierre Aguié [1]  Mathieu Even [2]  Laurent Massoulié [1]

## Abstract

We analyze the Accelerated Noisy Power Method, an algorithm for Principal Component Analysis in the setting where only inexact matrix-vector products are available, which can arise for instance in decentralized PCA. While previous works have established that acceleration can improve convergence rates compared to the standard Noisy Power Method, these guarantees require overly restrictive upper bounds on the magnitude of the perturbations, limiting their practical applicability. We provide an improved analysis of this algorithm, which preserves the accelerated convergence rate under much milder conditions on the perturbations. We show that our new analysis is worst-case optimal, in the sense that the convergence rate cannot be improved, and that the noise conditions we derive cannot be relaxed without sacrificing convergence guarantees. We demonstrate the practical relevance of our results by deriving an accelerated algorithm for decentralized PCA, which has similar communication costs to non-accelerated methods. To our knowledge, this is the first decentralized algorithm for PCA with provably accelerated convergence.

## 1. Introduction

Principal Component Analysis (PCA) is a ubiquitous task in machine learning and statistics. Given a symmetric positive semidefinite matrix $\mathbf{A} \succeq \mathbf{0}$ and a target rank $k$, the goal is to estimate the subspace spanned by the $k$ leading eigenvectors of $\mathbf{A}$. The Power Method (Golub & Van Loan, 2013) is a popular algorithm for PCA that iteratively refines an estimate of this subspace, using only matrix-vector products with $\mathbf{A}$. Such algorithms are called *matrix-free*, in that they

do not require explicit access to $\mathbf{A}$ but only to the product $\boldsymbol{x} \mapsto \mathbf{A}\boldsymbol{x}$. This operation can be done at a low computational cost even when $\mathbf{A}$ is very large if it has a favorable structure, such as sparsity or a low-rank factorization. More recently, there has been a growing interest in studying algorithms for PCA in cases where only an approximate matrix-vector product $\boldsymbol{x} \mapsto \mathbf{A}\boldsymbol{x} + \boldsymbol{\xi}$ is available, where $\boldsymbol{\xi}$ is a perturbation of *bounded magnitude*. Such inexact matrix-vector products naturally arise in various practical scenarios. For instance, in private PCA (Chaudhuri et al., 2012), random noise is added to $\mathbf{A}\boldsymbol{x}$ to ensure privacy. In stochastic or streaming PCA (Xu et al., 2018), $\mathbf{A}$ is the covariance matrix of a distribution, and only noisy estimates of $\mathbf{A}\boldsymbol{x}$ can be computed using samples from this distribution. In decentralized PCA (Wai et al., 2017), a network of agents each holding a local matrix $\mathbf{A}_i$ seeks to estimate the top-$k$ eigenspace of the average matrix $\mathbf{A} = n^{-1} \sum_{i=1}^{n} \mathbf{A}_i$, by only exchanging information with their neighbors, and with no central aggregating server. The agents can then only compute approximate matrix-vector products, where in that case, the approximation error $\boldsymbol{\xi}$ stems from the limited communication between the agents. We stress that in all of these examples, there is a trade-off between the magnitude of the noise $\boldsymbol{\xi}$ and the strength of the external constraints: in private PCA, stronger privacy guarantees require larger noise; in stochastic PCA, the noise magnitude increases with smaller sample sizes; in decentralized PCA, limited communication budgets lead to larger approximation errors.

Hardt & Price (2014) show that the Power Method with approximate matrix-vector products keeps the same convergence rate as in the noiseless case, provided that the magnitude of the noise remains $\varepsilon$-small, where $\varepsilon$ is the target precision of the estimate. The convergence rate of Hardt & Price (2014) is prohibitively slow for ill-conditioned problems, in which the $k$ and $(k+1)$-th eigenvalues of $\mathbf{A}$ are very close. Xu (2023) proposes an accelerated version of the Noisy Power Method that achieves a faster rate, which matches the optimal worst-case rate achievable by Krylov subspace methods in the noiseless case (Saad, 2011). This represents a significant speedup for poorly conditioned matrices, which often appear in practice (Musco & Musco, 2015). However, their analysis requires the noise magnitude to be $\varepsilon^{\mu}$-small, where $\mu$ is a very large power for

[1]Inria, DI ENS, PSL Research University, France [2]Inria, Inserm, Université de Montpellier, France. Correspondence to: Pierre Aguié <pierre.aguie@inria.fr>.

*Proceedings of the $43^{rd}$ International Conference on Machine Learning*, Seoul, South Korea. PMLR 306, 2026. Copyright 2026 by the author(s).

*Table 1.* Comparison of convergence rates and noise conditions for (Accelerated) Noisy Power Method. Notations defined in Section 1.3. $T$ is the number of iterations required to reach $\sin \theta_k(\mathbf{U}_k, \mathbf{X}_T) \leqslant \varepsilon$, $\mathbf{\Xi}_t$ is the noise at iteration $t$. Xu (2023)'s conditions were adapted to make comparisons more direct (see Appendix B for more details). $\mu_k, \mu_{k+1}$ are constants verifying $\mu_i = \Omega(\log(\lambda_1/\lambda_i)\sqrt{\lambda_k/(\lambda_k - \lambda_{k+1})})$. †Results for Accelerated Noisy Power Method, with optimal parameter choice $\beta = \lambda_{k+1}^2/4$.

| | $T$ | Cond. on $\|\mathbf{U}_{-k}^\top \mathbf{\Xi}_t\|_2$ | Cond. on $\|\mathbf{U}_k^\top \mathbf{\Xi}_t\|_2$ |
|---|---|---|---|
| Hardt & Price (2014) | $\mathcal{O}\left(\frac{\lambda_k}{\lambda_k - \lambda_{k+1}} \log\left(\frac{1}{\varepsilon}\right)\right)$ | $\mathcal{O}\left((\lambda_k - \lambda_{k+1})\varepsilon\right)$ | $\mathcal{O}\left((\lambda_k - \lambda_{k+1}) \cos\theta_k(\mathbf{U}_k, \mathbf{X}_t)\right)$ |
| Xu (2023)† | $\mathcal{O}\left(\sqrt{\frac{\lambda_k}{\lambda_k - \lambda_{k+1}}} \log\left(\frac{1}{\varepsilon}\right)\right)$ | $\tilde{\mathcal{O}}\left((\lambda_k - \lambda_{k+1})\varepsilon^{\mu_{k+1}}\right)$ | $\tilde{\mathcal{O}}\left((\lambda_k - \lambda_{k+1}) \cos\theta_k(\mathbf{U}_k, \mathbf{X}_t)\,\varepsilon^{\mu_k}\right)$ |
| Theorem 2.2 (**this paper**)† | $\mathcal{O}\left(\sqrt{\frac{\lambda_k}{\lambda_k - \lambda_{k+1}}} \log\left(\frac{1}{\varepsilon}\right)\right)$ | $\mathcal{O}\left((\lambda_k - \lambda_{k+1})\varepsilon\right)$ | $\mathcal{O}\left((\lambda_k - \lambda_{k+1}) \cos\theta_k(\mathbf{U}_k, \mathbf{X}_t)\right)$ |

ill-conditioned problems. Their conditions are thus significantly more restrictive than those of Hardt & Price (2014), as shown in Table 1, and render their results impractical for applications. As explained above, in practice the noise magnitude is determined by system constraints, and gets larger as the constraints get stronger. The relationship between the target precision and the magnitude of the noise leads to a trade-off between the utility of the estimate given by the algorithm and the constraints of the problem. It is as such crucial to have noise conditions that are as mild as possible, in order to allow for accurate estimates even under strong system constraints. As an example, while non-accelerated algorithms for decentralized PCA exist in the literature (Wai et al., 2017; Ye & Zhang, 2021), we are not aware of any algorithm for decentralized PCA that converges at accelerated rates. We believe that this hole in the literature is due to the overly restrictive noise conditions required by existing analyses of accelerated methods, which prevent their application to decentralized PCA under reasonable communication budgets. Note that in these scenarios, the number of iterations required by these algorithms also leads to increased costs in terms of privacy loss, communication rounds or number of samples used. Acceleration is thus essential to improve the utility-cost trade-off of those algorithms.

## 1.1. Related Work

**Accelerated rates for PCA.** The first matrix-free method to provably achieve accelerated convergence was proposed by Lanczos (1950) for large-scale sparse matrices. This method belongs to the class of Krylov subspace methods, described in (Saad, 2011). Musco & Musco (2015) provide accelerated gap-independent rates for Krylov methods. Taking inspiration from Polyak (1964)'s Heavy Ball method for convex optimization, Xu et al. (2018) propose a variant of the Power Method with a momentum term that achieves acceleration for appropriate parameter choices. Similar momentum-based methods were used previously to accelerate gossip algorithms (Liu & Morse, 2011).

**Noisy power method.** Hardt & Price (2014) give the first analysis of the Noisy Power Method (NPM). Letting $\Delta_k$

be the relative gap between the $k$ and $(k+1)$-th eigenvalues, they show that NPM converges in $\tilde{\mathcal{O}}(\Delta_k^{-1})$[1] iterations, assuming that the noise $\boldsymbol{\xi}$ scales like $\mathcal{O}(\Delta_k \varepsilon)$, where $\varepsilon$ is the target precision of the estimate. This analysis was later extended by Balcan et al. (2016) to account for wider gaps when the iterate $\mathbf{X}_t$ has more columns than the target rank $k$. The Accelerated Noisy Power Method (ANPM), which adds a momentum term to NPM, was first introduced by Mai & Johansson (2019). Xu & Li (2022) then proposed an analysis of ANPM which shows accelerated convergence in $\tilde{\mathcal{O}}(\Delta_k^{-1/2})$, but requires unnatural conditions on the noise that are hard to verify in practice. These unnatural conditions were later removed in Xu (2023)'s analysis, which however still requires restrictive bounds on the noise's magnitude, of the form $\mathcal{O}(\Delta_k \varepsilon^\mu)$, where $\mu = \tilde{\Omega}(\Delta_k^{-1/2})$. Table 1 provides a comparison of results for noisy power methods. All of these works consider adversarial noise with bounded spectral norm, as opposed to the centered stochastic noise considered for instance in Shamir (2016); Xu et al. (2018), which is orthogonal to our analysis. ANPM has been applied to fair PCA by Zhou et al. (2026).

**Decentralized PCA.** Many decentralized versions of the Power Method have been proposed (Kempe & McSherry, 2008; Raja & Bajwa, 2016; Wai et al., 2017), which leverage gossip algorithms (Boyd et al., 2006) to approximate the matrix-vector product $\mathbf{A}\boldsymbol{x}$ in a decentralized manner. Such approaches require a number of communication rounds that increase with the target accuracy. Ye & Zhang (2021) propose an improved version of the decentralized power method with a communication cost that does not increase with the target precision, inspired by gradient tracking methods in decentralized optimization (Koloskova et al., 2021). All of these algorithms converge at a non-accelerated rate. Other approaches for decentralized PCA include decentralized versions of Oja's algorithm (Gang & Bajwa, 2022), which converge at a non-accelerated linear rate, and approaches based on decentralized Riemannian optimization (Chen et al., 2021) which only guarantee convergence to a stationary point, with no guarantee of retrieving the top-$k$

---

[1] $\tilde{\mathcal{O}}, \tilde{\Omega}$ and $\tilde{\Theta}$ hide logarithmic factors.

eigenspace. We refer the reader to the survey of Wu et al. (2018) for a more complete overview of the literature on decentralized PCA.

### 1.2. Our Contributions

We propose a novel analysis of the Accelerated Noisy Power Method, which preserves the guarantee of an accelerated convergence rate under milder noise conditions than those given in previous works. Our contributions are as follows:

**(i)** We provide new guarantees for the Accelerated Noisy Power Method. Just like in Xu (2023)'s work, our analysis shows that the algorithm converges at a rate linear in $1/\sqrt{\Delta_k}$. However, our noise conditions are the same as those of Hardt & Price (2014) for the non-accelerated Noisy Power Method, and are significantly milder than those of Xu (2023) in cases where the eigengap $\Delta_k$ is small (see Table 1 for a comparison).

**(ii)** We show that our analysis is worst-case optimal up to constants: there are instances of the algorithm which converge at a rate slower than $1/\sqrt{\Delta_k}$, and there are instances verifying relaxed versions of our noise conditions that do not converge to the target precision.

**(iii)** We use our analysis to derive an accelerated algorithm for decentralized PCA, which has similar communication costs to non-accelerated methods (see Table 2 for a comparison with other decentralized algorithms for PCA). To our knowledge, this is the first decentralized algorithm for PCA with accelerated convergence.

We stress that in our work and those of Hardt & Price (2014) and Xu (2023), the perturbations $\xi$ are not assumed to be stochastic, but rather to be *adversarial* and of *bounded norm*.

### 1.3. Notations

For a positive semidefinite matrix (PSD) $\mathbf{A} \succeq \mathbf{0}$, we denote by $\lambda_1 \geqslant \lambda_2 \geqslant \cdots \geqslant \lambda_d \geqslant 0$ its eigenvalues in non-increasing order, and we let $u_1, \ldots, u_d$ be corresponding orthonormal eigenvectors. For all $k \in \{1, \ldots, d-1\}$, let $\mathbf{U}_k := [u_1, \ldots, u_k]$, $\mathbf{U}_{-k} := [u_{k+1}, \ldots, u_d]$, $\mathbf{\Lambda}_k := \text{diag}(\lambda_1, \ldots, \lambda_k)$ and $\mathbf{\Lambda}_{-k} := \text{diag}(\lambda_{k+1}, \ldots, \lambda_d)$.

For all integers $d \geqslant k \geqslant 1$, we denote by $\text{St}(d, k) := \{\mathbf{X} \in \mathbb{R}^{d \times k} : \mathbf{X}^\top \mathbf{X} = \mathbf{I}_k\}$ the set of $d \times k$ column-orthonormal matrices. For all $\mathbf{Y} \in \mathbb{R}^{d \times k}$, we denote by $\text{QR}(\mathbf{Y})$ the QR decomposition of $\mathbf{Y}$, which is a pair of matrices $\mathbf{X}, \mathbf{R}$ such that $\mathbf{Y} = \mathbf{X}\mathbf{R}$, $\mathbf{X} \in \text{St}(d, k)$ and $\mathbf{R} \in \mathbb{R}^{k \times k}$ is an upper triangular matrix with non-negative diagonal coefficients. If $\mathbf{Y}$ is of full column rank, the QR decomposition is unique, $\mathbf{R}$ is invertible, and its diagonal coefficients are positive (Trefethen & Bau, 2022). $\|\cdot\|_2$ and $\|\cdot\|_F$ denote the matrix spectral and Frobenius norms respectively. For a matrix $\mathbf{X}$, we denote by $\sigma_{\min}(\mathbf{X})$ its smallest singular value, and by

$\mathbf{X}^\dagger$ its Moore-Penrose pseudoinverse.

### 1.4. Principal Angles Between Subspaces

In this work, we study algorithms that aim to approximate linear subspaces of $\mathbb{R}^d$. For $\mathbf{X}, \mathbf{U} \in \text{St}(d, k)$, we quantify the distance between $\text{range}(\mathbf{X})$ and $\text{range}(\mathbf{U})$ with principal angles between subspaces.

**Definition 1.1** (Knyazev & Argentati (2002))**.** Let $k \in \{1, \ldots, d-1\}$ and $\mathbf{U}, \mathbf{X} \in \text{St}(d, k)$, and let $1 \geqslant \sigma_1 \geqslant \cdots \geqslant \sigma_k \geqslant 0$ be the singular values of $\mathbf{U}^\top \mathbf{X}$. The principal angles $\theta_1(\mathbf{U}, \mathbf{X}) \leqslant \cdots \leqslant \theta_k(\mathbf{U}, \mathbf{X})$ between the subspaces spanned by the columns of $\mathbf{U}$ and $\mathbf{X}$ are defined as

$$\forall i \in \{1, \ldots, k\}, \quad \theta_i(\mathbf{U}, \mathbf{X}) := \arccos(\sigma_i) \in [0, \pi/2].$$

Intuitively, $\theta_k(\mathbf{U}, \mathbf{X})$ is the smallest $\theta$ such that any unit vector in $\text{range}(\mathbf{U})$ lies within angle $\theta$ of some unit vector in $\text{range}(\mathbf{X})$. Notice that $\theta_k(\mathbf{U}, \mathbf{X}) = 0$ if and only if $\text{range}(\mathbf{U})$ and $\text{range}(\mathbf{X})$ coincide, and that the smaller $\theta_k(\mathbf{U}, \mathbf{X})$ is, the closer the subspaces are. In line with previous works on noisy power methods (Hardt & Price, 2014; Balcan et al., 2016; Xu, 2023), our convergence results are expressed in terms of $\sin \theta_k(\mathbf{U}_k, \mathbf{X}_t)$.

## 2. Accelerated Noisy Power Method

We now introduce the Accelerated Noisy Power Method (ANPM). We want to estimate the top-$k$ eigenspace $\mathbf{U}_k$ of $\mathbf{A} \succeq \mathbf{0}$. However, we assume that we only have access to the approximate product $\mathbf{X} \in \text{St}(d, k) \mapsto \mathbf{A}\mathbf{X} + \mathbf{\Xi}$, where $\mathbf{\Xi}$ is a perturbation. Given a sequence of perturbations $\{\mathbf{\Xi}_t\}_{t \geqslant 0}$, ANPM with momentum parameter $\beta > 0$ is given by

$$\mathbf{X}_0 \in \text{St}(d, k), \quad \mathbf{X}_1, \mathbf{R}_1 = \text{QR}\left(\frac{1}{2}\mathbf{A}\mathbf{X}_0 + \mathbf{\Xi}_0\right),$$

$$\forall t \geqslant 1, \quad \begin{cases} \mathbf{Y}_{t+1} = \mathbf{A}\mathbf{X}_t - \beta \mathbf{X}_{t-1}\mathbf{R}_t^{-1} + \mathbf{\Xi}_t, \\ \mathbf{X}_{t+1}, \mathbf{R}_{t+1} = \text{QR}(\mathbf{Y}_{t+1}). \end{cases} \quad (1)$$

Before presenting our main result, we briefly explain the idea behind the momentum term $-\beta \mathbf{X}_{t-1}\mathbf{R}_t^{-1}$. In the noiseless case (i.e. $\mathbf{\Xi}_t \equiv \mathbf{0}$), the unnormalized iterates $\mathbf{Z}_t := \mathbf{X}_t\mathbf{R}_t \cdots \mathbf{R}_1$ can be written as $\mathbf{Z}_t = p_t(\mathbf{A})\mathbf{X}_0$, where $p_t$ is a degree-$t$ scaled Chebyshev polynomial of the first kind, verifying $p_0(x) = 1$, $p_1(x) = x/2$ and

$$p_{t+1}(x) = xp_t(x) - \beta p_{t-1}(x). \quad (2)$$

Compared to the monomial $x^t$ that would be obtained without momentum (i.e. with the standard Power Method), $p_t$ offers a significantly more favorable ratio between its magnitude on the interval $[-2\sqrt{\beta}, 2\sqrt{\beta}]$ and its growth outside of it. This property is formally stated in the next result:

**Proposition 2.1.** *For all $t \geqslant 1$, $p_t$ satisfies*

$$p_t(x) = \arg \min_{\substack{\deg(p)=t \\ \mathrm{lc}(p)=1/2}} \max_{x \in [-2\sqrt{\beta}, 2\sqrt{\beta}]} |p(x)|,$$

*where $\mathrm{lc}(p)$ denotes the leading coefficient of $p$.*

Assuming that the interval $[-2\sqrt{\beta}, 2\sqrt{\beta}]$ contains only the eigenvalues of $\mathbf{A}$ smaller than $\lambda_k$, Proposition 2.1 implies that the polynomial $p_t$ is better at suppressing the effect of those smaller eigenvalues on the iterates than $x^t$, leading to accelerated convergence towards the top-$k$ eigenspace $\mathbf{U}_k$. We note that other orthogonal polynomials could be used to leverage additional structure in $\mathbf{A}$ (Berthier et al., 2020).

### 2.1. Main Result

Our main result is the following theorem, providing a convergence rate for ANPM under conditions on the noise matrices $\{\mathbf{\Xi}_t\}_{t \geqslant 0}$ and appropriate choices of the parameter $\beta$.

**Theorem 2.2.** *Let $\varepsilon \in (0, 1)$ and $\mathbf{A} \succeq 0$ such that $\lambda_k > \lambda_{k+1}$. Let $\mathbf{X}_0 \in \mathrm{St}(d, k)$ such that $\cos \theta_k(\mathbf{U}_k, \mathbf{X}_0) > 0$, and consider the ANPM iterates $\{\mathbf{X}_t\}_{t \geqslant 0}$ defined by (1) with momentum parameter $\beta > 0$ satisfying $\lambda_k > 2\sqrt{\beta} \geqslant \lambda_{k+1}$ and perturbations $\{\mathbf{\Xi}_t\}_{t \geqslant 0}$ satisfying, for all $t \geqslant 0$,*

$$\|\mathbf{U}_{-k}^\top \mathbf{\Xi}_t\|_2 \leqslant c(\lambda_k - 2\sqrt{\beta})\varepsilon, \tag{3}$$

$$\|\mathbf{U}_k^\top \mathbf{\Xi}_t\|_2 \leqslant c(\lambda_k - 2\sqrt{\beta}) \cos \theta_k(\mathbf{U}_k, \mathbf{X}_t), \tag{4}$$

*with $c := \frac{1}{32}$. Then, for $t \geqslant T$, $\sin \theta_k(\mathbf{U}_k, \mathbf{X}_t) \leqslant \varepsilon$, where*

$$T = \mathcal{O}\left( \sqrt{\frac{\lambda_k}{\lambda_k - 2\sqrt{\beta}}} \log \left( \frac{\tan \theta_k(\mathbf{U}_k, \mathbf{X}_0)}{\varepsilon} \right) \right).$$

We make the following comments regarding our result:

**Convergence rate.** Under our assumptions, the convergence rate of ANPM matches the rate of the noiseless Power Method with Momentum given in Corollary 2 of Xu et al. (2018). The optimal rate is obtained by choosing $\beta = \beta^\star := \lambda_{k+1}^2/4$, giving a convergence rate of order $\tilde{\mathcal{O}}(\sqrt{\lambda_k/(\lambda_k - \lambda_{k+1})})$, which is the optimal worst-case rate achievable by a Krylov subspace method in the noiseless setting. In that case, the rate improves by a square root factor of the eigengap over the non-accelerated Noisy Power Method, yielding substantial speedups for small gaps.

**Noise conditions.** Our conditions on the noise $\{\mathbf{\Xi}_t\}_{t \geqslant 0}$ match those of the Noisy Power Method given in Hardt & Price (2014) when $\beta = \beta^\star$. Our proof highlights the different impact of the two components $\mathbf{U}_k^\top \mathbf{\Xi}_t$ and $\mathbf{U}_{-k}^\top \mathbf{\Xi}_t$ on the convergence of ANPM. The component $\mathbf{U}_{-k}^\top \mathbf{\Xi}_t$ causes a constant term of order $\varepsilon$ to appear in the upper bound on $\sin \theta_k(\mathbf{U}_k, \mathbf{X}_t)$, while the component $\mathbf{U}_k^\top \mathbf{\Xi}_t$ affects a geometrically decaying term in the upper bound. Condition

(3) then ensures that the noise does not make the estimates drift too far away from $\mathbf{U}_k$, while condition (4) ensures that the impact of the noise on the geometric term does not overwhelm the impact of $\mathbf{A}$'s top-$k$ eigenvalues. In comparison to the work of Xu (2023), our noise conditions are significantly milder for small gaps, as shown in Table 1. In particular, our bounds scale proportionally with the gap, while those of Xu (2023) decay exponentially with it. Our conditions (3)-(4) depend on $\cos \theta_k(\mathbf{U}_k, \mathbf{X}_t)$ and involve the components of the noise in the directions $\mathbf{U}_k$ and $\mathbf{U}_{-k}$, which are typically unknown quantities in practice. However, for $\varepsilon \leqslant \tan \theta_k(\mathbf{U}_k, \mathbf{X}_0)$, using Lemma C.10 in the Appendix, one can show that a simple sufficient condition for (3)-(4) to hold is

$$\|\mathbf{\Xi}_t\|_2 \leqslant \mathcal{O}((\lambda_k - 2\sqrt{\beta}) \min(\cos \theta_k(\mathbf{U}_k, \mathbf{X}_0), \varepsilon)),$$

with $\|\mathbf{\Xi}_t\|_2$ and $\cos \theta_k(\mathbf{U}_k, \mathbf{X}_0)$ being generally easier to control. We add that while our results focus on adversarial noise with bounded norm, they can be used in the context of stochastic noise, by using matrix concentration inequalities (see e.g. Tropp (2015)) to ensure that the noise conditions (3)-(4) hold with high probability.

**Computational complexity.** Denote by $c(\mathbf{A})$ the number of operations required to perform a noisy matrix-vector product $\boldsymbol{x} \mapsto \mathbf{A}\boldsymbol{x} + \boldsymbol{\xi}$. Then, the cost of an iteration of ANPM is $\mathcal{O}(kc(\mathbf{A}) + dk^2)$, where the second term is the cost of a QR factorization and of the product $\mathbf{X}_{t-1}\mathbf{R}_t^{-1}$. In particular, the inversion of $\mathbf{R}_t$ is relatively cheap, since it is a triangular matrix of size $k \times k$. This is the same complexity as an iteration of the non-accelerated noisy power method.

**Choice of $\beta$.** Theorem 2.2 requires $\beta$ to belong to the interval $[\lambda_{k+1}^2/4, \lambda_k^2/4)$, which gets smaller as the eigengap decreases, and the optimal choice $\beta^\star$ requires knowledge of $\lambda_{k+1}$, which is a priori unknown. However, we prove in Theorem C.12 in Appendix C.3 that for all $0 < \beta < \lambda_{k+1}^2/4$, ANPM still converges faster than the non-accelerated Noisy Power Method, under the same noise conditions as those of Hardt & Price (2014), showing that there is generally no drawback to using ANPM with smaller values of $\beta$.

**Adaptive $\beta$.** Taking inspiration from Xu (2023), we propose a heuristic to adaptively tune $\beta$. Letting $\mathbf{X}_t$ have $k+1$ columns[2] instead of $k$, we set at each iteration $\beta_t$ as

$$\beta_t = \min_{j=1,\ldots,k+1} [\mathbf{X}_t^\top (\mathbf{A}\mathbf{X}_t + \mathbf{\Xi}_t)]_{j,j}^2/4. \tag{5}$$

Typically, $\beta_t \leqslant \beta^\star$ and $\beta_t$ approaches $\beta^\star$ as $t$ increases. We show in our experiments that this tuning-free method performs similarly to using the optimal value $\beta^\star$ in practice.

**Random initialization.** The condition $\cos \theta_k(\mathbf{U}_k, \mathbf{X}_0) > 0$ is satisfied almost surely when $\mathbf{X}_0$ spans the column space

---

[2]Note that adding a column to $\mathbf{X}_t$ leaves the evolution of the $k$ first columns unchanged.

of a random matrix with i.i.d. standard Gaussian entries. In this case, using Lemma 2.4 from (Hardt & Price, 2014), with probability at least $1 - \tau^{-\Omega(1)} - e^{-\Omega(d)}$, we have that $\tan \theta_k(\mathbf{U}_k, \mathbf{X}_0) \leqslant \frac{\tau \sqrt{d}}{\sqrt{k} - \sqrt{k-1}}$ for all $\tau > 0$.

**Proof sketch.** The full proof of Theorem 2.2 is deferred to Appendix C.2. We provide here a proof sketch. We start by analyzing the evolution of the matrix $\mathbf{H}_t$, defined by

$$\mathbf{H}_t := (\mathbf{U}_{-k}^\top \mathbf{X}_t)(\mathbf{U}_k^\top \mathbf{X}_t)^{-1} \in \mathbb{R}^{(d-k) \times k},$$

whose spectral norm is $\tan \theta_k(\mathbf{U}_k, \mathbf{X}_t)$ (see Proposition A.1 in the Appendix). This matrix is convenient to study, as its homogeneous structure allows us to write it as $\mathbf{H}_t = (\mathbf{U}_{-k}^\top \mathbf{Y}_t)(\mathbf{U}_k \mathbf{Y}_t)^{-1}$. We can then derive the following three-term recurrence relation for $\mathbf{H}_t$ using the ANPM iteration (1) linking $\mathbf{Y}_{t+1}$ to $\mathbf{X}_t$, $\mathbf{X}_{t-1}$ and $\mathbf{R}_t$:

$$\mathbf{H}_{t+1}\mathbf{C}_{t+1} = \mathbf{\Lambda}_{-k}\mathbf{H}_t\mathbf{C}_t - \beta\mathbf{H}_{t-1}\mathbf{C}_{t-1} + \mathbf{\Psi}_t\mathbf{C}_t,$$

where $\mathbf{\Psi}_t := (\mathbf{U}_{-k}^\top \mathbf{\Xi}_t)(\mathbf{U}_k^\top \mathbf{X}_t)^{-1}$ is a noise term controlled by condition (3), $\mathbf{C}_t$ satisfies

$$\mathbf{C}_{t+1} = \mathbf{\Lambda}_k\mathbf{C}_t - \beta\mathbf{C}_{t-1} + \mathbf{\Lambda}_k\mathbf{E}_t\mathbf{C}_t,$$

and $\mathbf{E}_t := \mathbf{\Lambda}_k(\mathbf{U}_k^\top \mathbf{\Xi}_t)(\mathbf{U}_k^\top \mathbf{X}_t)^{-1}$ is a noise term controlled by condition (4). These recursions allow us respectively to express $\mathbf{H}_t$ in terms of scaled Chebyshev polynomials verifying (2), and to tightly control the spectral norm of the factors depending on $\mathbf{C}_t$. This leads to an upper bound on $\tan \theta_k(\mathbf{U}_k, \mathbf{X}_t)$ that can be decomposed as the sum of a constant term of order $\varepsilon$, stemming from $\mathbf{\Psi}_t$, and a term that decays geometrically at a rate $\tilde{\mathcal{O}}(\sqrt{\lambda_k/(\lambda_k - 2\sqrt{\beta})})$.

The key argument in our proof lies in a precise analysis the evolution of the matrix $\mathbf{H}_t$, enabled by the introduction of the sequence $\{\mathbf{C}_t\}$. This allows us to sharply control the impact of the noise on the convergence of $\tan \theta_k(\mathbf{U}_k, \mathbf{X}_t)$. In contrast, Xu (2023)'s proof instead starts by analyzing the evolution of $\mathbf{X}_t$, which requires to use coarse upper bounds on $\|\mathbf{R}_t\|_2$ depending on $\lambda_1$ to derive upper bounds on $\|\mathbf{H}_t\|_2$. This leads to the suboptimal noise conditions involving $\mu_k$ and $\mu_{k+1}$, as defined in Table 1.

## 2.2. Complexity Lower Bounds, Tightness of the Noise Conditions

Our improved analysis provides milder noise conditions than Xu (2023)'s. The theorems in this section show that our analysis is in fact tight (up to constants), in the sense that we can exhibit instances of ANPM that 1) satisfy (3) and (4), and need at least $\tilde{\Omega}(\sqrt{\lambda_k/(\lambda_k - 2\sqrt{\beta})})$ iterations to reach $\tan \theta_k(\mathbf{U}_k, \mathbf{X}_t) \leqslant \varepsilon$ and 2) satisfy either one of the conditions (3) or (4) with a constant larger than $c$, and fail to reach $\tan \theta_k(\mathbf{U}_k, \mathbf{X}_t) \leqslant \varepsilon$ in any number of iterations. For all of the theorems in this section, we let $\lambda_k > 2\sqrt{\beta} >$

0. All of our results are based on ANPM on the matrix $\mathbf{A} := \text{diag}(\lambda_k, \ldots, \lambda_k, 2\sqrt{\beta}, \ldots, 2\sqrt{\beta})$. The first result shows that even with no noise, the iteration complexity in $\tilde{\mathcal{O}}(\sqrt{\lambda_k/(\lambda_k - 2\sqrt{\beta})})$ generally cannot be improved.

**Theorem 2.3** (Complexity lower bound). *Let $\varepsilon \in (0, 1)$ and $\mathbf{X}_0 \in \text{St}(d, k)$ such that $\cos \theta_k(\mathbf{U}_k, \mathbf{X}_0) > 0$, and consider the ANPM iterates $\{\mathbf{X}_t\}_{t \geqslant 0}$ defined by (1) with momentum parameter $\beta$ and perturbations $\mathbf{\Xi}_t \equiv \mathbf{0}$. Then, for all $t < T$, $\tan \theta_k(\mathbf{U}_k, \mathbf{X}_t) > \varepsilon$, where*

$$T = \Omega\left(\sqrt{\frac{\lambda_k}{\lambda_k - 2\sqrt{\beta}}} \log\left(\frac{\tan \theta_k(\mathbf{U}_k, \mathbf{X}_0)}{\varepsilon}\right)\right).$$

The next two results respectively show the tightness of the noise conditions (3) and (4). Indeed, in each theorem, we exhibit an instance of ANPM where $\mathbf{\Xi}_t$ satisfies one of the two noise conditions with a larger constant than in Theorem 2.2, and such that $\tan \theta_k(\mathbf{U}_k, \mathbf{X}_t) \leqslant \varepsilon$ is never reached.

**Theorem 2.4** (Tightness of condition (3)). *Let $\varepsilon \in (0, 1)$. There exists $\mathbf{X}_0 \in \text{St}(d, k)$ such that $\cos \theta_k(\mathbf{U}_k, \mathbf{X}_0) > 0$ and perturbations $\{\mathbf{\Xi}_t\}_{t \geqslant 0}$ verifying $\mathbf{U}_k^\top \mathbf{\Xi}_t = \mathbf{0}$ and*

$$\|\mathbf{U}_{-k}^\top \mathbf{\Xi}_t\|_2 \leqslant 8(\lambda_k - 2\sqrt{\beta})\varepsilon,$$

*such that the ANPM iterates $\{\mathbf{X}_t\}_{t \geqslant 0}$ defined by (1) with momentum $\beta$ verify $\tan \theta_k(\mathbf{U}_k, \mathbf{X}_t) > \varepsilon$ for all $t \geqslant 0$.*

**Theorem 2.5** (Tightness of condition (4)). *Let $\varepsilon \in (0, 1)$. There exists $\mathbf{X}_0 \in \text{St}(d, k)$ such that $\cos \theta_k(\mathbf{U}_k, \mathbf{X}_0) > 0$ and perturbations $\{\mathbf{\Xi}_t\}_{t \geqslant 0}$ verifying $\mathbf{U}_{-k}^\top \mathbf{\Xi}_t = \mathbf{0}$ and*

$$\|\mathbf{U}_k^\top \mathbf{\Xi}_t\|_2 \leqslant (\lambda_k - 2\sqrt{\beta}) \cos \theta_k(\mathbf{U}_k, \mathbf{X}_t),$$

*such that the ANPM iterates $\{\mathbf{X}_t\}_{t \geqslant 0}$ defined by (1) with momentum $\beta$ verify $\tan \theta_k(\mathbf{U}_k, \mathbf{X}_t) > \varepsilon$ for all $t \geqslant 0$.*

The proofs for these results can be found in Appendix C.4. The result from Theorem 2.3 is not surprising: it corresponds to the worst-case complexity of block Krylov methods for top-$k$ PCA in the noiseless setting. The results from Theorems 2.4 and 2.5 provide insights on the impact of the two noise components $\mathbf{U}_{-k}^\top \mathbf{\Xi}_t$ and $\mathbf{U}_k^\top \mathbf{\Xi}_t$ on the evolution of $\{\mathbf{X}_t\}$. In the proof of Theorem 2.4, we show that a sufficiently large component $\mathbf{U}_{-k}^\top \mathbf{\Xi}_t$ makes the estimates drift away from the subspace spanned by $\mathbf{U}_k$, preventing convergence. In the proof of Theorem 2.5, we show that a sufficiently large component $\mathbf{U}_k^\top \mathbf{\Xi}_t$ can effectively render ANPM equivalent to a Power Method with momentum on a matrix with eigengap 0, which does not converge to $\mathbf{U}_k$.

## 3. Application to Decentralized PCA

We now apply our results on ANPM to the problem of decentralized PCA. We consider a connected undirected graph

*Table 2.* Number of iterations $T$ and number of gossip rounds per iteration $L$ required for decentralized algorithms to reach $\sin \theta_k(\mathbf{U}_k, \mathbf{X}_{i,t}) \leqslant \varepsilon$. Here, $M := \max_{i=1,\dots,n} \|\mathbf{A}_i\|_2$ and $\gamma_\mathbf{W}$ is defined in Definition 3.1. The third row corresponds to applying the results of Xu (2023) to ADePM, while the last row corresponds to Theorem 3.3. [†]Results for the optimal parameter $\beta = \lambda_{k+1}^2/4$.

| Algorithm | $T$ | $L$ |
|---|---|---|
| DePM (Wai et al., 2017) | $\mathcal{O}\left(\frac{\lambda_k}{\lambda_k - \lambda_{k+1}} \log\left(\frac{1}{\varepsilon}\right)\right)$ | $\mathcal{O}\left(\frac{1}{\sqrt{\gamma_\mathbf{W}}} \log\left(\frac{M}{\lambda_k} \frac{\lambda_k}{\lambda_k - \lambda_{k+1}} \frac{1}{\varepsilon}\right)\right)$ |
| DeEPCA (Ye & Zhang, 2021) | $\mathcal{O}\left(\frac{\lambda_k}{\lambda_k - \lambda_{k+1}} \log\left(\frac{1}{\varepsilon}\right)\right)$ | $\mathcal{O}\left(\frac{1}{\sqrt{\gamma_\mathbf{W}}} \log\left(\frac{M}{\lambda_k} \frac{\lambda_k}{\lambda_k - \lambda_{k+1}}\right)\right)$ |
| ADePM (using (Xu, 2023))[†] | $\mathcal{O}\left(\sqrt{\frac{\lambda_k}{\lambda_k - \lambda_{k+1}}} \log\left(\frac{1}{\varepsilon}\right)\right)$ | $\mathcal{O}\left(\frac{\log(\lambda_1/\lambda_{k+1})}{\sqrt{\gamma_\mathbf{W}}} \sqrt{\frac{\lambda_k}{\lambda_k - \lambda_{k+1}}} \log\left(\frac{M}{\lambda_k} \frac{\lambda_k}{\lambda_k - \lambda_{k+1}} \frac{1}{\varepsilon}\right)\right)$ |
| ADePM (Theorem 3.3)[†] | $\mathcal{O}\left(\sqrt{\frac{\lambda_k}{\lambda_k - \lambda_{k+1}}} \log\left(\frac{1}{\varepsilon}\right)\right)$ | $\mathcal{O}\left(\frac{1}{\sqrt{\gamma_\mathbf{W}}} \log\left(\frac{M}{\lambda_k} \frac{\lambda_k}{\lambda_k - \lambda_{k+1}} \frac{1}{\varepsilon}\right)\right)$ |

$G = (V, E)$ with $V := \{1, \dots, n\}$, representing a decentralized communication network with $n$ agents. Each agent $i \in V$ has access locally to a matrix-vector product $\boldsymbol{x} \mapsto \mathbf{A}_i \boldsymbol{x}$. The objective of decentralized PCA is to compute the top-$k$ eigenspace of the matrix $\mathbf{A} := n^{-1} \sum_{i=1}^n \mathbf{A}_i \succeq \mathbf{0}$ through local computations and communications between neighboring agents only. This setting arises for instance when a dataset $\boldsymbol{\Phi} = [\boldsymbol{\Phi}_1^\top, \dots, \boldsymbol{\Phi}_n^\top]^\top \in \mathbb{R}^{m \times d}$ is distributed over $G$ so that agent $i \in V$ locally holds $\boldsymbol{\Phi}_i \in \mathbb{R}^{m_i \times d}$ with $\sum_{i=1}^n m_i = m$. The goal is then to estimate the principal components of the empirical covariance matrix $\mathbf{A} = \frac{1}{m} \boldsymbol{\Phi}^\top \boldsymbol{\Phi} = \frac{1}{n} \sum_{i=1}^n \mathbf{A}_i$, where $\mathbf{A}_i := \frac{n}{m} \boldsymbol{\Phi}_i^\top \boldsymbol{\Phi}_i$. We provide another application in our experimental section (Section 4.2) to decentralized spectral clustering.

### 3.1. Gossip Algorithms

The method we propose for decentralized PCA is based on the idea of approximating at each iteration the matrix vector product $\boldsymbol{x} \mapsto \mathbf{A}\boldsymbol{x}$ using only neighbor-to-neighbor communications. Gossip algorithms (Boyd et al., 2006) are iterative methods for decentralized averaging over networks. At each iteration, each agent $i$ performs a weighted averaging of their estimate with those of their neighbors $j \in \mathcal{N}_i$. These weights define the gossip matrix:

**Definition 3.1.** A gossip matrix $\mathbf{W} \in \mathbb{R}^{n \times n}$ is a symmetric matrix with non-negative coefficients which is doubly stochastic (i.e. $\mathbf{W}\mathbf{1} = \mathbf{W}^\top\mathbf{1} = \mathbf{1}$) and such that for all $i, j \in \{1, \dots, n\}$, $w_{i,j} > 0$ if and only if $i = j$ or $(i, j) \in E$. We define its absolute spectral gap[3] as $\gamma_\mathbf{W} := 1 - \max\{|\lambda_2(\mathbf{W})|, |\lambda_n(\mathbf{W})|\} \in (0, 1]$, where $1 = \lambda_1(\mathbf{W}) \geqslant \cdots \geqslant \lambda_n(\mathbf{W})$ are the eigenvalues of $\mathbf{W}$.

The convergence speed of each agent's estimate to the network-wide average depends on the spectral gap $\gamma_\mathbf{W}$ of the gossip matrix. For our decentralized PCA application, we will use an accelerated gossip algorithm introduced in (Liu & Morse, 2011) which is described in Algorithm 1.

[3]From the Perron-Frobenius theorem (see e.g. Chapter 7 from Meyer (2023)), we have $1 > |\lambda_i(\mathbf{W})|$ for all $i = 2, \dots, n$.

---

**Algorithm 1** Accelerated Gossip

**Require:** Gossip matrix $\mathbf{W} \in \mathbb{R}^{n \times n}$, $L \geqslant 1$, initialization $\{\mathbf{Y}_{i,0}\}_{i=1}^n = \{\mathbf{Y}_{i,-1}\}_{i=1}^n$ in $\mathbb{R}^{d \times k}$.

1: $\omega := \frac{1 - \sqrt{\gamma_\mathbf{W}(2 - \gamma_\mathbf{W})}}{1 + \sqrt{\gamma_\mathbf{W}(2 - \gamma_\mathbf{W})}}$
2: **for** $\ell = 0$ to $L - 1$ **do**
3:     **for** each agent $i \in \{1, \dots, n\}$ **in parallel do**
4:         $\mathbf{Y}_{i,\ell+1} = (1+\omega) \sum_{j \in \mathcal{N}_i \cup \{i\}} w_{i,j} \mathbf{Y}_{j,\ell} - \omega \mathbf{Y}_{i,\ell-1}$
5:     **end for**
6: **end for**

---

Instead of simply performing a weighted averaging at each iteration with their neighbors, each agent adds a momentum term to the weighted average. As shown in Proposition 3.2, this allows the algorithm to converge at the rate $\tilde{\mathcal{O}}(1/\sqrt{\gamma_\mathbf{W}})$ instead of the standard $\tilde{\mathcal{O}}(1/\gamma_\mathbf{W})$ rate achieved by classical gossip algorithms (Boyd et al., 2006), thus reducing the communication costs of our method.

**Proposition 3.2** (Ye & Zhang (2021)). *Let* $\bar{\mathbf{Y}} := n^{-1} \sum_{i=1}^n \mathbf{Y}_{i,0}$. *For all* $L \geqslant 1$, *for all agents* $i \in \{1, \dots, n\}$, *Algorithm 1 outputs* $\mathbf{Y}_{i,L}$ *satisfying*

$$\|\mathbf{Y}_{i,L} - \bar{\mathbf{Y}}\|_\mathrm{F} \leqslant (1 - \sqrt{\gamma_\mathbf{W}})^L \sqrt{n} \max_{j=1,\dots,n} \|\mathbf{Y}_{j,0} - \bar{\mathbf{Y}}\|_\mathrm{F}.$$

### 3.2. ADePM: Accelerated Decentralized Power Method

We now present our Accelerated Decentralized Power Method (ADePM) for decentralized PCA, which is described in Algorithm 2. The idea is to approximate at each iteration the matrix vector product $\boldsymbol{x} \mapsto \mathbf{A}\boldsymbol{x}$ through gossiping. Each agent $i$ maintains a local estimate $\mathbf{X}_{i,t} \in \mathrm{St}(d, k)$ of the top-$k$ eigenspace of $\mathbf{A}$, and at each iteration performs a local matrix vector product with $\mathbf{A}_i$, adds momentum, and gossips to approximate the average over the network. The next theorem provides convergence guarantees for ADePM.

**Theorem 3.3.** *Let* $\varepsilon \in (0, 1)$ *and* $\{\mathbf{A}_i\}_{i=1}^n$ *be matrices in* $\mathbb{R}^{d \times d}$ *locally held by each node in* $G$, *and let* $\mathbf{A} := n^{-1} \sum_{i=1}^n \mathbf{A}_i \succeq \mathbf{0}$ *such that* $\lambda_k > \lambda_{k+1}$. *Let* $\mathbf{X}_0 \in \mathrm{St}(d, k)$ *such that* $\cos \theta_k(\mathbf{U}_k, \mathbf{X}_0) > 0$, *and con-*

**Algorithm 2** ADePM

---

**Require:** Gossip matrix $\mathbf{W} \in \mathbb{R}^{n \times n}$, $\beta > 0$, $L \geqslant 1$, $T \geqslant 1$, $\mathbf{X}_0 \in \mathrm{St}(d, k)$.

    **Initialization:**
1: $\forall i = 1, \ldots, n$, $\mathbf{X}_{i,0} = \mathbf{X}_0$
2: $\{\mathbf{Y}_{i,1}\}_{i=1}^n = \mathrm{AccGossip}(\mathbf{W}, L, \{\frac{1}{2}\mathbf{A}_i\mathbf{X}_0\}_{i=1}^n)$
3: **for** each agent $i \in \{1, \ldots, n\}$ **in parallel do**
4:     $\mathbf{X}_{i,1}, \mathbf{R}_{i,1} = \mathrm{QR}(\mathbf{Y}_{i,1})$
5: **end for**
    **Iterations:**
6: **for** $t = 1$ to $T - 1$ **do**
7:     **for** each agent $i \in \{1, \ldots, n\}$ **in parallel do**
8:         $\mathbf{Y}_{i,t+1/2} = \mathbf{A}_i\mathbf{X}_{i,t} - \beta\mathbf{X}_{i,t-1}\mathbf{R}_{i,t}^{-1}$
9:     **end for**
10:    $\{\mathbf{Y}_{i,t+1}\}_{i=1}^n = \mathrm{AccGossip}(\mathbf{W}, L, \{\mathbf{Y}_{i,t+1/2}\}_{i=1}^n)$
11:    **for** each agent $i \in \{1, \ldots, n\}$ **in parallel do**
12:       $\mathbf{X}_{i,t+1}, \mathbf{R}_{i,t+1} = \mathrm{QR}(\mathbf{Y}_{i,t+1})$
13:    **end for**
14: **end for**

---

*sider the ADePM iterates $\{\mathbf{X}_{i,t}\}$ given by Algorithm 2 with momentum $\beta > 0$ satisfying $\lambda_k > 2\sqrt{\beta} \geqslant \lambda_{k+1}$. Assume that the number of gossip rounds per iteration $L$ satisfies*

$$L \geqslant \mathcal{O}\left(\frac{1}{\sqrt{\gamma_\mathbf{W}}} \log\left(\frac{M}{\lambda_k} \frac{\lambda_k}{\lambda_k - 2\sqrt{\beta}} \frac{\tan\theta_k(\mathbf{U}_k, \mathbf{X}_0)}{\varepsilon}\right)\right)$$

*where $M := \max_i \|\mathbf{A}_i\|_2$. Then, for all $i \in \{1, \ldots, n\}$, for all $t \geqslant T$, we have $\sin\theta_k(\mathbf{U}_k, \mathbf{X}_{i,t}) \leqslant \varepsilon$, where*

$$T = \mathcal{O}\left(\sqrt{\frac{\lambda_k}{\lambda_k - 2\sqrt{\beta}}} \log\left(\frac{\tan\theta_k(\mathbf{U}_k, \mathbf{X}_0)}{\varepsilon}\right)\right).$$

**Convergence rate.** For the optimal parameter $\beta = \beta^\star = \lambda_{k+1}^2/4$, ADePM converges at the accelerated rate $\tilde{\mathcal{O}}(\sqrt{\lambda_k/(\lambda_k - \lambda_{k+1})})$, significantly improving over the standard rate $\tilde{\mathcal{O}}(\lambda_k/(\lambda_k - \lambda_{k+1}))$ reached by other classical methods. We are not aware of any other decentralized PCA algorithm achieving this accelerated rate.

**Communication cost.** In comparison to other decentralized power methods (Wai et al., 2017; Ye & Zhang, 2021), ADePM requires a comparable number of gossip steps per iteration, as shown in Table 2. The communication costs are negatively impacted by small eigengaps $1 - \lambda_{k+1}/\lambda_k$, client heterogeneity (which is quantified by the constant $M$), and poorly connected communication networks (i.e. small values of $\gamma_\mathbf{W}$). We show in Table 2 the communication costs of ADePM had we used the result from Xu (2023), which represents a significant increase over our result and over previous decentralized algorithms. This shows the importance of our refined analysis of ANPM for the design of communication-efficient decentralized algorithms.

*Remark* 3.4. Ye & Zhang (2021) achieve a communication cost $L$ independent of $\varepsilon$ using a subspace tracking technique, relying on tight inequalities. While we do not consider such methods in this paper, our tight analysis of ANPM would be a necessary first step towards accelerating DeEPCA.

The proof for Theorem 3.3 is deferred to Appendix D. The idea is to use Theorem 2.2 to obtain the convergence rate. To do so, we define a "network-average" iterate $\bar{\mathbf{X}}_t$ which remains close to all local estimates $\mathbf{X}_{i,t}$ and follows the ANPM iteration on the average matrix $\mathbf{A}$ and with noise $\mathbf{\Xi}_t$ induced by the gossiping errors. The remainder of the then proof consists in establishing a relation between the number of gossip communications $L$ at each step and the magnitude of the noise $\mathbf{\Xi}_t$, to show that the conditions (3)-(4) are satisfied whenever $L$ satisfies the assumption in Theorem 3.3. These relations are derived from Proposition 3.2, and from perturbation bounds on the QR decomposition.

# 4. Experiments

We provide experimental results for ANPM on synthetic instances, and for ADePM on real datasets. More details on the experimental setups and additional experimental results are provided in Appendix E. The code used for the experiments is available at https://github.com/pierreaguie/ANPM.

## 4.1. ANPM

We conduct experiments on synthetic datasets for NPM and ANPM. The aim is to highlight the impact of the eigengap $\Delta_k := 1 - \lambda_{k+1}/\lambda_k$, the norm of the noise $\xi := \|\mathbf{\Xi}_t\|_2$, and the momentum parameter $\beta$ on the convergence speed and final precision of (A)NPM. Here, the noise $\mathbf{\Xi}_t$ is sampled randomly using a distribution inspired by the adversarial examples used for the proofs of Section 2.2. We show in Figure 1 the impact of these parameters on the evolution of $\sin\theta_k(\mathbf{U}_k, \mathbf{X}_t)$ by varying $\beta$, $\Delta_k$ and $\xi$, all other parameters being fixed. $\beta^\star = \lambda_{k+1}^2/4$ refers to the optimal momentum parameter, $\beta_c := \lambda_k^2/4$ to the upper bound on valid choices of $\beta$ in Theorem 2.2, and $\beta_t$ to the adaptive tuning heuristic defined in (5). More details on the synthetic instance generation are provided in Appendix E.1. We make several comments on our results:

**Transient and stationary regimes.** All plots shown in Figure 1 display a transient regime, in which $\sin\theta_k(\mathbf{U}_k, \mathbf{X}_t)$ decays geometrically, and a stationary regime, in which it stays almost constant at a final accuracy $\varepsilon$, which depends on $\Delta_k$ and $\xi$. This was expected from our proof of Theorem 2.2.

**Impact of $\beta$.** ANPM with $\beta = \beta^\star$ significantly improves the convergence speed over NPM (i.e. $\beta = 0$), especially for small eigengaps, while leaving the final accuracy unchanged. This confirms our theoretical results, which show

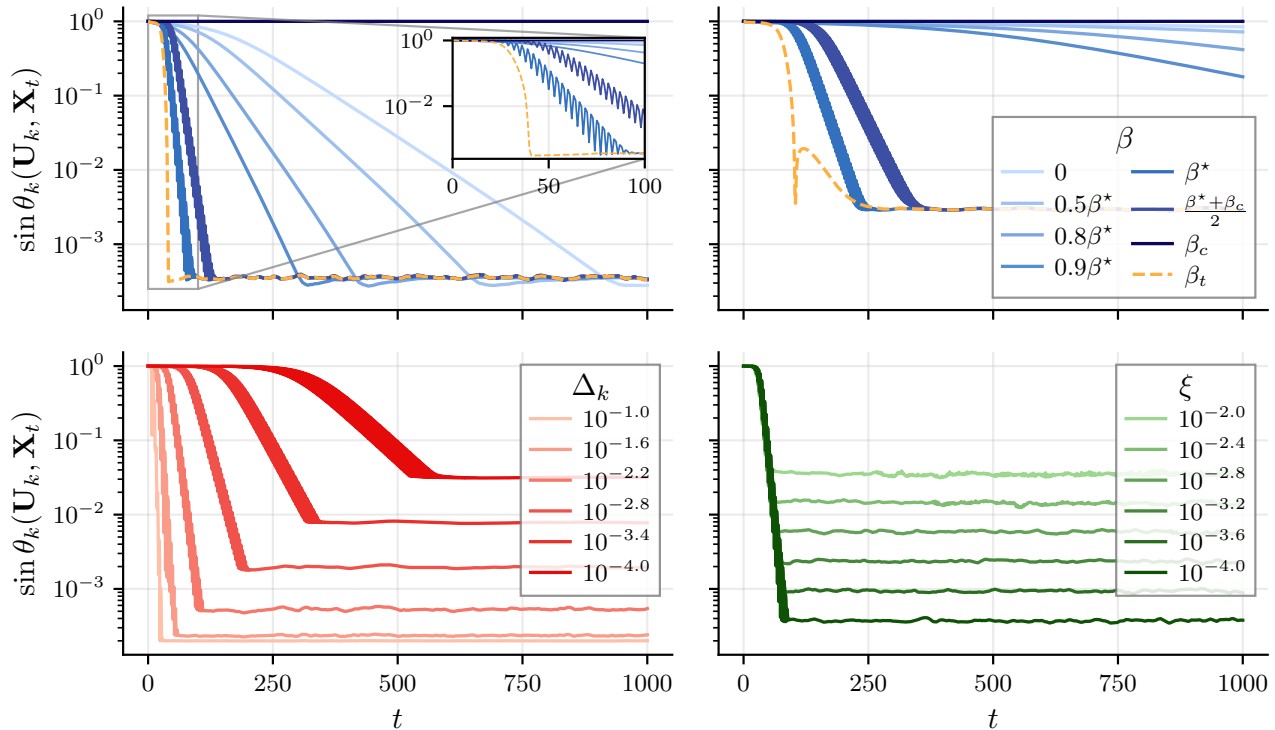

*Figure 1.* Results for (A)NPM. **(Top left)** fixed $\xi = 10^{-4}$ and $\Delta_k = 10^{-2}$, varying $\beta$; **(Top right)** fixed $\xi = 10^{-4}$ and $\Delta_k = 10^{-3}$, varying $\beta$; **(Bottom left)** fixed $\xi = 10^{-4}$ and $\beta = \beta^{\star}(\Delta_k)$, varying $\Delta_k$; **(Bottom right)** fixed $\Delta_k = 10^{-2}$ and $\beta = \beta^{\star}$, varying $\xi$.

that acceleration comes at no extra cost in terms of final precision in comparison to NPM. We note that smaller values $0 < \beta < \beta^{\star}$ and larger values $\beta^{\star} < \beta < \beta_c$ still lead to faster convergence than NPM (though slower than the optimal tuning), and that setting $\beta = \beta_c$ does not allow the algorithm to converge, suggesting that the interval of valid $\beta$ values in Theorem 2.2 cannot be improved. The tuning heuristic $\beta_t$ attains similar convergence speed to the optimal tuning. For larger values of $\beta$, we observe oscillations in the transient regime, corresponding to the oscillatory behavior of the Chebyshev polynomials $p_t$ defined in (2) in the interval $[-2\sqrt{\beta}, 2\sqrt{\beta}]$.

**Impact of the eigengap.** Smaller gaps lead to slower convergence and worse final accuracies at fixed noise magnitude. The relationship between final accuracy and gap in Figure 1 is near linear (as suggested by Theorem 2.2), except between $\Delta_k = 10^{-1}$ and $10^{-1.6}$, which we suspect is due to the fact that the component in $\mathrm{range}(\mathbf{U}_{-k})$ of the noise we generate is not fully contained in $\mathrm{Span}(\boldsymbol{u}_{k+1})$.

**Impact of the noise magnitude.** The final accuracy scales proportionally with $\xi$, as suggested by Theorem 2.2. On the ranges of noise norm $\xi$ considered, the convergence rate in the transient regime is not significantly impacted by $\xi$.

### 4.2. ADePM

We present results for ADePM on decentralized PCA on the Fed-Heart-Disease dataset from FLamby (Ogier du Terrail et al., 2022) and two different splits (homogeneous and heterogeneous) of the digits dataset (Alpaydin & Kaynak, 1998), and on decentralized spectral clustering on a subset of the Ego-Facebook graph from (Leskovec & Mcauley, 2012). More details are provided in Appendix E.3. We compare ADePM to DePM (Wai et al., 2017) and DeEPCA (Ye & Zhang, 2021). The results are shown in Figure 2. $\beta_t$ refers to an adaptation of the heuristic defined in (5) to the decentralized setting, which is detailed in Appendix E.3.

**Communication costs.** Just like for ANPM, we observe for DePM and ADePM an exponentially decaying transient regime, followed by a stationary regime where the error stabilizes, due to the dependence of the final accuracy on the number of gossip communications $L$. The final accuracy reached is roughly the same for DePM and ADePM at fixed $L$. DeEPCA does not reach a stationary regime, which is consistent with the independence of $L$ from the target accuracy $\varepsilon$ for this algorithm.

**Impact of heterogeneity.** For the digits dataset, we consider two ways of splitting the data across agents: one where the data is split randomly across agents (homogeneous split),

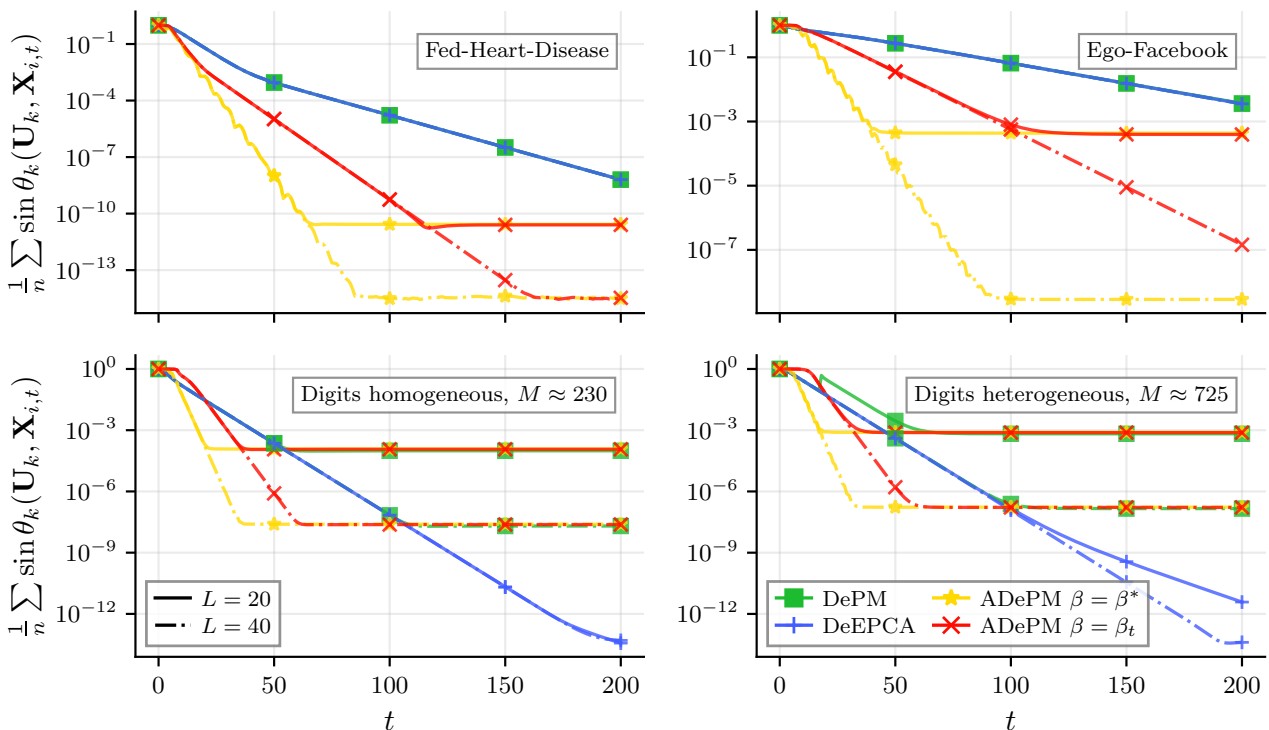

*Figure 2.* Results for decentralized PCA.

and one where each agent only has access to data points corresponding to a specific digit (heterogeneous split). $M$ is significantly larger in the heterogeneous setting. The impact of heterogeneity is reflected in the final accuracy reached at fixed $L$, which is worse in the heterogeneous setting than in the homogeneous one for all algorithms and all values of $L$.

**Convergence speed.** Both versions of ADePM (with fixed optimal $\beta = \beta^\star$ or with adaptive $\beta = \beta_t$) significantly outperform DePM and DeEPCA in terms of convergence speed. In scenarios where fast convergence is prioritized over final accuracy, ADePM is a better choice than DeEPCA.

## 5. Conclusion

We provided convergence guarantees for ANPM, showing that it converges at an accelerated rate under milder noise conditions than previous analyses. We showed that our analysis is tight, and applied our results to design ADePM, an accelerated algorithm for decentralized PCA with comparable communication costs to non-accelerated decentralized algorithms. While our work is mainly of theoretical nature, our experimental results show that using heuristics to adaptively tune the momentum can lead to significant speedups over non-accelerated methods without requiring manual parameter tuning.

Balcan et al. (2016) and Xu (2023) provide convergence rates for (A)NPM that depend on the wider gap $\lambda_k - \lambda_{p+1}$ whenever $\mathbf{X}_t$ has $p$ columns but only the top-$k$ eigenspace is estimated, with $p > k$. This can represent significant improvements in terms of convergence speed and noise conditions in some cases. Extending our analysis to this setting is an interesting direction for future work.

## Acknowledgements

PA acknowledges funding from PEPR IA (grant REDEEM ANR-23-PEIA-0005). LM acknowledges funding from PR[AI]RIE-PSAI – Paris School of Artificial Intelligence, reference ANR-23-IACL-0008.

## Impact Statement

This paper presents work whose goal is to advance the field of Machine Learning. There are many potential societal consequences of our work, none of which we feel must be specifically highlighted here.

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

# A. Useful Linear Algebra Results

## A.1. Formulas for Principal Angles Between Subspaces

We provide here useful formulas for the cosines, sines and tangents of the principal angles between two subspaces in terms of their orthonormal bases. These formulas will be used extensively in our proofs, in particular to control the evolution of $\tan \theta_k(\mathbf{U}_k, \mathbf{X}_t)$ along the iterations of the algorithms we study.

**Proposition A.1** (Stewart & Sun (1990), Corollary 5.4)**.** *Let* $\mathbf{U}, \mathbf{X} \in \mathrm{St}(d, k)$*, and let* $\mathbf{V} \in \mathrm{St}(d, d-k)$ *be a matrix whose columns span the orthogonal complement of the range of* $\mathbf{U}$*. Then,*

$$\cos \theta_k(\mathbf{U}, \mathbf{X}) = \sigma_{\min}(\mathbf{U}^\top \mathbf{X}),$$
$$\sin \theta_k(\mathbf{U}, \mathbf{X}) = \|\mathbf{V}^\top \mathbf{X}\|_2.$$

*If* $\cos \theta_k(\mathbf{U}, \mathbf{X}) > 0$*,* $\mathbf{U}^\top \mathbf{X}$ *is invertible and*

$$\tan \theta_k(\mathbf{U}, \mathbf{X}) = \|(\mathbf{V}^\top \mathbf{X})(\mathbf{U}^\top \mathbf{X})^{-1}\|_2.$$

We give here a simple proof of these formulas for completeness.

*Proof.* By definition of the $k$-th principal angle, we have that $\theta_k(\mathbf{U}, \mathbf{X}) = \arccos(\sigma_k(\mathbf{U}^\top \mathbf{X}))$, where $\sigma_k(\mathbf{U}^\top \mathbf{X})$ is the $k$-th largest (i.e. the smallest) singular value of $\mathbf{U}^\top \mathbf{X}$. This proves the first part of the proposition.

Let $\mathbf{P} \in \mathrm{St}(k, k), \mathbf{Q} \in \mathrm{St}(k, k)$ be the left and right singular vectors of $\mathbf{U}^\top \mathbf{X}$ such that $\mathbf{U}^\top \mathbf{X} = \mathbf{P} \boldsymbol{\Sigma} \mathbf{Q}^\top$, where $\boldsymbol{\Sigma} := \mathrm{diag}(\sigma_1(\mathbf{U}^\top \mathbf{X}), \dots, \sigma_k(\mathbf{U}^\top \mathbf{X})) = \mathrm{diag}(\cos \theta_1(\mathbf{U}, \mathbf{X}), \dots, \cos \theta_k(\mathbf{U}, \mathbf{X}))$. Since $\mathbf{V}$ spans the orthogonal complement of the range of $\mathbf{U}$, we have $\mathbf{V}\mathbf{V}^\top + \mathbf{U}\mathbf{U}^\top = \mathbf{I}_d$. Then, we can check that the right singular vectors of $\mathbf{V}^\top \mathbf{X}$ are also $\mathbf{Q}$ and that its singular values are $\sigma_i(\mathbf{V}^\top \mathbf{X}) = \sin \theta_i(\mathbf{U}, \mathbf{X})$ for all $i \in \{1, \dots, k\}$. Indeed,

$$\begin{aligned} \mathbf{X}^\top \mathbf{V}\mathbf{V}^\top \mathbf{X} = \mathbf{X}^\top (\mathbf{I}_d - \mathbf{U}\mathbf{U}^\top)\mathbf{X} &= \mathbf{I}_k - \mathbf{X}^\top \mathbf{U}\mathbf{U}^\top \mathbf{X} = \mathbf{Q}(\mathbf{I}_k - \boldsymbol{\Sigma}^2)\mathbf{Q}^\top, \\ &= \mathbf{Q}\mathrm{diag}\left(1 - \cos^2 \theta_1(\mathbf{U}, \mathbf{X}), \dots, 1 - \cos^2 \theta_k(\mathbf{U}, \mathbf{X})\right)\mathbf{Q}^\top \\ &= \mathbf{Q}\underbrace{\mathrm{diag}\left(\sin \theta_1(\mathbf{U}, \mathbf{X}), \dots, \sin \theta_k(\mathbf{U}, \mathbf{X})\right)^2}_{\boldsymbol{\Sigma}'}\mathbf{Q}^\top. \end{aligned}$$

From this, we deduce that the largest singular value of $\mathbf{V}^\top \mathbf{X}$ is $\sin \theta_k(\mathbf{U}, \mathbf{X})$.

Then, assuming that $\cos \theta_k(\mathbf{U}, \mathbf{X}) > 0$, $\mathbf{U}^\top \mathbf{X}$'s smallest singular value is positive and it is thus invertible. Denote by $\mathbf{P}' \in \mathrm{St}(d-k, k)$ the left singular vectors of $\mathbf{V}^\top \mathbf{X}$ such that $\mathbf{V}^\top \mathbf{X} = \mathbf{P}' \boldsymbol{\Sigma}' \mathbf{Q}^\top$. We then have

$$(\mathbf{V}^\top \mathbf{X})(\mathbf{U}^\top \mathbf{X})^{-1} = \mathbf{P}' \boldsymbol{\Sigma}' \mathbf{Q}^\top \mathbf{Q} \boldsymbol{\Sigma}^{-1} \mathbf{P}^\top = \mathbf{P}' \mathrm{diag}\left(\frac{\sin \theta_1(\mathbf{U}, \mathbf{X})}{\cos \theta_1(\mathbf{U}, \mathbf{X})}, \dots, \frac{\sin \theta_k(\mathbf{U}, \mathbf{X})}{\cos \theta_k(\mathbf{U}, \mathbf{X})}\right) \mathbf{P}^\top.$$

As such, the singular values of $(\mathbf{V}^\top \mathbf{X})(\mathbf{U}^\top \mathbf{X})^{-1}$ are $\tan \theta_1(\mathbf{U}, \mathbf{X}), \dots, \tan \theta_k(\mathbf{U}, \mathbf{X})$, and its spectral norm is $\tan \theta_k(\mathbf{U}, \mathbf{X})$. □

## A.2. Perturbation Bounds for the QR Decomposition

We state here two useful perturbation bounds for the QR decomposition, which will be used for the analysis of our decentralized algorithm. The first one provides a bound on the perturbation of the Q-factor of a full column rank matrix under additive perturbations.

**Theorem A.2** (Chang (2012), Theorem 3.1)**.** *Let* $\mathbf{X} \in \mathbb{R}^{d \times p}$ *(with* $p \leqslant d$*) be of full column rank with QR factorization* $\mathbf{X} = \mathbf{Q}\mathbf{R}$*, and* $\Delta \mathbf{X} \in \mathbb{R}^{d \times p}$ *a perturbation. If*

$$\|\mathbf{X}^\dagger\|_2 \|\Delta \mathbf{X}\|_2 < 1,$$

*then* $\mathbf{X} + \Delta \mathbf{X}$ *has the unique QR factorization*

$$\mathbf{X} + \Delta \mathbf{X} = (\mathbf{Q} + \Delta \mathbf{Q})(\mathbf{R} + \Delta \mathbf{R}),$$

*and the following bound holds*

$$\|\Delta\mathbf{Q}\|_{\mathrm{F}} \leqslant \frac{\sqrt{2}\|\mathbf{X}^{\dagger}\|_2\|\Delta\mathbf{X}\|_{\mathrm{F}}}{1 - \|\mathbf{X}^{\dagger}\|_2\|\Delta\mathbf{X}\|_2}.$$

**Theorem A.3** (Sun (1991), Theorem 1.6). *Under the same hypotheses as Theorem A.2, the following bound holds:*

$$\|\Delta\mathbf{R}\|_{\mathrm{F}} \leqslant \frac{\sqrt{2}\|\mathbf{X}^{\dagger}\|_2\|\Delta\mathbf{X}\|_{\mathrm{F}}}{1 - \|\mathbf{X}^{\dagger}\|_2\|\Delta\mathbf{X}\|_2}\|\mathbf{R}\|_2.$$

### A.3. Weyl's Inequalities

We will often need to bound the difference between the singular values of two matrices. To do so, a useful result will be the following theorem, which is a consequence of Weyl's inequality.

**Theorem A.4** (Horn & Johnson (1985), Corollary 7.3.5). *Let $\mathbf{X}, \mathbf{Y} \in \mathbb{R}^{n \times m}$ and $q := \min(m, n)$. Let $\sigma_1(\mathbf{X}) \geqslant \cdots \geqslant \sigma_q(\mathbf{X}) \geqslant 0$ (resp. $\sigma_1(\mathbf{Y}) \geqslant \cdots \geqslant \sigma_q(\mathbf{Y}) \geqslant 0$) be the non-increasingly ordered singular values of $\mathbf{X}$ (resp. $\mathbf{Y}$). Then, for all $i \in \{1, \ldots, q\}$,*

$$|\sigma_i(\mathbf{X}) - \sigma_i(\mathbf{Y})| \leqslant \|\mathbf{X} - \mathbf{Y}\|_2.$$

Another useful consequence of Weyl's inequality is the following result on the impact of deleting a row of a thin matrix on its smallest singular value.

**Theorem A.5** (Horn & Johnson (1985), Corollary 7.3.6). *Let $\mathbf{X} \in \mathbb{R}^{d \times p}$ and with $d \geqslant p$. Let $\hat{\mathbf{X}}$ be a matrix obtained from $\mathbf{X}$ by deleting one of its rows. Then,*

$$\sigma_{\min}(\mathbf{X}) \geqslant \sigma_{\min}(\hat{\mathbf{X}}).$$

## B. Derivation of the Noise Conditions for (Xu, 2023) in Table 1

The noise conditions shown in our work and those of Hardt & Price (2014) and Balcan et al. (2016) are time-independent (except for the dependence of $\|\mathbf{U}_k^{\top}\mathbf{\Xi}_t\|_2$ in $\cos\theta_k(\mathbf{U}_k, \mathbf{X}_t)$). This is not the case for the conditions of Xu (2023), where the upper bounds decay geometrically as the iteration count increases. In order to compare our results with those of Xu (2023), we report an upper bound on the spectral norms of $\mathbf{U}_k^{\top}\mathbf{\Xi}_t$ and $\mathbf{U}_{-k}^{\top}\mathbf{\Xi}_t$ that must be verified at some iteration of ANPM for their result to hold. We first restate their main theorem below.

**Theorem B.1** (Xu (2023), Theorem 3.1). *Let $\varepsilon \in (0, 1)$, $k \in \{1, \ldots, d-1\}$, $\mathbf{A} \succeq \mathbf{0}$ with eigenvalues $\lambda_1 \geqslant \cdots \geqslant \lambda_k > \lambda_{k+1} \geqslant \cdots \geqslant \lambda_d \geqslant 0$ and $\mathbf{U}_k \in \mathrm{St}(d, k)$ its top-$k$ eigenvectors. Let $\mathbf{X}_0 \in \mathrm{St}(d, k)$ such that $\cos\theta_k(\mathbf{U}_k, \mathbf{X}_0) > 0$, and consider the ANPM iterates $\{\mathbf{X}_t\}_{t\geqslant 0}$ defined by (1) with momentum parameter $\beta > 0$ satisfying $\lambda_k > 2\sqrt{\beta} \geqslant \lambda_{k+1}$ and perturbations $\{\mathbf{\Xi}_t\}_{t\geqslant 0}$ satisfying, for all $t \in \{0, \ldots, T\}$,*

$$\|\mathbf{U}_{-k}^{\top}\mathbf{\Xi}_t\|_2 = \mathcal{O}\left(\frac{1}{T(T-t+1)}\left(\frac{\sqrt{\beta}}{\lambda_1^+}\right)^t \sqrt{\beta}\sin\theta_k(\mathbf{U}_k, \mathbf{X}_0)\right), \tag{6}$$

$$\|\mathbf{U}_k^{\top}\mathbf{\Xi}_t\|_2 = \mathcal{O}\left(\frac{1}{T(T-t+1)}\left(\frac{\lambda_k^+}{\lambda_1^+}\right)^t \lambda_k^+\cos\theta_k(\mathbf{U}_k, \mathbf{X}_0)\right), \tag{7}$$

*where the $\lambda_i^+$'s are defined as in (29):*

$$\lambda_1^+ := \frac{\lambda_1 + \sqrt{\lambda_1^2 - 4\beta}}{2}, \qquad \lambda_k^+ := \frac{\lambda_k + \sqrt{\lambda_k^2 - 4\beta}}{2}.$$

*Then, for all $t \geqslant T$, $\sin\theta_k(\mathbf{U}_k, \mathbf{X}_t) \leqslant \varepsilon$, where*

$$T = \Theta\left(\sqrt{\frac{\lambda_k}{\lambda_k - 2\sqrt{\beta}}}\log\left(\frac{\tan\theta_k(\mathbf{U}_k, \mathbf{X}_0)}{\varepsilon}\right)\right).$$

*Remark* B.2. In the original statement in (Xu, 2023), the condition on $\|\mathbf{U}_{-k}^\top \mathbf{\Xi}_t\|_2$ is actually a condition on $\|\mathbf{\Xi}_t\|_2$, which is more restrictive. However, their proof actually only requires a bound on $\|\mathbf{U}_{-k}^\top \mathbf{\Xi}_t\|_2$. We chose to state the theorem in this slightly improved form, in order to make the comparison with our results more direct. Similarly, Hardt & Price (2014) state their noise condition in terms of $\|\mathbf{\Xi}_t\|_2$, but their proof only requires a bound on $\|\mathbf{U}_{-k}^\top \mathbf{\Xi}_t\|_2$.

The next proposition shows that at a certain time step $t$, the noise conditions (6) and (7) imply the bounds given in Table 1.

**Proposition B.3.** *Consider the same setting as in Theorem B.1. Then, at a certain iteration $t \in \{0, \ldots, T\}$, the conditions* (6) *and* (7) *imply*

$$\|\mathbf{U}_{-k}^\top \mathbf{\Xi}_t\|_2 = \tilde{\mathcal{O}}\left((\lambda_k - 2\sqrt{\beta})\left(\frac{\varepsilon}{\tan\theta_k(\mathbf{U}_k, \mathbf{X}_0)}\right)^{\mu_\beta}\right),$$

$$\|\mathbf{U}_k^\top \mathbf{\Xi}_t\|_2 = \tilde{\mathcal{O}}\left((\lambda_k - 2\sqrt{\beta})\left(\frac{\varepsilon}{\tan\theta_k(\mathbf{U}_k, \mathbf{X}_0)}\right)^{\mu_k}\cos\theta_k(\mathbf{U}_k, \mathbf{X}_t)\right),$$

*where $\mu_\beta, \mu_k$ are constants verifying*

$$\mu_\beta = \Omega\left(\log\left(\frac{\lambda_1}{2\sqrt{\beta}}\right)\sqrt{\frac{\lambda_k}{\lambda_k - 2\sqrt{\beta}}}\right),$$

$$\mu_k = \Omega\left(\log\left(\frac{\lambda_1}{\lambda_k}\right)\sqrt{\frac{\lambda_k}{\lambda_k - 2\sqrt{\beta}}}\right).$$

*Proof.* At $t = \lfloor T/2 \rfloor$, the conditions (6) and (7) become

$$\|\mathbf{U}_{-k}^\top \mathbf{\Xi}_t\|_2 = \mathcal{O}\left(\frac{1}{T^2}\left(\frac{\sqrt{\beta}}{\lambda_1^+}\right)^{\lfloor T/2\rfloor}\sqrt{\beta}\sin\theta_k(\mathbf{U}_k, \mathbf{X}_0)\right),$$

$$\|\mathbf{U}_k^\top \mathbf{\Xi}_t\|_2 = \mathcal{O}\left(\frac{1}{T^2}\left(\frac{\lambda_k^+}{\lambda_1^+}\right)^{\lfloor T/2\rfloor}\lambda_k^+\cos\theta_k(\mathbf{U}_k, \mathbf{X}_0)\right).$$

From Theorem B.1, we have that

$$\frac{1}{T^2} = \tilde{\Theta}\left(\frac{\lambda_k - 2\sqrt{\beta}}{\lambda_k}\right).$$

We also have that $\sqrt{\beta} \leqslant \lambda_k^+ \leqslant \lambda_k$ and $\sin\theta_k(\mathbf{U}_k, \mathbf{X}_0) \leqslant 1$. Furthermore, from the proof of Theorem B.1 in (Xu, 2023), $\tan\theta_k(\mathbf{U}_k, \mathbf{X}_t)$ decays geometrically, so that $\cos\theta_k(\mathbf{U}_k, \mathbf{X}_0) = \mathcal{O}(\cos\theta_k(\mathbf{U}_k, \mathbf{X}_t))$. Using these inequalities, we obtain

$$\|\mathbf{U}_{-k}^\top \mathbf{\Xi}_t\|_2 = \tilde{\mathcal{O}}\left((\lambda_k - 2\sqrt{\beta})\left(\frac{\sqrt{\beta}}{\lambda_1^+}\right)^{\lfloor T/2\rfloor}\right),$$

$$\|\mathbf{U}_k^\top \mathbf{\Xi}_t\|_2 = \tilde{\mathcal{O}}\left((\lambda_k - 2\sqrt{\beta})\left(\frac{\lambda_k^+}{\lambda_1^+}\right)^{\lfloor T/2\rfloor}\cos\theta_k(\mathbf{U}_k, \mathbf{X}_t)\right).$$

We then have

$$\left(\frac{\sqrt{\beta}}{\lambda_1^+}\right)^{\lfloor T/2\rfloor} = \exp\left(\Theta\left(\sqrt{\frac{\lambda_k}{\lambda_k - 2\sqrt{\beta}}}\log\left(\frac{\lambda_1^+}{\sqrt{\beta}}\right)\log\left(\frac{\varepsilon}{\tan\theta_k(\mathbf{U}_k, \mathbf{X}_0)}\right)\right)\right) = \left(\frac{\varepsilon}{\tan\theta_k(\mathbf{U}_k, \mathbf{X}_0)}\right)^{\mu_\beta},$$

$$\left(\frac{\lambda_k^+}{\lambda_1^+}\right)^{\lfloor T/2\rfloor} = \exp\left(\Theta\left(\sqrt{\frac{\lambda_k}{\lambda_k - 2\sqrt{\beta}}}\log\left(\frac{\lambda_1^+}{\lambda_k^+}\right)\log\left(\frac{\varepsilon}{\tan\theta_k(\mathbf{U}_k, \mathbf{X}_0)}\right)\right)\right) = \left(\frac{\varepsilon}{\tan\theta_k(\mathbf{U}_k, \mathbf{X}_0)}\right)^{\mu_k},$$

where $\mu_\beta$ and $\mu_k$ verify

$$\mu_\beta = \Omega\left(\log\left(\frac{\lambda_1}{2\sqrt{\beta}}\right)\sqrt{\frac{\lambda_k}{\lambda_k - 2\sqrt{\beta}}}\right), \quad \mu_k = \Omega\left(\log\left(\frac{\lambda_1}{\lambda_k}\right)\sqrt{\frac{\lambda_k}{\lambda_k - 2\sqrt{\beta}}}\right),$$

since $\lambda_1^+/\lambda_k^+ \geqslant \lambda_1/\lambda_k$ and $\lambda_1^+ \geqslant \lambda_1/2$. This concludes the proof. $\square$

# C. Proofs for Section 2

## C.1. Variational Property of Scaled Chebyshev Polynomials

Let $\beta > 0$. We prove here Proposition 2.1, which states that the polynomial $p_t$ defined recursively as

$$p_0(x) = 1, \quad p_1(x) = \frac{x}{2}, \quad p_{t+1}(x) = xp_t(x) - \beta p_{t-1}(x), \quad \forall t \geqslant 1,$$

minimizes the infinity norm on the interval $[-2\sqrt{\beta}, 2\sqrt{\beta}]$ among all degree-$t$ polynomials with leading coefficient equal to $1/2$. The result is restated here for convenience.

**Proposition C.1** (Proposition 2.1)**.** *For all $t \geqslant 1$, the polynomial $p_t$ defined above satisfies*

$$p_t(x) = \arg \min_{\substack{p \in \mathbb{R}[x] \\ \deg(p) = t \\ \mathrm{lc}(p) = 1/2}} \max_{x \in [-2\sqrt{\beta}, 2\sqrt{\beta}]} |p(x)|, \tag{8}$$

*where $\mathrm{lc}(p)$ denotes the leading coefficient of $p$.*

*Proof.* This proof relies on the oscillatory behavior of the scaled Chebyshev polynomials on $[-2\sqrt{\beta}, 2\sqrt{\beta}]$. For all $t \geqslant 1$, let $T_t(x) := \frac{p_t(2\sqrt{\beta}x)}{\sqrt{\beta}^t}$. $T_t$ is the $t$-th Chebyshev polynomial of the first kind, as it satisfies

$$T_0(x) = 1, \quad T_1(x) = x, \quad T_{t+1}(x) = 2xT_t(x) - T_{t-1}(x), \quad \forall t \geqslant 1.$$

Then, from Definition 1.1 of (Mason & Handscomb, 2002), we have that for all $t \geqslant 0$ and for all $\theta \in \mathbb{R}$,

$$T_t(\cos \theta) = \cos(t\theta).$$

We deduce from it that for all $x \in [-2\sqrt{\beta}, 2\sqrt{\beta}]$,

$$p_t(x) = \sqrt{\beta}^t \cos\left(t \arccos\left(\frac{x}{2\sqrt{\beta}}\right)\right).$$

As such,

$$\max_{x \in [-2\sqrt{\beta}, 2\sqrt{\beta}]} |p_t(x)| = \sqrt{\beta}^t.$$

Now, let $q$ be a degree-$t$ polynomial with leading coefficient equal to $1/2$, and such that $\max_{x \in [-2\sqrt{\beta}, 2\sqrt{\beta}]} |q(x)| < \sqrt{\beta}^t$. Since $q$ and $p_t$ have the same leading coefficient and are of degree $t$, the polynomial $r := p_t - q$ is of degree at most $t - 1$. However, since $|q(x)| < \sqrt{\beta}^t$ for all $x \in [-2\sqrt{\beta}, 2\sqrt{\beta}]$, we have that for all $k \in \{0, \ldots, t\}$,

$$r\left(2\sqrt{\beta}\cos\left(\frac{k\pi}{t}\right)\right) = p_t\left(\cos\left(\frac{k\pi}{t}\right)2\sqrt{\beta}\right) - q\left(\cos\left(\frac{k\pi}{t}\right)2\sqrt{\beta}\right)$$

$$= (-1)^k \sqrt{\beta}^t - q\left(\cos\left(\frac{k\pi}{t}\right)2\sqrt{\beta}\right)$$

$$\begin{cases} > 0, & \text{if } k \text{ is even,} \\ < 0, & \text{if } k \text{ is odd.} \end{cases}$$

Then, from the intermediate value theorem, $r$ has at least one root in each interval $\left(2\sqrt{\beta}\cos\left(\frac{(k+1)\pi}{t}\right), 2\sqrt{\beta}\cos\left(\frac{k\pi}{t}\right)\right)$ for all $k \in \{0, \ldots, t-1\}$. As such, $r$ has at least $t$ distinct roots, which is impossible since $\deg(r) \leqslant t - 1$. We deduce that no such polynomial $q$ exists, which concludes the proof.

$\square$

## C.2. Proof of Theorem 2.2

We recall the assumptions and the notations introduced in the main body. We consider a PSD matrix $\mathbf{A} \succeq \mathbf{0}$ of size $d \times d$ with eigenvalues $\lambda_1 \geqslant \cdots \geqslant \lambda_d \geqslant 0$ and corresponding eigenvectors $\mathbf{u}_1, \ldots, \mathbf{u}_d$. For $k \in \{1, \ldots, d-1\}$, we assume that $\lambda_k > \lambda_{k+1}$ and introduce the following matrices:

$$\mathbf{U}_k := [\boldsymbol{u}_1, \ldots, \boldsymbol{u}_k] \in \mathrm{St}(d, k), \qquad\qquad \mathbf{U}_{-k} := [\boldsymbol{u}_{k+1}, \ldots, \boldsymbol{u}_d] \in \mathrm{St}(d, d-k),$$

$$\mathbf{\Lambda}_k := \mathrm{diag}(\lambda_1, \ldots, \lambda_k) \in \mathbb{R}^{k \times k}, \qquad\qquad \mathbf{\Lambda}_{-k} := \mathrm{diag}(\lambda_{k+1}, \ldots, \lambda_d) \in \mathbb{R}^{(d-k) \times (d-k)}.$$

Given $\mathbf{X}_0 \in \mathrm{St}(d, k)$, such that $\cos\theta_k(\mathbf{U}_k, \mathbf{X}_0) > 0$, we consider the ANPM iterates $\{\mathbf{X}_t\}_{t \geqslant 0}$ defined by

$$\mathbf{Y}_1 := \frac{1}{2}\mathbf{A}\mathbf{X}_0 + \mathbf{\Xi}_0, \quad \mathbf{X}_1, \ \mathbf{R}_1 = \mathrm{QR}\left(\mathbf{Y}_1\right), \tag{9}$$

$$\forall t \geqslant 1, \quad \begin{cases} \mathbf{Y}_{t+1} = \mathbf{A}\mathbf{X}_t - \beta\mathbf{X}_{t-1}\mathbf{R}_t^{-1} + \mathbf{\Xi}_t, \\ \mathbf{X}_{t+1}, \ \mathbf{R}_{t+1} = \mathrm{QR}(\mathbf{Y}_{t+1}). \end{cases} \tag{10}$$

with momentum parameter $\beta > 0$ satisfying $\lambda_k > 2\sqrt{\beta} \geqslant \lambda_{k+1}$ and perturbations $\{\mathbf{\Xi}_t\}_{t \geqslant 0}$ satisfying for some $\varepsilon \in (0, 1)$, for all $t \geqslant 0$,

$$\|\mathbf{U}_{-k}^\top\mathbf{\Xi}_t\|_2 \leqslant c(\lambda_k - 2\sqrt{\beta})\varepsilon, \tag{11}$$

$$\|\mathbf{U}_k^\top\mathbf{\Xi}_t\|_2 \leqslant c(\lambda_k - 2\sqrt{\beta})\cos\theta_k(\mathbf{U}_k, \mathbf{X}_t), \tag{12}$$

where $c := 1/32$. For convenience, we restate here Theorem 2.2.

**Theorem C.2** (Theorem 2.2). *Let $\varepsilon \in (0, 1)$ and consider the ANPM iterates $\{\mathbf{X}_t\}_{t \geqslant 0}$ defined by (9)-(10) with momentum parameter $\beta > 0$ satisfying $\lambda_k > 2\sqrt{\beta} \geqslant \lambda_{k+1}$ and perturbations $\{\mathbf{\Xi}_t\}_{t \geqslant 0}$ satisfying the noise conditions (11) and (12). Then, for all $t \geqslant T$, $\sin\theta_k(\mathbf{U}_k, \mathbf{X}_t) \leqslant \varepsilon$, where*

$$T := \frac{1}{-\log\left(1 - \frac{1}{2}\sqrt{\frac{\lambda_k - 2\sqrt{\beta}}{\lambda_k}}\right)} \log\left(\frac{2h_0}{\varepsilon}\right) = \mathcal{O}\left(\sqrt{\frac{\lambda_k}{\lambda_k - 2\sqrt{\beta}}}\log\left(\frac{\tan\theta_k(\mathbf{U}_k, \mathbf{X}_0)}{\varepsilon}\right)\right).$$

As explained in Section 2, the proof relies on studying the evolution of $\tan\theta_k(\mathbf{U}_k, \mathbf{X}_t)$. More specifically, we will show that $\tan\theta_k(\mathbf{U}_k, \mathbf{X}_t)$ is upper bounded by a constant term of order $\varepsilon$, which is related to the component $\mathbf{U}_{-k}^\top\mathbf{\Xi}_t$ of the noise, plus a term that decreases geometrically in $t$. To do so, we study the evolution of the matrix $\mathbf{H}_t$ defined as

$$\mathbf{H}_t := (\mathbf{U}_{-k}^\top\mathbf{X}_t)(\mathbf{U}_k^\top\mathbf{X}_t)^{-1} \in \mathbb{R}^{(d-k) \times k}, \tag{13}$$

whose spectral norm is $h_t := \tan\theta_k(\mathbf{U}_k, \mathbf{X}_t)$. We also introduce the matrix sequence $\{\mathbf{G}_t\}$ defined by

$$\forall t \geqslant 0, \quad \mathbf{G}_{t+1} = (\mathbf{I}_k - \beta\mathbf{\Lambda}_k^{-1}\mathbf{G}_t\mathbf{\Lambda}_k^{-1} + \mathbf{E}_{t+1})^{-1}, \tag{14}$$

$$\mathbf{G}_0 = (\mathbf{I}_k/2 + \mathbf{E}_0)^{-1},$$

where for all $t \geqslant 0$, $\mathbf{E}_t$ is a scaled noise matrix defined by

$$\mathbf{E}_t := \mathbf{\Lambda}_k^{-1}(\mathbf{U}_k^\top\mathbf{\Xi}_t)(\mathbf{U}_k^\top\mathbf{X}_t)^{-1}. \tag{15}$$

In particular, because of (12), $\|\mathbf{E}_t\|_2 \leqslant c\Delta$, where $\Delta$ is the gap defined as

$$\Delta := \frac{\lambda_k - 2\sqrt{\beta}}{\lambda_k} \in (0, 1). \tag{16}$$

Before getting to the proof of Theorem 2.2, we draw attention to a point that was not addressed in Section 2, regarding the well-definedness of the quantities we will manipulate. We will first prove that the various sequences we introduced up until now are well-defined. More specifically, we need to show:

- that the ANPM iterates given by (10) are well defined, i.e. that $\mathbf{Y}_t$ is of full column rank for all $t \geqslant 1$, which justifies that $\mathbf{R}_t$ is invertible for all $t \geqslant 1$ and that the QR decomposition is unique,

- that the matrices $\mathbf{H}_t$ given by (13) and $\mathbf{E}_t$ given by (15) are well defined, i.e. that $\mathbf{U}_k^\top \mathbf{X}_t$ is invertible for all $t \geqslant 0$,

- that the sequence $\{\mathbf{G}_t\}$ given by (14) is well defined, i.e. that the matrices $\mathbf{I}_k/2 + \mathbf{E}_0$ and $\mathbf{I}_k - \beta \mathbf{\Lambda}_k^{-1} \mathbf{G}_t \mathbf{\Lambda}_k^{-1} + \mathbf{E}_t$ are invertible for all $t \geqslant 0$.

We show in the next proposition that all of these properties are verified under the noise condition (12) and the assumption that $\cos\theta_k(\mathbf{U}_k, \mathbf{X}_0) > 0$. To do so, we leverage a relationship between $\mathbf{Y}_{t+1}$, $\mathbf{G}_t$ and $\mathbf{U}_k^\top \mathbf{X}_t$ and a uniform upper bound on the spectral norm of $\mathbf{G}_t$. This lemma also provides an expression of $\mathbf{G}_t$ in terms of $\mathbf{X}_t$, $\mathbf{X}_{t-1}$ and $\mathbf{R}_t$, which will be useful later for the proof of Theorem 2.2.

**Proposition C.3.** *Assume that condition* (12) *holds for all* $t \geqslant 0$ *and that* $\cos\theta_k(\mathbf{U}_k, \mathbf{X}_0) > 0$. *Then, for all* $t \geqslant 0$, $\mathbf{U}_k^\top \mathbf{X}_t$ *is invertible (which implies that* $\mathbf{E}_t$ *is well defined),* $\mathbf{G}_t$ *is well defined, and* $\mathbf{U}_k^\top \mathbf{Y}_{t+1}$ *and* $\mathbf{Y}_{t+1}$ *are of rank* $k$. *Furthermore,*

$$\|\mathbf{G}_t\|_2 \leqslant \frac{1}{1/2 - c\Delta}, \tag{17}$$

$\mathbf{G}_t$ *has the following closed-form expression for all* $t \geqslant 0$:

$$\mathbf{G}_t = \left( \mathbf{I}_k - \beta \mathbf{\Lambda}_k^{-1} (\mathbf{U}_k^\top \mathbf{X}_{t-1})(\mathbf{U}_k^\top \mathbf{X}_t \mathbf{R}_t)^{-1} + \mathbf{E}_t \right)^{-1}, \tag{18}$$

*where* $\mathbf{X}_{-1} := \frac{1}{2\beta} \mathbf{A} \mathbf{X}_0$, *and the following relationship holds for all* $t \geqslant 0$:

$$\mathbf{U}_k^\top \mathbf{Y}_{t+1} = \mathbf{\Lambda}_k \mathbf{G}_t^{-1} (\mathbf{U}_k^\top \mathbf{X}_t). \tag{19}$$

*Proof.* We will prove the result by induction.

**Base case:** For $t = 0$, by assumption $\cos\theta_k(\mathbf{U}_k, \mathbf{X}_0) > 0$, so that $\mathbf{U}_k^\top \mathbf{X}_0$ is invertible. Then $\mathbf{E}_0$ is well defined, and $\|\mathbf{E}_0\|_2 \leqslant c\Delta$. Notice that

$$\frac{1}{2}\mathbf{I}_k + \mathbf{E}_0 = \mathbf{I}_k - \left( \frac{1}{2}\mathbf{I}_k - \mathbf{E}_0 \right),$$

and that $\|\mathbf{I}_k/2 - \mathbf{E}_0\|_2 \leqslant 1/2 + c\Delta \leqslant 1/2 + c < 1$. As such, $\mathbf{I}_k/2 + \mathbf{E}_0$ is non-singular and $\mathbf{G}_0$ is well defined. Furthermore, we have that

$$\|\mathbf{G}_0\|_2 \leqslant \frac{1}{1 - \|\mathbf{I}_k/2 - \mathbf{E}_0\|_2} \leqslant \frac{1}{1 - (1/2 + c\Delta)} = \frac{1}{1/2 - c\Delta}.$$

We show the closed-form expression of $\mathbf{G}_0$ by simply plugging in the definition of $\mathbf{X}_{-1}$ into the right-hand side of (18):

$$\mathbf{I}_k - \beta \mathbf{\Lambda}_k^{-1}(\mathbf{U}_k^\top \mathbf{X}_{-1})(\mathbf{U}_k^\top \mathbf{X}_0 \mathbf{R}_0)^{-1} + \mathbf{E}_0 = \mathbf{I}_k - \beta \mathbf{\Lambda}_k^{-1} \left( \frac{1}{2\beta} \mathbf{\Lambda}_k (\mathbf{U}_k^\top \mathbf{X}_0) \right) (\mathbf{U}_k^\top \mathbf{X}_0)^{-1} + \mathbf{E}_0$$

$$= \mathbf{I}_k - \frac{1}{2}\mathbf{I}_k + \mathbf{E}_0 = \frac{1}{2}\mathbf{I}_k + \mathbf{E}_0,$$

which gives

$$\mathbf{G}_0 = \left( \frac{1}{2}\mathbf{I}_k + \mathbf{E}_0 \right)^{-1} = \left( \mathbf{I}_k - \beta \mathbf{\Lambda}_k^{-1}(\mathbf{U}_k^\top \mathbf{X}_{-1})(\mathbf{U}_k^\top \mathbf{X}_0 \mathbf{R}_0)^{-1} + \mathbf{E}_0 \right)^{-1}.$$

We now need to show that $\mathbf{U}_k^\top \mathbf{Y}_1$ is of rank $k$, which immediately implies that $\mathbf{Y}_1$ is of rank $k$. To do so, we will prove (19): we have from (9) that

$$\mathbf{U}_k^\top \mathbf{Y}_1 = \frac{1}{2}\mathbf{U}_k^\top \mathbf{A} \mathbf{X}_0 + \mathbf{U}_k^\top \mathbf{\Xi}_0 = \frac{1}{2}\mathbf{\Lambda}_k(\mathbf{U}_k^\top \mathbf{X}_0) + \mathbf{U}_k^\top \mathbf{\Xi}_0 = \mathbf{\Lambda}_k \left( \frac{\mathbf{I}_k}{2} + \mathbf{E}_0 \right)(\mathbf{U}_k^\top \mathbf{X}_0) = \mathbf{\Lambda}_k \mathbf{G}_0^{-1}(\mathbf{U}_k^\top \mathbf{X}_0).$$

Since $\mathbf{\Lambda}_k$, $\mathbf{G}_0^{-1}$ and $(\mathbf{U}_k^\top \mathbf{X}_0)$ are all non-singular, $\mathbf{U}_k^\top \mathbf{Y}_1$ is non-singular, and thus $\mathbf{Y}_1$ is of rank $k$. From this, we conclude that $\mathbf{X}_1, \mathbf{R}_1$ are well defined. This concludes the base case.

**Induction:** Now let $t \geqslant 0$ and assume that Proposition C.3 is true for all steps in $\{0, \ldots, t\}$. Then, we have that

$$\mathbf{U}_k^\top \mathbf{X}_{t+1} = \mathbf{U}_k^\top \mathbf{Y}_{t+1} \mathbf{R}_{t+1}^{-1}.$$

The right hand side is well-defined and invertible because of the induction hypothesis which is verified for step $t$. Thus $\mathbf{U}_k^\top \mathbf{X}_{t+1}$ is invertible and $\mathbf{E}_{t+1}$ is well defined. We now prove that $\mathbf{G}_{t+1}$ is well defined. By the induction hypothesis, we have that

$$\|\beta \mathbf{\Lambda}_k^{-1} \mathbf{G}_t \mathbf{\Lambda}_k^{-1} - \mathbf{E}_{t+1}\|_2 \leqslant \beta \|\mathbf{\Lambda}_k^{-1}\|_2^2 \|\mathbf{G}_t\|_2 + \|\mathbf{E}_{t+1}\|_2 \leqslant \frac{\beta}{\lambda_k^2} \cdot \frac{1}{1/2 - c\Delta} + c\Delta$$

$$\leqslant \frac{1}{2 - 4c\Delta} + c\Delta \leqslant \frac{1}{2 - 4c} + c < 1,$$

where we used the fact that $\beta \leqslant \lambda_k^2/4$ and $\Delta \leqslant 1$. As such, $\mathbf{I}_k - \beta \mathbf{\Lambda}_k^{-1} \mathbf{G}_t \mathbf{\Lambda}_k^{-1} + \mathbf{E}_{t+1}$ is non-singular and $\mathbf{G}_{t+1}$ is well defined. Furthermore, we have that

$$\|\mathbf{G}_{t+1}\|_2 \leqslant \frac{1}{1 - \|\beta \mathbf{\Lambda}_k^{-1} \mathbf{G}_t \mathbf{\Lambda}_k^{-1} - \mathbf{E}_{t+1}\|_2} \leqslant \frac{1}{1 - \frac{\beta}{\lambda_k^2} \|\mathbf{G}_t\|_2 - c\Delta} = \frac{1}{1 - \frac{1}{4} \frac{(1-\Delta)^2}{1/2 - c\Delta} - c\Delta} \overset{\Delta \in [0,1]}{\leqslant} \frac{1}{1/2 - c\Delta}.$$

We now prove the closed-form expression of $\mathbf{G}_{t+1}$. By the induction hypothesis, we have that

$$\begin{aligned}
\mathbf{G}_{t+1} &= (\mathbf{I}_k - \beta \mathbf{\Lambda}_k^{-1} \mathbf{G}_t \mathbf{\Lambda}_k^{-1} + \mathbf{E}_{t+1})^{-1} \\
&= \left(\mathbf{I}_k - \beta \mathbf{\Lambda}_k^{-1} \left(\mathbf{I}_k - \beta \mathbf{\Lambda}_k^{-1} (\mathbf{U}_k^\top \mathbf{X}_{t-1})(\mathbf{R}_t)^{-1}(\mathbf{U}_k^\top \mathbf{X}_t)^{-1} + \mathbf{E}_t\right)^{-1} \mathbf{\Lambda}_k^{-1} + \mathbf{E}_{t+1}\right)^{-1} \\
&= \left(\mathbf{I}_k - \beta \mathbf{\Lambda}_k^{-1} (\mathbf{U}_k^\top \mathbf{X}_t) \left(\mathbf{\Lambda}_k \mathbf{U}_k^\top \mathbf{X}_t - \beta (\mathbf{U}_k^\top \mathbf{X}_{t-1})(\mathbf{R}_t)^{-1} + \mathbf{U}_k^\top \mathbf{\Xi}_t\right)^{-1} + \mathbf{E}_{t+1}\right)^{-1} \\
&= \left(\mathbf{I}_k - \beta \mathbf{\Lambda}_k^{-1} (\mathbf{U}_k^\top \mathbf{X}_t) \left(\mathbf{U}_k^\top \left(\mathbf{A}\mathbf{X}_t - \beta \mathbf{X}_{t-1}(\mathbf{R}_t)^{-1} + \mathbf{\Xi}_t\right)\right)^{-1} + \mathbf{E}_{t+1}\right)^{-1} \\
&= \left(\mathbf{I}_k - \beta \mathbf{\Lambda}_k^{-1} (\mathbf{U}_k^\top \mathbf{X}_t)(\mathbf{U}_k^\top \mathbf{X}_{t+1} \mathbf{R}_{t+1})^{-1} + \mathbf{E}_t\right)^{-1},
\end{aligned}$$

where the last equality is because $\mathbf{X}_{t+1} \mathbf{R}_{t+1} = \mathbf{A}\mathbf{X}_t - \beta \mathbf{X}_{t-1} \mathbf{R}_t^{-1} + \mathbf{\Xi}_t$ since $\mathbf{X}_{t+1}, \mathbf{R}_{t+1} = \mathrm{QR}(\mathbf{A}\mathbf{X}^{(t)} - \beta \mathbf{X}_{t-1} \mathbf{R}_t^{-1} + \mathbf{\Xi}_t)$. Note that this is also verified for $t = 0$ using the definition of $\mathbf{X}_{-1}$.

Finally, we show that $\mathbf{U}_k^\top \mathbf{Y}_{t+2}$ is of rank $k$. We have from (10) that

$$\begin{aligned}
\mathbf{U}_k^\top \mathbf{Y}_{t+2} &= \mathbf{U}_k^\top \mathbf{A}\mathbf{X}_{t+1} - \beta \mathbf{U}_k^\top \mathbf{X}_t \mathbf{R}_{t+1}^{-1} + \mathbf{U}_k^\top \mathbf{\Xi}_{t+1} \\
&= \mathbf{\Lambda}_k \left(\mathbf{I}_k - \beta \mathbf{\Lambda}_k^{-1} (\mathbf{U}_k^\top \mathbf{X}_t)(\mathbf{U}_k^\top \mathbf{X}_{t+1} \mathbf{R}_{t+1})^{-1} + \mathbf{E}_{t+1}\right)(\mathbf{U}_k^\top \mathbf{X}_{t+1}) \\
&= \mathbf{\Lambda}_k \mathbf{G}_{t+1}^{-1}(\mathbf{U}_k^\top \mathbf{X}_{t+1}), \qquad (20)
\end{aligned}$$

where the last equality is due to (18). Both $\mathbf{G}_{t+1}$ and $\mathbf{U}_k^\top \mathbf{X}_{t+1}$ are of rank $k$, which proves that $\mathbf{U}_k^\top \mathbf{Y}_{t+2}$ is of rank $k$, and thus that $\mathbf{Y}_{t+2}$ is of rank $k$. This concludes the induction and the proof. $\qquad\square$

---

We now have the necessary tools to start the proof of Theorem 2.2. The proof is structured around multiple technical lemmas. The roadmap is the following: in Lemma C.4, we derive a three-term recurrence relation on the matrices $\mathbf{H}_t$, in which the matrix sequence $\{\mathbf{G}_t\}$ appears. This allows us to rewrite $\mathbf{H}_t$ using scaled Chebyshev polynomials in Lemma C.5. Then, by upper bounding the spectral norm of the different factors in this expression, which is done in Lemma C.8, we show in Lemma C.9 that $h_t$ is upper bounded by a geometrically decaying term, a constant term of order $\varepsilon$, and a linear combination of the previous $h_i$'s. We derive from this the geometric decay of $h_t$ up to a constant term of order $\varepsilon$ in Lemma C.10, which allows us to conclude the proof of Theorem 2.2.

We first prove the three-term recurrence relation on $\mathbf{H}_t$.

**Lemma C.4.** *For all $t \geqslant 1$,*

$$\mathbf{H}_{t+1}\mathbf{C}_{t+1} = \mathbf{\Lambda}_{-k}\mathbf{H}_t\mathbf{C}_t - \beta\mathbf{H}_{t-1}\mathbf{C}_{t-1} + \mathbf{\Psi}_t\mathbf{C}_t, \tag{21}$$

*where for all $t \geqslant 0$,*

$$\mathbf{C}_t := \prod_{s=0}^{t-1} \mathbf{\Lambda}_k \mathbf{G}_{t-1-s}^{-1}, \tag{22}$$

$$\mathbf{\Psi}_t := (\mathbf{U}_{-k}^{\top}\mathbf{\Xi}_t)(\mathbf{U}_k^{\top}\mathbf{X}_t)^{-1}. \tag{23}$$

*Furthermore, we have that*

$$\mathbf{H}_1\mathbf{C}_1 = \frac{1}{2}\mathbf{\Lambda}_{-k}\mathbf{H}_0\mathbf{C}_0 + \mathbf{\Psi}_0\mathbf{C}_0. \tag{24}$$

*Proof.* Let $t \geqslant 1$. From the definition of $\mathbf{H}_{t+1}$ in (13) and the ANPM update in (10), we have that

$$
\begin{aligned}
\mathbf{H}_{t+1} &= \left(\mathbf{U}_{-k}^{\top}\mathbf{X}_{t+1}\mathbf{R}_{t+1}\right)\left(\mathbf{U}_k^{\top}\mathbf{X}_{t+1}\mathbf{R}_{t+1}\right)^{-1} \\
&= \left(\mathbf{U}_{-k}^{\top}\left(\mathbf{A}\mathbf{X}_t - \beta\mathbf{X}_{t-1}(\mathbf{R}_t)^{-1} + \mathbf{\Xi}_t\right)\right)\left(\mathbf{U}_k^{\top}\left(\mathbf{A}\mathbf{X}_t - \beta\mathbf{X}_{t-1}(\mathbf{R}_t)^{-1} + \mathbf{\Xi}_t\right)\right)^{-1} \\
&= \left(\mathbf{\Lambda}_{-k}\mathbf{U}_{-k}^{\top}\mathbf{X}_t - \beta\mathbf{U}_{-k}^{\top}\mathbf{X}_{t-1}(\mathbf{R}_t)^{-1} + \mathbf{U}_{-k}^{\top}\mathbf{\Xi}_t\right)\left(\mathbf{\Lambda}_k\mathbf{U}_k^{\top}\mathbf{X}_t - \beta\mathbf{U}_k^{\top}\mathbf{X}_{t-1}(\mathbf{R}_t)^{-1} + \mathbf{U}_k^{\top}\mathbf{\Xi}_t\right)^{-1} \\
&= \left(\mathbf{\Lambda}_{-k}\mathbf{U}_{-k}^{\top}\mathbf{X}_t - \beta\mathbf{U}_{-k}^{\top}\mathbf{X}_{t-1}(\mathbf{R}_t)^{-1} + \mathbf{U}_{-k}^{\top}\mathbf{\Xi}_t\right)\left(\mathbf{U}_k^{\top}\mathbf{X}_t\right)^{-1}\mathbf{G}_t\mathbf{\Lambda}_k^{-1} \\
&= \left(\mathbf{\Lambda}_{-k}\mathbf{U}_{-k}^{\top}\mathbf{X}_t - \beta\mathbf{U}_{-k}^{\top}\mathbf{X}_{t-1}(\mathbf{R}_t)^{-1} + \mathbf{U}_{-k}^{\top}\mathbf{\Xi}_t\right)\left(\mathbf{U}_k^{\top}\mathbf{X}_t\right)^{-1}\mathbf{G}_t\mathbf{\Lambda}_k^{-1} \\
&= \mathbf{\Lambda}_{-k}\mathbf{H}_t\mathbf{G}_t\mathbf{\Lambda}_k^{-1} - \beta(\mathbf{U}_{-k}^{\top}\mathbf{X}_{t-1})(\mathbf{U}_k^{\top}\mathbf{X}_t\mathbf{R}_t)^{-1}\mathbf{G}_t\mathbf{\Lambda}_k^{-1} + \mathbf{\Psi}_t\mathbf{G}_t\mathbf{\Lambda}_k^{-1}. 
\end{aligned} \tag{25}
$$

Then, from the expression of $\mathbf{U}_k^{\top}\mathbf{Y}_t$ in (19), we have that for all $t \geqslant 1$,

$$\mathbf{U}_k^{\top}\mathbf{X}_t\mathbf{R}_t = \mathbf{U}_k^{\top}\mathbf{Y}_t = \mathbf{\Lambda}_k\mathbf{G}_{t-1}^{-1}(\mathbf{U}_k^{\top}\mathbf{X}_{t-1}),$$

so that

$$(\mathbf{U}_k^{\top}\mathbf{X}_t\mathbf{R}_t)^{-1} = (\mathbf{U}_k^{\top}\mathbf{X}_{t-1})^{-1}\mathbf{G}_{t-1}\mathbf{\Lambda}_k^{-1}.$$

Plugging this into the expression of $\mathbf{H}_{t+1}$ in (25), we obtain

$$\mathbf{H}_{t+1} = \mathbf{\Lambda}_{-k}\mathbf{H}_t\mathbf{G}_t\mathbf{\Lambda}_k^{-1} - \beta\mathbf{H}_{t-1}\mathbf{G}_{t-1}\mathbf{\Lambda}_k^{-1}\mathbf{G}_t\mathbf{\Lambda}_k^{-1} + \mathbf{\Psi}_t\mathbf{G}_t\mathbf{\Lambda}_k^{-1}.$$

Multiplying both sides by $\mathbf{C}_{t+1} = \mathbf{\Lambda}_k\mathbf{G}_t^{-1}\mathbf{C}_t = \mathbf{\Lambda}_k\mathbf{G}_t^{-1}\mathbf{\Lambda}_k^{-1}\mathbf{G}_{t-1}^{-1}\mathbf{C}_{t-1}$ on the right, we obtain the desired recurrence:

$$\mathbf{H}_{t+1}\mathbf{C}_{t+1} = \mathbf{\Lambda}_{-k}\mathbf{H}_t\mathbf{C}_t - \beta\mathbf{H}_{t-1}\mathbf{C}_{t-1} + \mathbf{\Psi}_t\mathbf{C}_t.$$

We now prove the equality $\mathbf{H}_1\mathbf{C}_1 = \frac{1}{2}\mathbf{\Lambda}_{-k}\mathbf{H}_0\mathbf{C}_0 + \mathbf{\Psi}_0\mathbf{C}_0$. From the initialization (9), we have that

$$
\begin{aligned}
\mathbf{H}_1 &= \left(\mathbf{U}_{-k}^{\top}\mathbf{Y}_1\right)\left(\mathbf{U}_k^{\top}\mathbf{Y}_1\right)^{-1} \\
&= \left(\frac{1}{2}\mathbf{\Lambda}_{-k}\mathbf{U}_{-k}^{\top}\mathbf{X}_0 + \mathbf{U}_{-k}^{\top}\mathbf{\Xi}_0\right)\left(\frac{1}{2}\mathbf{\Lambda}_k\mathbf{U}_k^{\top}\mathbf{X}_0 + \mathbf{U}_k^{\top}\mathbf{\Xi}_0\right)^{-1} \\
&= \left(\frac{1}{2}\mathbf{\Lambda}_{-k}\mathbf{U}_{-k}^{\top}\mathbf{X}_0 + \mathbf{U}_{-k}^{\top}\mathbf{\Xi}_0\right)\left(\mathbf{U}_k^{\top}\mathbf{X}_0\right)^{-1}\mathbf{G}_0\mathbf{\Lambda}_k^{-1} \\
&= \frac{1}{2}\mathbf{\Lambda}_{-k}\mathbf{H}_0\mathbf{G}_0\mathbf{\Lambda}_k^{-1} + \mathbf{\Psi}_0\mathbf{G}_0\mathbf{\Lambda}_k^{-1},
\end{aligned}
$$

which gives the desired result after multiplying both sides by $\mathbf{C}_1 = \mathbf{\Lambda}_k\mathbf{G}_0^{-1}\mathbf{C}_0$ on the right. $\qquad\square$

Define the following sequences of scaled Chebyshev polynomials $\{p_t\}$ and $\{q_t\}$ as

$$p_0(x) = 1, \quad p_1(x) = \frac{x}{2}, \quad p_{t+1}(x) = xp_t(x) - \beta p_{t-1}(x), \tag{26}$$

$$q_0(x) = 1, \quad q_1(x) = x, \quad q_{t+1}(x) = xq_t(x) - \beta q_{t-1}(x). \tag{27}$$

We show using the previously obtained three-term recurrence relation that $\mathbf{H}_t$ can be simply written in terms of the polynomials $\{p_t\}$ and $\{q_t\}$.

**Lemma C.5.** *For all $t \geqslant 0$,*

$$\mathbf{H}_t = p_t(\mathbf{\Lambda}_{-k})\mathbf{H}_0\mathbf{C}_t^{-1} + \sum_{s=0}^{t-1} q_s(\mathbf{\Lambda}_{-k})\mathbf{\Psi}_{t-1-s}\mathbf{C}_{t-1-s}\mathbf{C}_t^{-1}. \tag{28}$$

*Proof.* The proof is by induction. The case $t = 0$ is immediate as $p_0(x) = 1$ and $\mathbf{C}_0 = \mathbf{I}_k$. For $t = 1$, we have from (24) that $\mathbf{H}_1\mathbf{C}_1 = p_1(\mathbf{\Lambda}_{-k})\mathbf{H}_0\mathbf{C}_0 + q_0(\mathbf{\Lambda}_{-k})\mathbf{\Psi}_0\mathbf{C}_0$, which gives the desired result after multiplying both sides by $\mathbf{C}_1^{-1}$ on the right. Now, let $t \geqslant 1$ and assume that the result is true for $t - 1$ and $t$. From (21), we have that

$$\mathbf{H}_{t+1}\mathbf{C}_{t+1} = \mathbf{\Lambda}_{-k}\mathbf{H}_t\mathbf{C}_t - \beta\mathbf{H}_{t-1}\mathbf{C}_{t-1} + \mathbf{\Psi}_t\mathbf{C}_t.$$

Plugging in the induction hypothesis, we obtain, if $t \geqslant 2$,

$$\mathbf{H}_{t+1}\mathbf{C}_{t+1} = \mathbf{\Lambda}_{-k}\left(p_t(\mathbf{\Lambda}_{-k})\mathbf{H}_0\mathbf{C}_t^{-1} + \sum_{s=0}^{t-1} q_s(\mathbf{\Lambda}_{-k})\mathbf{\Psi}_{t-1-s}\mathbf{C}_{t-1-s}\mathbf{C}_t^{-1}\right)\mathbf{C}_t$$

$$-\beta\left(p_{t-1}(\mathbf{\Lambda}_{-k})\mathbf{H}_0\mathbf{C}_{t-1}^{-1} + \sum_{s=0}^{t-2} q_s(\mathbf{\Lambda}_{-k})\mathbf{\Psi}_{t-2-s}\mathbf{C}_{t-2-s}\mathbf{C}_{t-1}^{-1}\right)\mathbf{C}_{t-1} + \mathbf{\Psi}_t\mathbf{C}_t$$

$$= \left(\mathbf{\Lambda}_{-k}p_t(\mathbf{\Lambda}_{-k}) - \beta p_{t-1}(\mathbf{\Lambda}_{-k})\right)\mathbf{H}_0 + \sum_{s=0}^{t-1}\left(\mathbf{\Lambda}_{-k}q_s(\mathbf{\Lambda}_{-k}) - \beta q_{s-1}(\mathbf{\Lambda}_{-k})\right)\mathbf{\Psi}_{t-1-s}\mathbf{C}_{t-1-s} + \mathbf{\Psi}_t\mathbf{C}_t,$$

$$= p_{t+1}(\mathbf{\Lambda}_{-k})\mathbf{H}_0 + \sum_{s=0}^{t} q_s(\mathbf{\Lambda}_{-k})\mathbf{\Psi}_{t-s}\mathbf{C}_{t-s}.$$

If $t = 1$, we have

$$\mathbf{H}_2\mathbf{C}_2 = \mathbf{\Lambda}_{-k}\left(p_1(\mathbf{\Lambda}_{-k})\mathbf{H}_0\mathbf{C}_1^{-1} + q_0(\mathbf{\Lambda}_{-k})\mathbf{\Psi}_0\mathbf{C}_0\mathbf{C}_1^{-1}\right)\mathbf{C}_1 - \beta p_0(\mathbf{\Lambda}_{-k})\mathbf{H}_0\mathbf{C}_0^{-1} + \mathbf{\Psi}_1\mathbf{C}_1$$

$$= \left(\mathbf{\Lambda}_{-k}p_1(\mathbf{\Lambda}_{-k}) - \beta p_0(\mathbf{\Lambda}_{-k})\right)\mathbf{H}_0 + \mathbf{\Lambda}_{-k}q_0(\mathbf{\Lambda}_{-k})\mathbf{\Psi}_0\mathbf{C}_0 + \mathbf{\Psi}_1\mathbf{C}_1,$$

$$= p_2(\mathbf{\Lambda}_{-k})\mathbf{H}_0 + \sum_{s=0}^{1} q_s(\mathbf{\Lambda}_{-k})\mathbf{\Psi}_{1-s}\mathbf{C}_{1-s}.$$

In both cases, we obtain the desired result after multiplying both sides by $\mathbf{C}_{t+1}^{-1}$ on the right. This concludes the induction and the proof. □

The next step of the proof consists in upper bounding the different factors appearing in the expression of $\mathbf{H}_t$ in (28). Before proving these bounds in Lemma C.8, we need two intermediate results. The first one regards the sequences of polynomials $\{p_t\}$ and $\{q_t\}$. More specifically, we give closed-form expressions of these polynomials, which will allow us to upper bound their values on the interval $[0, 2\sqrt{\beta}]$.

**Lemma C.6** (Xu (2023), Lemma 3.4). *For all $t \geqslant 0$ and $x \in \mathbb{R}$, we have*

$$p_t(x) = \frac{1}{2}\left((x^+)^t + (x^-)^t\right),$$

$$q_t(x) = \sum_{s=0}^{t}(x^+)^s(x^-)^{t-s},$$

*where for all $x \in \mathbb{R}$, $x^{\pm}$ are the roots of the polynomial $z^2 - xz + \beta$:*

$$x^{\pm} := \begin{cases} \frac{x \pm \sqrt{x^2 - 4\beta}}{2} & \text{if} \quad x^2 \geqslant 4\beta, \\ \frac{x \pm i\sqrt{4\beta - x^2}}{2} & \text{if} \quad x^2 < 4\beta, \end{cases} \tag{29}$$

*where $i$ is the imaginary unit.*

*Proof.* Our proof follows the same principle as the proof of Lemma 20 in (Xu et al., 2018). Let $\{\pi_t\} \in \{\{p_t\}, \{q_t\}\}$. Then, for all $t \geqslant 0$, we have $\pi_{t+2}(x) = x\pi_{t+1}(x) - \beta\pi_t(x)$. Let $x \in \mathbb{R}$ and denote $\Pi(z) := \sum_{t=0}^{\infty} \pi_t(x)z^t$ the generating function of the sequence $\{\pi_t(x)\}$. Then, we have

$$\Pi(z) - \pi_0(x) - \pi_1(x)z = xz\left(\Pi(z) - \pi_0(x)\right) - \beta z^2 \Pi(z),$$

$$\Pi(z) = \frac{\pi_0(x) + z(\pi_1(x) - x\pi_0(x))}{1 - xz + \beta z^2},$$

$$\Pi(z) = \frac{C_+(x)}{1 - \beta z/x^+} + \frac{C_-(x)}{1 - \beta z/x^-},$$

where $C_{\pm}(x)$ are constants that depend on $x$ and the initial conditions of the sequence $\{\pi_t\}$. These equalities are valid for all $z$ such that $|z| < \min\{|x^+/\beta|, |x^-/\beta|\}$, which is non-zero. Then, multiplying both sides by $(1 - \beta z/x^+)$ (resp. $(1 - \beta z/x^-)$) and taking the limit $z \to x^+/\beta$ (resp. $z \to x^-/\beta$) gives

$$C_+(x) = \frac{\pi_0(x) + \frac{x^+}{\beta}(\pi_1(x) - x\pi_0(x))}{1 - x^+/x^-},$$

$$C_-(x) = \frac{\pi_0(x) + \frac{x^-}{\beta}(\pi_1(x) - x\pi_0(x))}{1 - x^-/x^+}.$$

Furthermore, we have that

$$\Pi(z) = \sum_{t=0}^{+\infty} \left(C_+(x)\left(\frac{\beta}{x^+}\right)^t + C_-(x)\left(\frac{\beta}{x^-}\right)^t\right)z^t.$$

Identifying the coefficients then gives for all $t \geqslant 0$,

$$\pi_t(x) = C_+(x)\left(\frac{\beta}{x^+}\right)^t + C_-(x)\left(\frac{\beta}{x^-}\right)^t,$$
$$= C_+(x)(x^-)^t + C_-(x)(x^+)^t,$$

Where the last equality is because $x^+ x^- = \beta$. Plugging in the initial conditions for $\{p_t\}$ and $\{q_t\}$ gives the desired closed-form expressions. For $\{p_t\}$, we have $p_0(x) = 1, p_1(x) = x/2$, which gives

$$C_+(x) = \frac{1 - \frac{x^+}{\beta}\frac{x}{2}}{1 - x^+/x^-} = \frac{1}{2}, \qquad C_-(x) = \frac{1 - \frac{x^-}{\beta}\frac{x}{2}}{1 - x^-/x^+} = \frac{1}{2},$$

which gives for all $t \geqslant 0$,

$$p_t(x) = \frac{1}{2}(x^+)^t + \frac{1}{2}(x^-)^t.$$

For $\{q_t\}$, we have $q_0(x) = 1, q_1(x) = x$, which gives

$$C_+(x) = \frac{x^-}{x^- - x^+}, \qquad C_-(x) = \frac{x^+}{x^+ - x^-},$$

which gives for all $t \geqslant 0$,

$$q_t(x) = \frac{(x^+)^{t+1} - (x^-)^{t+1}}{x^+ - x^-} = \sum_{s=0}^{t}(x^+)^s(x^-)^{t-s}.$$

$\square$

The second intermediate result we need is an upper bound on the spectral norm of $\mathbf{G}_t$ that is tighter than the one shown in Proposition C.3.

**Lemma C.7.** *For all $t \geqslant 0$,*

$$\|\mathbf{G}_t\|_2 \leqslant \frac{r_+^t + \kappa r_-^t}{r_+^{t+1} + \kappa r_-^{t+1}},$$

*where*

$$r_\pm := \frac{(1 - c\Delta) \pm \sqrt{(1 - c\Delta)^2 - \frac{4\beta}{\lambda_k^2}}}{2},$$

$$\kappa := 1 + 2c\frac{\Delta}{\sqrt{(1 - c\Delta)^2 - 4\beta/\lambda_k^2} - c\Delta}.$$

*Furthermore, we have $\kappa \leqslant 16/15$.*

*Proof.* Notice first that $(1 - c\Delta)^2 - 4\beta/\lambda_k^2 > 0$, so that $r_\pm$ are well-defined. Consider the sequence $\{m_t\}$ defined by the following Riccati difference equation:

$$m_{t+1} = \frac{1}{(1 - c\Delta) - \frac{\beta}{\lambda_k^2} m_t},$$

$$m_0 = \frac{1}{1/2 - c\Delta}.$$

Since $\beta/\lambda_k^2 = (1 - \Delta)^2$, we have that for all $t \geqslant 0$, $0 < m_t \leqslant (1/2 - c\Delta)^{-1}$ (this can be shown by a simple induction).

We will now prove by induction that for all $t \geqslant 0$, $\|\mathbf{G}_t\|_2 \leqslant m_t$. The base case $t = 0$ is true from Proposition C.3. Now, let $t \geqslant 0$ and assume that $\|\mathbf{G}_t\|_2 \leqslant m_t$. Then, we have

$$\|\mathbf{G}_{t+1}\|_2 = \|(\mathbf{I}_k - \beta \mathbf{\Lambda}_k^{-1} \mathbf{G}_t \mathbf{\Lambda}_k^{-1} + \mathbf{E}_{t+1})^{-1}\|_2 \leqslant \frac{1}{1 - \frac{\beta}{\lambda_k^2} \|\mathbf{G}_t\|_2 - c\Delta} \leqslant \frac{1}{(1 - c\Delta) - \frac{\beta}{\lambda_k^2} m_t} = m_{t+1},$$

where the second inequality is valid due to the fact that $m_t \leqslant (1/2 - c\Delta)^{-1} < (1 - c\Delta)\lambda_k^2/\beta$. This concludes the induction.

We will now derive a closed-form expression of $m_t$. Let $\{y_t\}$ be the sequence defined as $y_t := \prod_{s=0}^{t-1} m_s^{-1}$ (with the convention $y_0 := 1$). Then, we have that for all $t \geqslant 1$,

$$y_{t+1} = \frac{y_t}{m_t} = y_t \left((1 - c\Delta) - \frac{\beta}{\lambda_k^2} m_{t-1}\right) = y_t \left((1 - c\Delta) - \frac{\beta}{\lambda_k^2} \frac{y_{t-1}}{y_t}\right),$$

$$y_{t+1} = (1 - c\Delta)y_t - \frac{\beta}{\lambda_k^2} y_{t-1}.$$

With initial conditions $y_0 = 1$ and $y_1 = 1/2 - c\Delta$, we can solve this linear difference equation to obtain for all $t \geqslant 0$,

$$y_t = \kappa_+ r_+^t + \kappa_- r_-^t,$$

where $r_\pm$ are defined in the statement of the lemma, and where the constants $\kappa_\pm$ are defined as

$$\kappa_\pm := \frac{1}{2}\left(1 \mp \frac{c\Delta}{\sqrt{(1 - c\Delta)^2 - 4\beta/\lambda_k^2}}\right).$$

Then, since $m_t = y_t/y_{t+1}$ and $\|\mathbf{G}_t\|_2 \leqslant m_t$ for all $t \geqslant 0$, we have that

$$\|\mathbf{G}_t\|_2 \leqslant \frac{\kappa_+ r_+^t + \kappa_- r_-^t}{\kappa_+ r_+^{t+1} + \kappa_- r_-^{t+1}} = \frac{r_+^t + \kappa r_-^t}{r_+^{t+1} + \kappa r_-^{t+1}},$$

where

$$\kappa := \frac{\kappa_-}{\kappa_+} = 1 + 2\frac{c\Delta}{\sqrt{(1 - c\Delta)^2 - 4\beta/\lambda_k^2} - c\Delta}.$$

Finally, we prove the upper bound on $\kappa$ by noticing that because of the concavity of the function $u \mapsto \sqrt{u^2 - 4\beta/\lambda_k^2}$ over $[2\sqrt{\beta}/\lambda_k, +\infty)$, we have

$$\sqrt{(1 - c\Delta)^2 - 4\beta/\lambda_k^2} = \sqrt{\left((1 - c) + c\frac{2\sqrt{\beta}}{\lambda_k}\right)^2 - \frac{4\beta}{\lambda_k^2}} \geqslant (1 - c)\sqrt{1 - 4\beta/\lambda_k^2} \geqslant (1 - c)\Delta,$$

from which we deduce

$$\kappa \leqslant 1 + 2\frac{c\Delta}{(1 - c)\Delta - c\Delta} \leqslant 1 + 2\frac{c}{1 - 2c} = \frac{16}{15}.$$

$\square$

We can now prove the upper bounds on the spectral norms of the factors appearing in (28).

**Lemma C.8.** *For all $t \geqslant 1$, for all $s \in \{0, \ldots, t - 1\}$,*

$$\|\mathbf{C}_t^{-1}\|_2 \leqslant \frac{c_1}{\left((1 - c)\lambda_k^+ + c\sqrt{\beta}\right)^t},$$

$$\|\mathbf{C}_{t-1-s}\mathbf{C}_t^{-1}\|_2 \leqslant \frac{c_1}{\left((1 - c)\lambda_k^+ + c\sqrt{\beta}\right)^{s+1}},$$

$$\|p_t(\mathbf{\Lambda}_{-k})\|_2 \leqslant \sqrt{\beta}^t,$$

$$\|q_t(\mathbf{\Lambda}_{-k})\|_2 \leqslant (t + 1)\sqrt{\beta}^t,$$

*where $c_1 := 31/15$ and $\lambda_k^+$ is defined as in (29) (i.e. as the largest root of $x^2 - \lambda_k x + \beta$).*

*Proof.* Recall that $\mathbf{C}_t = \prod_{s=0}^{t-1}\mathbf{\Lambda}_k\mathbf{G}_{t-1-s}^{-1}$. Thus, we have

$$\|\mathbf{C}_t^{-1}\|_2 \leqslant \prod_{u=0}^{t-1}\|\mathbf{G}_u\|_2\|\mathbf{\Lambda}_k^{-1}\|_2,$$

$$\|\mathbf{C}_{t-1-s}\mathbf{C}_t^{-1}\|_2 \leqslant \prod_{u=t-1-s}^{t-1}\|\mathbf{G}_u\|_2\|\mathbf{\Lambda}_k^{-1}\|_2.$$

From Lemma C.7, we have for all $u \geqslant 0$,

$$\|\mathbf{G}_u\|_2 \leqslant \frac{r_+^u + \kappa r_-^u}{r_+^{u+1} + \kappa r_-^{u+1}},$$

and $\|\mathbf{\Lambda}_k^{-1}\|_2 = 1/\lambda_k$. Thus, we have

$$\|\mathbf{C}_t^{-1}\|_2 \leqslant \frac{1}{\lambda_k^t}\prod_{u=0}^{t-1}\frac{r_+^u + \kappa r_-^u}{r_+^{u+1} + \kappa r_-^{u+1}} = \frac{1 + \kappa}{\lambda_k^t(r_+^t + \kappa r_-^t)} \leqslant \frac{c_1}{(\lambda_k r_+)^t},$$

$$\|\mathbf{C}_{t-1-s}\mathbf{C}_t^{-1}\|_2 \leqslant \frac{1}{\lambda_k^{s+1}}\prod_{u=t-1-s}^{t-1}\frac{r_+^u + \kappa r_-^u}{r_+^{u+1} + \kappa r_-^{u+1}} = \frac{r_+^{t-1-s} + \kappa r_-^{t-1-s}}{\lambda_k^{s+1}(r_+^t + \kappa r_-^t)} \leqslant \frac{c_1}{(\lambda_k r_+)^{s+1}},$$

since $\kappa \leqslant c_1 - 1$ from Lemma C.7 and since $r_+ \geqslant r_-$. Analyzing the factor $\lambda_k r_+$, we have

$$
\begin{aligned}
\lambda_k r_+ &= \frac{1}{2} \left( (1 - c\Delta)\lambda_k + \sqrt{(1 - c\Delta)^2 \lambda_k^2 - 4\beta} \right) \\
&= \frac{1}{2} \left( (1 - c\Delta)\lambda_k + \sqrt{\left( (1 - c)\lambda_k + c\sqrt{\beta} \right)^2 - 4\beta} \right) \\
&\geqslant \frac{1}{2} \left( (1 - c\Delta)\lambda_k + (1 - c)\sqrt{\lambda_k^2 - 4\beta} \right) \\
&= (1 - c)\lambda_k^+ + c\sqrt{\beta},
\end{aligned}
$$

where the inequality is due to the concavity of the function $u \mapsto \sqrt{u^2 - 4\beta}$ over $[2\sqrt{\beta}, +\infty)$. This gives the desired bounds on $\|\mathbf{C}_t^{-1}\|_2$ and $\|\mathbf{C}_{t-1-s}\mathbf{C}_t^{-1}\|_2$.

The bounds on $\|p_t(\mathbf{\Lambda}_{-k})\|_2$ and $\|q_t(\mathbf{\Lambda}_{-k})\|_2$ are obtained through the closed-form expressions of $p_t$ and $q_t$ shown in Lemma C.6. Recall that $\mathbf{\Lambda}_{-k}$ is a diagonal matrix whose diagonal coefficients are all in the interval $[0, 2\sqrt{\beta}]$. Furthermore, for all $x \in [0, 2\sqrt{\beta}]$, we have that

$$
|x^{\pm}| = \frac{1}{2}\sqrt{x^2 + 4\beta - x^2} = \sqrt{\beta}
$$

Then, from Lemma C.6, we have that for all $x \in [0, 2\sqrt{\beta}]$,

$$
|p_t(x)| \leqslant \frac{1}{2} \left( |x^+|^t + |x^-|^t \right) = \sqrt{\beta}^t,
$$

$$
|q_t(x)| \leqslant \sum_{u=0}^{t} |x^+|^u |x^-|^{t-u} = (t+1)\sqrt{\beta}^t,
$$

which concludes the proof. $\qquad\square$

Using these bounds, we can now prove a recurrence inequality on $h_t = \|\mathbf{H}_t\|_2 = \tan\theta_k(\mathbf{U}_k, \mathbf{X}_t)$ using the expression of $\mathbf{H}_t$ given in (28) which depends on $p_t(\mathbf{\Lambda}_{-k})$, $q_t(\mathbf{\Lambda}_{-k})$, $\mathbf{C}_t^{-1}$ and $\mathbf{C}_{t-s-1}\mathbf{C}_t^{-1}$.

**Lemma C.9.** *For all $t \geqslant 1$, we have*

$$
h_t \leqslant c_1 \gamma^t h_0 + \eta \sum_{s=0}^{t-1} (s+1)\gamma^s (1 + h_{t-1-s}), \tag{30}
$$

*where*

$$
\gamma := \frac{\sqrt{\beta}}{(1 - c)\lambda_k^+ + c\sqrt{\beta}} \in [0, 1),
$$

$$
\eta := c_2 \Delta\varepsilon,
$$

*and $c_2 := 2/15$.*

*Proof.* Let $t \geqslant 1$. From (28), we have

$$
h_t = \|\mathbf{H}_t\|_2 \leqslant \|p_t(\mathbf{\Lambda}_{-k})\|_2 \|\mathbf{H}_0\|_2 \|\mathbf{C}_t^{-1}\|_2 + \sum_{s=0}^{t-1} \|q_s(\mathbf{\Lambda}_{-k})\|_2 \|\mathbf{\Psi}_{t-1-s}\|_2 \|\mathbf{C}_{t-1-s}\mathbf{C}_t^{-1}\|_2.
$$

Using the bounds shown in Lemma C.8, we deduce the following upper bound on $h_t$:

$$
h_t \leqslant \sqrt{\beta}^t h_0 \frac{c_1}{\left( (1 - c)\lambda_k^+ + c\sqrt{\beta} \right)^t} + \sum_{s=0}^{t-1} (s+1)\sqrt{\beta}^s \cdot \|\mathbf{\Psi}_{t-1-s}\|_2 \cdot \frac{c_1}{\left( (1 - c)\lambda_k^+ + c\sqrt{\beta} \right)^{s+1}}. \tag{31}
$$

Furthermore, from the definition of $\boldsymbol{\Psi}_t$ in (23), we have for all $t \geqslant 0$,

$$\|\boldsymbol{\Psi}_t\|_2 = \left\|\left(\mathbf{U}_{-k}^\top \boldsymbol{\Xi}_t\right)\left(\mathbf{U}_k^\top \mathbf{X}_t\right)^{-1}\right\|_2 \leqslant \|\mathbf{U}_{-k}^\top \boldsymbol{\Xi}_t\|_2 \|(\mathbf{U}_k^\top \mathbf{X}_t)^{-1}\|_2$$
$$\leqslant c\lambda_k \Delta\varepsilon(\cos\theta_k(\mathbf{U}_k, \mathbf{X}_t))^{-1} = c\lambda_k \Delta\varepsilon(1 + h_t^2)^{1/2}$$
$$\leqslant c\lambda_k \Delta\varepsilon(1 + h_t),$$

where the second inequality is from the noise condition on $\mathbf{U}_{-k}^\top \boldsymbol{\Xi}_t$ (11), the second equality is from the trigonometric identity $\cos\theta = (1 + \tan^2\theta)^{-1/2}$ and the last inequality holds because $\sqrt{1 + x^2} \leqslant 1 + x$ for all $x \geqslant 0$. Plugging the bound on $\|\boldsymbol{\Psi}_t\|_2$ into the bound on $h_t$ (31) gives:

$$h_t \leqslant c_1\gamma^t h_0 + \frac{cc_1\lambda_k \Delta\varepsilon}{(1-c)\lambda_k^+ + c\sqrt{\beta}} \sum_{s=0}^{t-1}(s+1)\gamma^s(1 + h_{t-1-s}). \tag{32}$$

Since $\lambda_k^+ \geqslant \lambda_k/2$, we get

$$\frac{cc_1\lambda_k}{(1-c)\lambda_k^+ + c\sqrt{\beta}} \leqslant \frac{cc_1\lambda_k}{(1-c)\lambda_k/2} = c_2,$$

which gives the desired result when plugged into (32). $\qquad\square$

Finally, we show that any sequence $\{h_t\}$ satisfying the recurrence inequality (30) decays exponentially as $(1 - \sqrt{\Delta}/2)^t$ until it becomes $\varepsilon$-small.

**Lemma C.10.** *For all $t \geqslant 0$,*

$$h_t \leqslant \varepsilon/2 + h_0\left(1 - \sqrt{\Delta}/2\right)^t. \tag{33}$$

*Proof.* Let $\{y_t\}$ be the sequence defined as $y_0 = h_0$ and for all $t \geqslant 1$,

$$y_t = c_1\gamma^t h_0 + \eta\sum_{s=0}^{t-1}(s+1)\gamma^s(1 + y_{t-1-s}). \tag{34}$$

Then, for all $t \geqslant 0$, we have $h_t \leqslant y_t$ (this can be shown by a simple induction using the recurrence inequality (30)). The proof will then consist in upper bounding $y_t$ by the right-hand side of (33).

To do so, consider the generating function of the sequence $\{y_t\}$ defined as $Y(z) := \sum_{t=0}^{+\infty} y_t z^t$. We will use the following identities, which are true for all $|z| < 1/\gamma$:

$$\sum_{t=0}^{+\infty}\gamma^t z^t = \frac{1}{1-\gamma z}, \qquad \sum_{t=0}^{+\infty}(t+1)\gamma^t z^t = \frac{1}{(1-\gamma z)^2}.$$

Then, by multiplying both sides of (34) by $z^t$ and summing over all $t \geqslant 1$, we have

$$Y(z) - y_0 = c_1 h_0\sum_{t=1}^{+\infty}\gamma^t z^t + \eta z\sum_{t=1}^{+\infty}\sum_{s=0}^{t-1}(s+1)\gamma^s z^{t-1} + \eta z\sum_{t=1}^{+\infty}\sum_{s=0}^{t-1}(s+1)\gamma^s y_{t-1-s}z^{t-1},$$

$$Y(z) - h_0 = c_1 h_0\left(\frac{1}{1-\gamma z} - 1\right) + \eta z\frac{1}{(1-\gamma z)^2}\frac{1}{1-z} + \eta z Y(z)\frac{1}{(1-\gamma z)^2},$$

$$Y(z)\left(1 - \frac{\eta z}{(1-\gamma z)^2}\right) = h_0\left(1 + c_1\frac{\gamma z}{1-\gamma z}\right) + \eta z\frac{1}{(1-\gamma z)^2}\frac{1}{1-z},$$

$$Y(z) = \frac{h_0\left(1 + c_1\frac{\gamma z}{1-\gamma z}\right) + \eta z\frac{1}{(1-\gamma z)^2}\frac{1}{1-z}}{1 - \frac{\eta z}{(1-\gamma z)^2}}.$$

Here, the second inequality used the fact that the generating function of the convolution of two sequences is the product of the two associated generating functions. These equalities are true for all $z$ whose magnitude is less than the pole of the right-hand side with lowest magnitude, which is non-zero. Indeed, the poles of $Y(z)$ are 1 and the roots of the polynomial $(1 - \gamma z)^2 - \eta z$, which are

$$\rho_\pm := \frac{2\gamma + \eta \pm \sqrt{(2\gamma + \eta)^2 - 4\gamma^2}}{2\gamma^2} > 0.$$

We can then decompose $Y(z)$ into partial fractions:

$$Y(z) = \frac{\kappa_1}{1 - z} + \frac{\kappa_+}{1 - z/\rho_+} + \frac{\kappa_-}{1 - z/\rho_-}, \tag{35}$$

from which we deduce that for all $t \geq 0$,

$$y_t = \kappa_1 + \kappa_- \left(\frac{1}{\rho_-}\right)^t + \kappa_+ \left(\frac{1}{\rho_+}\right)^t \leq \kappa_1 + (\kappa_+ + \kappa_-) \left(\frac{1}{\rho_-}\right)^t$$

$$h_t \leq \kappa_1 + (\kappa_+ + \kappa_-) \left(\frac{1}{\rho_-}\right)^t. \tag{36}$$

Here, the first inequality is due to the fact that $\kappa_+$ is non-negative. Indeed, multiplying both sides of (49) by $(1 - z/\rho_+)$ and taking the limit $z \to \rho_+$ gives after simplification

$$\kappa_+ = \frac{h_0(1 - \gamma\rho_+)(1 - \gamma\rho_+ + c_1\gamma\rho_+) + \eta\rho_+/(1 - \rho_+)}{1 - \rho_+/\rho_-},$$

which is non-negative since $\rho_+ > \rho_-$, $1 - \gamma\rho_+ < 0$, $1 - \rho_+ < 0$ and $c_1 \geq 1$.

Multiplying both sides of the partial fraction decomposition of $Y(z)$ in (49) by $(1 - z)$ and taking the limit $z \to 1$ gives

$$\kappa_1 = \frac{\eta}{(1 - \gamma)^2 - \eta}. \tag{37}$$

Evaluating (49) at $z = 0$ gives

$$\kappa_+ + \kappa_- = h_0 - \kappa_1. \tag{38}$$

We now analyze in detail the term $1 - \gamma$. We have

$$1 - \gamma = 1 - \frac{\sqrt{\beta}}{(1 - c)\lambda_k^+ + c\sqrt{\beta}}$$

$$\geq 1 - (1 - c)\frac{\sqrt{\beta}}{\lambda_k^+} - c$$

$$= (1 - c)\left(1 - \frac{\sqrt{\beta}}{\lambda_k^+}\right), \tag{39}$$

where the inequality is due to the convexity of $u \mapsto 1/u$. Rewriting $1 - \sqrt{\beta}/\lambda_k^+$ in terms of $\Delta = 1 - 2\sqrt{\beta}/\lambda_k$ gives

$$1 - \frac{\sqrt{\beta}}{\lambda_k^+} = \frac{\lambda_k + \sqrt{\lambda_k^2 - 4\beta} - 2\sqrt{\beta}}{\lambda_k + \sqrt{\lambda_k^2 - 4\beta}} = \sqrt{\Delta} \underbrace{\frac{\sqrt{2 - \Delta} - \sqrt{\Delta}}{1 - \Delta}}_{=:f(\Delta)}. \tag{40}$$

Then, according to Lemma C.11, for all $\Delta' \in (0, 1)$, we have $f(\Delta') \geq 1$, so that

$$1 - \frac{\sqrt{\beta}}{\lambda_k^+} \geq \sqrt{\Delta}.$$

Plugging this inequality into the previously found lower bound on $(1 - \gamma)$ (39) gives

$$1 - \gamma \geqslant (1 - c)\sqrt{\Delta}. \tag{41}$$

Then, $(1 - \gamma)^2 - \eta \geqslant (1 - c)^2 \Delta - c_2 \varepsilon \Delta > 0$, so that $\kappa_1 > 0$. We deduce from (38) that $\kappa_+ + \kappa_- \leqslant h_0$. Furthermore, from the expression of $\kappa_1$ in (37), we have

$$\kappa_1 = \frac{c_2 \Delta \varepsilon}{(1 - \gamma)^2 - c_2 \Delta \varepsilon} \leqslant \frac{c_2 \Delta \varepsilon}{(1 - c)^2 \Delta - c_2 \Delta \varepsilon} \leqslant \frac{c_2 \varepsilon}{(1 - c)^2 - c_2} \leqslant \varepsilon/2.$$

Plugging these inequalities into the upper bound on $h_t$ (36) gives

$$h_t \leqslant \varepsilon/2 + h_0 \left( \frac{1}{\rho_-} \right)^t. \tag{42}$$

Finally, we analyze the factor $1/\rho_-$. We have

$$\frac{1}{\rho_-} = \gamma \frac{1}{1 + \frac{\eta}{2\gamma} - \sqrt{\left(1 + \frac{\eta}{2\gamma}\right)^2 - 1}} = \gamma \left(1 + \frac{\eta}{2\gamma} + \sqrt{\left(1 + \frac{\eta}{2\gamma}\right)^2 - 1}\right)$$

$$= \gamma + \frac{\eta}{2} + \sqrt{\left(\gamma + \frac{\eta}{2}\right)^2 - \gamma^2} = \gamma + \frac{\eta}{2} + \sqrt{\eta\gamma + \frac{\eta^2}{4}}$$

$$\leqslant \gamma + \frac{\eta}{2} + \sqrt{\eta + \eta^2/4},$$

where the last inequality is because $\gamma \leqslant 1$. Since, $\eta = c_2 \varepsilon \Delta \in [0, 1]$, we have $\eta^2 \leqslant \eta \leqslant \sqrt{\eta}$, so that

$$\frac{1}{\rho_-} \leqslant \gamma + \frac{1 + \sqrt{5}}{2}\sqrt{\eta} \leqslant 1 - (1 - c)\sqrt{\Delta} + \frac{1 + \sqrt{5}}{2}\sqrt{c_2 \varepsilon \Delta}$$

$$\frac{1}{\rho_-} \leqslant 1 - \sqrt{\Delta}/2, \tag{43}$$

where the second equality follows from the inequality of $1 - \gamma$ (41). Plugging the above bound on $1/\rho_-$ (43) into the inequality on $h_t$ in (42) gives the desired result:

$$h_t \leqslant \varepsilon/2 + h_0 \left(1 - \sqrt{\Delta}/2\right)^t.$$

$\square$

Using the geometric decay of $h_t$ we just showed, we can finally prove Theorem 2.2.

*Proof of Theorem 2.2.* The proof directly follows from Lemma C.10 and the definition of $h_t = \tan \theta_k(\mathbf{U}_k, \mathbf{X}_t)$. Indeed, from the inequality $\sin \theta \leqslant \tan \theta$ that holds for all $\theta \in [0, \pi/2)$, we have for all $t \geqslant 0$,

$$\sin \theta_k(\mathbf{U}_k, \mathbf{X}_t) \leqslant h_t.$$

Thus, from Lemma C.10, we have for all $t \geqslant 0$,

$$\sin \theta_k(\mathbf{U}_k, \mathbf{X}_t) \leqslant \varepsilon/2 + \tan \theta_k(\mathbf{U}_k, \mathbf{X}_0)(1 - \sqrt{\Delta}/2)^t.$$

For all $t \geqslant T$ where $T$ is defined in the statement of Theorem C.2, we have that the second term on the right-hand side is less than $\varepsilon/2$, so that $\sin \theta_k(\mathbf{U}_k, \mathbf{X}_t) \leqslant \varepsilon$, which concludes the proof.

$\square$

We conclude this section by proving the technical lemma used in the proof of Lemma C.10, which bounds a function $f$ of the gap $\Delta$ over $[0, 1)$. We prove additional bounds on this function beyond the one that is necessary for the proof of Theorem 2.2, as these additional bounds will be useful in the proofs of the theorems in Section 2.2.

**Lemma C.11.** *For all $\Delta \in [0, 1)$, let*

$$f(\Delta) := \frac{\sqrt{2 - \Delta} - \sqrt{\Delta}}{1 - \Delta}.$$

*Then, $f$ is decreasing over $[0, 1)$. In particular, we have $1 \leqslant f(\Delta) \leqslant \sqrt{2}$ for all $\Delta \in [0, 1)$, and $f(\Delta) \leqslant \sqrt{3} - \sqrt{2} < 1$ for all $\Delta \in [1/2, 1)$.*

*Proof.* For all $\Delta \in [0, 1)$, we have that

$$f(\Delta) = 2\frac{1 - \Delta}{\sqrt{2 - \Delta} + \sqrt{\Delta}}\frac{1}{1 - \Delta} = \frac{2}{\sqrt{2 - \Delta} + \sqrt{\Delta}}.$$

Letting $g(\Delta) := \sqrt{2 - \Delta} + \sqrt{\Delta}$, we have $f(\Delta) = 2/g(\Delta)$. $g$ is differentiable over $(0, 1)$ with derivative $g'(\Delta) = -1/(2\sqrt{2 - \Delta}) + 1/(2\sqrt{\Delta})$. Thus, $g'(\Delta) > 0$ for all $\Delta \in (0, 1)$, so that $g$ is increasing over $[0, 1)$. Thus, $f$ is decreasing over $[0, 1)$, for all $\Delta \in [0, 1)$,

$$1 = \frac{2}{g(1)} \leqslant f(\Delta) \leqslant \frac{2}{g(0)} = \sqrt{2}.$$

and for all $\Delta \in [1/2, 1)$,

$$f(\Delta) \leqslant f(1/2) = \sqrt{3} - \sqrt{2} < 1.$$

$\square$

## C.3. Behavior of ANPM with $0 < \beta < \lambda_{k+1}^2/4$

In this section, we prove a result which is not stated in Section 2, which guarantees that performing ANPM with a momentum parameter $\beta$ smaller than $\lambda_{k+1}^2/4$ (a regime not covered by Theorem 2.2) still improves the convergence rate compared to the non-accelerated noisy power method. This result is interesting as it shows that using a small momentum parameter does not make convergence worse than NPM, and that it can still be useful in practice even when the condition $\beta \geqslant \lambda_{k+1}^2/4$ is not satisfied. For all of this section, we will use the same notations as those used in Appendix C.2.

**Theorem C.12.** *Let $\varepsilon \in (0, 1)$, let $\mathbf{A} \succeq \mathbf{0}$ such that $\lambda_k > \lambda_{k+1}$ and let $\mathbf{X}_0$ such that $\cos \theta_k(\mathbf{U}_k, \mathbf{X}_0) > 0$. Consider the ANPM iterates $\{\mathbf{X}_t\}_{t \geqslant 0}$ defined by (9)-(10) with momentum parameter $\beta > 0$ satisfying $\lambda_{k+1} > 2\sqrt{\beta}$ and perturbations $\{\mathbf{\Xi}_t\}_{t \geqslant 0}$ satisfying*

$$\|\mathbf{U}_{-k}^\top \mathbf{\Xi}_t\|_2 \leqslant c(\lambda_k - \lambda_{k+1})\varepsilon, \tag{44}$$

$$\|\mathbf{U}_k^\top \mathbf{\Xi}_t\|_2 \leqslant c(\lambda_k - \lambda_{k+1}) \cos \theta_k(\mathbf{U}_k, \mathbf{X}_t), \tag{45}$$

*where $c = 1/32$. Then, for all $t \geqslant T$, $\sin \theta_k(\mathbf{U}_k, \mathbf{X}_t) \leqslant \varepsilon$, where*

$$T = \mathcal{O}\left(\frac{\lambda_k^+}{\lambda_k^+ - \lambda_{k+1}^+} \log\left(\frac{\tan \theta_k(\mathbf{U}_k, \mathbf{X}_0)}{\varepsilon}\right)\right).$$

*Remark* C.13. For all $\beta \in (0, \lambda_{k+1}^2/4)$, we have that

$$\frac{\lambda_k^+}{\lambda_k^+ - \lambda_{k+1}^+} = \frac{1}{1 - \frac{\lambda_{k+1}}{\lambda_k}\frac{1+\sqrt{1-4\beta/\lambda_{k+1}^2}}{1+\sqrt{1-4\beta/\lambda_k^2}}} < \frac{\lambda_k}{\lambda_k - \lambda_{k+1}},$$

so that the convergence rate of ANPM in this regime is better than that of the non-accelerated Noisy Power Method shown by Hardt & Price (2014).

*Proof.* It is easy to see that under the noise condition on $\mathbf{U}_k^\top \Xi_t$ (45) and the assumption that $\beta < \lambda_{k+1}^2/4$, the proofs of Proposition C.3 and Lemmas C.4 and C.5 can be adapted and that the results are valid with $\Delta$ defined to be $\Delta := 1 - \lambda_{k+1}/\lambda_k \in (0,1)$ instead of $1 - 2\sqrt{\beta}/\lambda_k$. We get in particular from that that all of the sequences introduced in Appendix C.2 are well defined, and that for all $t \geqslant 0$,

$$\mathbf{H}_t = p_t(\mathbf{\Lambda}_{-k})\mathbf{H}_0\mathbf{C}_t^{-1} + \sum_{s=0}^{t-1} q_s(\mathbf{\Lambda}_{-k})\mathbf{\Psi}_{t-1-s}\mathbf{C}_{t-1-s}\mathbf{C}_t^{-1}, \tag{46}$$

where $\mathbf{C}_t$ and $\mathbf{\Psi}_t$ were defined in Lemma C.4, and $p_t$ and $q_t$ are the scaled Chebyshev polynomials defined in (26)-(27). Furthermore, the bounds shown in Lemma C.7 are also valid with our new definition of $\Delta$, and we have that for all $t \geqslant 0$,

$$\|\mathbf{G}_t\|_2 \leqslant \frac{r_+^t + \kappa r_-^t}{r_+^{t+1} + \kappa_-^{t+1}},$$

$$r_\pm := \frac{(1 - c\Delta) \pm \sqrt{(1 - c\Delta)^2 - 4\beta/\lambda_k^2}}{2},$$

$$\kappa \in [0, 16/15].$$

Then, the proof for the bounds on $\|\mathbf{C}_t^{-1}\|_2$ and $\|\mathbf{C}_{t-1-s}\mathbf{C}_t^{-1}\|_2$ shown in Lemma C.8 can be adapted to our new setting, using the definition of $\mathbf{C}_t = \mathbf{\Lambda}_k\mathbf{G}_{t-1}^{-1}\ldots\mathbf{\Lambda}_k\mathbf{G}_0^{-1}$. We then have for all $t \geqslant 1$, $s \in \{0,\ldots,t-1\}$,

$$\|\mathbf{C}_t^{-1}\|_2 \leqslant \frac{c_1}{\lambda_k^t r_+^t},$$

$$\|\mathbf{C}_{t-1-s}\mathbf{C}_t^{-1}\|_2 \leqslant \frac{c_1}{\lambda_k^{s+1} r_+^{s+1}}.$$

where $c_1 := 31/15$. We can then bound $\lambda_k r_+$ as

$$\begin{aligned}
\lambda_k r_+ &= \frac{1}{2}\left((1 - c\Delta)\lambda_k + \sqrt{(1-c\Delta)^2\lambda_k^2 - 4\beta}\right) \\
&= \frac{1}{2}\left((1 - c\Delta)\lambda_k + \sqrt{\left((1-c)\lambda_k + c\sqrt{\beta}\right)^2 - 4\beta}\right) \\
&\geqslant \frac{1}{2}\left((1 - c\Delta)\lambda_k + (1-c)\sqrt{\lambda_k^2 - 4\beta}\right) \\
&= (1-c)\lambda_k^+ + c\lambda_{k+1} \geqslant (1-c)\lambda_k^+ + c\lambda_{k+1}^+,
\end{aligned}$$

where the first inequality is due to the concavity of the function $u \mapsto \sqrt{u^2 - 4\beta}$ over $[2\sqrt{\beta}, +\infty)$. The final bound we need is on $\|p_t(\mathbf{\Lambda}_{-k})\|_2$. Since $\lambda_{k+1} > 2\sqrt{\beta}$, the bound from Lemma C.8 is not valid anymore, and we instead have

$$\|p_t(\mathbf{\Lambda}_{-k})\|_2 = p_t(\lambda_{k+1}) = \frac{(\lambda_{k+1}^+)^t + (\lambda_{k+1}^-)^t}{2} \leqslant (\lambda_{k+1}^+)^t.$$

We now upper bound $h_t := \|\mathbf{H}_t\|_2$ using (46) and the bounds we just obtained. We then have, using the same techniques as those used in the proof of Lemma C.9, for all $t \geqslant 1$,

$$h_t \leqslant c_1\gamma^t h_0 + \frac{\eta}{b}\sum_{s=0}^{t-1} q_s(\lambda_{k+1})\frac{1}{b^s}(1 + h_{t-1-s}), \tag{47}$$

where

$$b := (1-c)\lambda_k^+ + c\lambda_{k+1}^+, \qquad \gamma := \lambda_{k+1}^+/b \in (0,1),$$

$$\eta := c_1 c\lambda_k\Delta\varepsilon, \qquad c_2 := 2/15.$$

Note that in (47), we do not upper bound $q_s(\lambda_{k+1})$ as in Lemma C.9, which would lead to a loose bound since in the case where $\lambda_{k+1} > 2\sqrt{\beta}$, $\lambda_{k+1}^+$ and $\lambda_{k+1}^-$ can be very different from each other. We will thus rely on a finer analysis of the above recurrence inequality, directly involving $q_s(\lambda_{k+1})$. Letting $y_t$ be the sequence defined as $y_0 = h_0$ and for all $t \geqslant 1$,

$$y_t = c_1 \gamma^t h_0 + \frac{\eta}{b} \sum_{s=0}^{t-1} q_s(\lambda_{k+1}) \frac{1}{b^s} (1 + y_{t-1-s}),$$

we have for all $t \geqslant 0$, $h_t \leqslant y_t$, since $q_s(\lambda_{k+1}) \geqslant 0$ for all $s$. We can then analyze the generating function $Y(z) := \sum_{t=0}^{+\infty} y_t z^t$ of the sequence $\{y_t\}$. Using the same techniques as those used in the proof of Lemma C.10, we have

$$Y(z) - h_0 = c_1 h_0 \frac{\gamma z}{1 - \gamma z} + \frac{\eta z}{b} Q(z) \left( \frac{1}{1-z} + Y(z) \right),$$

where $Q(z)$ is defined as

$$Q(z) = \sum_{s=0}^{+\infty} q_s(\lambda_{k+1}) \left( \frac{z}{b} \right)^s = \frac{1}{1 - \lambda_{k+1} z/b + \beta z^2/b^2}.$$

Here, the expression of the generating function of the sequence $\{q_s(\lambda_{k+1})\}$ comes from the proof of Lemma C.6. We deduce the following closed-form expression for $Y(z)$:

$$Y(z) = \frac{h_0 \left( 1 + c_1 \frac{\gamma z}{1 - \gamma z} \right) + \frac{\eta z}{b} \frac{1}{1-z} \frac{1}{1 - \lambda_{k+1} z/b + \beta z^2/b^2}}{1 - \frac{\eta z}{b} \frac{1}{1 - \lambda_{k+1} z/b + \beta z^2/b^2}}. \tag{48}$$

From this expression, we can decompose $Y(z)$ into partial fractions:

$$Y(z) = \frac{\kappa_1}{1-z} + \frac{\kappa_+}{1 - z/\rho_+} + \frac{\kappa_-}{1 - z/\rho_-}, \tag{49}$$

where $\rho_\pm$ are the roots of the polynomial $\frac{\beta}{b^2} z^2 - \frac{\lambda_{k+1} + \eta}{b} z + 1$, i.e.,

$$\rho_\pm := b \frac{\lambda_{k+1} + \eta \pm \sqrt{(\lambda_{k+1} + \eta)^2 - 4\beta}}{2\beta}.$$

Multiplying both sides of the expressions of $Y(z)$ in (48) and (49) by $1 - z$ and evaluating at $z = 1$ gives

$$\kappa_1 = \frac{\eta/b}{1 - \lambda_{k+1}/b + \beta/b^2 - \eta/b} = \frac{\eta}{b - \lambda_{k+1} + \beta/b - \eta}. \tag{50}$$

We will first upper bound $\kappa_1$, to show that it is of order $\varepsilon$. To do so, consider the function $g(x) = x + \beta/x$ for $x > 0$. This function is convex, and we have the following inequality:

$$g(b) - g(\lambda_{k+1}^+) \geqslant g'(\lambda_{k+1}^+)(b - \lambda_{k+1}^+)$$
$$= g'(\lambda_{k+1}^+)(1 - c)(\lambda_k^+ - \lambda_{k+1}^+)$$

where the equality is from the definition of $b = (1-c)\lambda_k^+ + c\lambda_{k+1}^+$. Notice that because $\lambda_{k+1}^+$ is a root of $z^2 - \lambda_{k+1} z + \beta$, we have $g(\lambda_{k+1}^+) = \lambda_{k+1}$. Similarly, we have $g(\lambda_k^+) = \lambda_k$. The above inequality can be rewritten as

$$b - \lambda_{k+1} + \beta/b \geqslant g'(\lambda_{k+1}^+)(1 - c)(\lambda_k^+ - \lambda_{k+1}^+). \tag{51}$$

Next, by the mean value theorem, we have that

$$0 \leqslant \frac{g(\lambda_k^+) - g(\lambda_{k+1}^+)}{\lambda_k^+ - \lambda_{k+1}^+} \leqslant \sup_{x \in [\lambda_{k+1}^+, \lambda_k^+]} |g'(x)| = g'(\lambda_{k+1}^+),$$

so that we finally deduce the following lower bound on $b - \lambda_{k+1} + \beta/b$ using (51):

$$b - \lambda_{k+1} + \beta/b \geqslant (1-c)\frac{g(\lambda_k^+) - g(\lambda_{k+1}^+)}{\lambda_k^+ - \lambda_{k+1}^+}(\lambda_k^+ - \lambda_{k+1}^+) = (1-c)(\lambda_k - \lambda_{k+1}) = (1-c)\lambda_k\Delta.$$

Plugging this lower bound into the expression of $\kappa_1$ in (50) gives

$$\kappa_1 \leqslant \frac{\eta}{(1-c)\lambda\Delta - \eta} = \frac{c_1 c\lambda\Delta\varepsilon}{(1-c)\lambda\Delta - c_1 c\lambda\Delta\varepsilon} = \frac{c_1 c}{1 - c - c_1 c}\varepsilon \leqslant \varepsilon/2.$$

We now focus on the other terms of the partial fraction decomposition of $Y(z)$ in (49). We can show that $\kappa_+ \geqslant 0$ by multiplying both sides of (49) by $1 - z/\rho_+$ and taking the limit $z \to \rho_+$. Furthermore, evaluating (49) at $z = 0$ gives

$$\kappa_+ + \kappa_- + \kappa_1 = Y(0) = h_0.$$

Then, we get from the partial fraction decomposition of $Y(z)$ in (49) the following upper bound on $h_t$:

$$h_t \leqslant y_t \leqslant \kappa_1 + (\kappa_+ + \kappa_-)\left(\frac{1}{\rho_-}\right)^t \leqslant \varepsilon/2 + h_0\left(\frac{1}{\rho_-}\right)^t. \tag{52}$$

What remains to show is that $\rho_-^{-t}$ decays geometrically with a rate $\tilde{\mathcal{O}}\left(\frac{\lambda_k^+}{\lambda_k^+ - \lambda_{k+1}^+}\right)$. We can write $1/\rho_-$ as

$$\begin{aligned}
\frac{1}{\rho_-} &= \frac{2\beta}{b}\frac{1}{(\lambda_{k+1} + \eta) + \sqrt{(\lambda_{k+1} + \eta)^2 - 4\beta}} \\
&= \frac{1}{2}\frac{(\lambda_{k+1} + \eta) + \sqrt{(\lambda_{k+1} + \eta)^2 - 4\beta}}{(1-c)\lambda_k^+ + c\lambda_{k+1}^+}.
\end{aligned}$$

Since $\eta \leqslant cc_1(\lambda_k - \lambda_{k+1})$, we have

$$\frac{1}{\rho_-} \leqslant \frac{1}{2}\frac{(1 - cc_1)\lambda_{k+1} + cc_1\lambda_k + \sqrt{((1 - cc_1)\lambda_{k+1} + cc_1\lambda_k)^2 - 4\beta}}{(1-c)\lambda_k^+ + c\lambda_{k+1}^+}.$$

Then, by concavity of the function $u \mapsto \sqrt{u^2 - 4\beta}$ over $[2\sqrt{\beta}, +\infty)$, we have

$$\frac{1}{\rho_-} \leqslant \frac{(1 - cc_1)\lambda_{k+1}^+ + c_1 c\lambda_k^+}{(1-c)\lambda_k^+ + c\lambda_{k+1}^+} \leqslant \frac{(1 - cc_1)\lambda_{k+1}^+ + c_1 c\lambda_k^+}{(1 - c_1 c)\lambda_k^+} \leqslant 1 - (1 - c_1 - cc_1)\frac{\lambda_k^+ - \lambda_k^-}{\lambda_k^+}.$$

Denoting $c_2 := 1 - c_1 - cc_1 > 0$, we finally have

$$h_t \leqslant \varepsilon/2 + h_0\left(1 - c_2\frac{\lambda_k^+ - \lambda_{k+1}^+}{\lambda_k^+}\right)^t,$$

so that for all $t \geqslant T$, $\sin\theta_k(\mathbf{U}_k, \mathbf{X}_t) \leqslant h_t \leqslant \varepsilon$, where $T$ is defined as

$$T = \frac{1}{-\log\left(1 - c_2\frac{\lambda_k^+ - \lambda_{k+1}^+}{\lambda_k^+}\right)}\log\left(\frac{2h_0}{\varepsilon}\right) = \mathcal{O}\left(\frac{\lambda_k^+}{\lambda_k^+ - \lambda_{k+1}^+}\log\left(\frac{\tan\theta_k(\mathbf{U}_k, \mathbf{X}_0)}{\varepsilon}\right)\right).$$

$\square$

## C.4. Proofs of Theorems 2.3 to 2.5

We recall the notations introduced in Section 2.2 related to the adversarial examples we use to prove the tightness of our analysis in the proof of Theorem 2.2. Let $d \geqslant k \geqslant 1$, $\lambda_k > 2\sqrt{\beta} > 0$ and $\Delta = \frac{\lambda_k - 2\sqrt{\beta}}{\lambda_k} \in (0,1)$. We define $\mathbf{A} := \mathrm{diag}(\lambda_k, \ldots, \lambda_k, 2\sqrt{\beta}, \ldots, 2\sqrt{\beta}) \in \mathbb{R}^{d \times d}$, where $\lambda_k$ is repeated $k$ times and $2\sqrt{\beta}$ is repeated $d-k$ times. We denote by $\mathbf{U}_k \in \mathbb{R}^{d \times k}$ the matrix whose columns are the first $k$ standard basis vectors of $\mathbb{R}^d$, and by $\mathbf{U}_{-k} \in \mathbb{R}^{d \times (d-k)}$ the matrix whose columns are the last $d-k$ standard basis vectors of $\mathbb{R}^d$. We also denote $\mathbf{\Lambda}_k := \lambda_k \mathbf{I}_k$ and $\mathbf{\Lambda}_{-k} := 2\sqrt{\beta}\mathbf{I}_{d-k}$. It will also be convenient to introduce $\mathbf{v}_1, \ldots, \mathbf{v}_d$ the standard basis vectors of $\mathbb{R}^d$, and $\tilde{\mathbf{v}}_1, \tilde{\mathbf{v}}_2$ the standard basis vectors of $\mathbb{R}^2$. All of the proofs in this section will consider ANPM instances with the matrix $\mathbf{A}$, aiming to approximate the subspace spanned by $\mathbf{U}_k$.

We restate the theorems from Section 2.2 before proving them.

**Theorem C.14** (Theorem 2.3). *Let $\varepsilon \in (0,1)$ and $\mathbf{X}_0 \in \mathrm{St}(d,k)$ such that $\cos\theta_k(\mathbf{U}_k, \mathbf{X}_0) > 0$, and consider the ANPM iterates $\{\mathbf{X}_t\}_{t \geqslant 0}$ defined by (1) with momentum parameter $\beta$ and perturbations $\mathbf{\Xi}_t \equiv \mathbf{0}$. Then, for all $t < T$, $\tan\theta_k(\mathbf{U}_k, \mathbf{X}_t) > \varepsilon$, where*

$$T = \Omega\left(\frac{1}{\sqrt{\Delta}} \log\left(\frac{\tan\theta_k(\mathbf{U}_k, \mathbf{X}_0)}{\varepsilon}\right)\right).$$

*Proof.* In the case where $\mathbf{\Xi}_t \equiv \mathbf{0}$, the ANPM iterations read:

$$\mathbf{X}_1, \mathbf{R}_1 = \mathrm{QR}\left(\frac{1}{2}\mathbf{A}\mathbf{X}_0\right),$$

$$\forall t \geqslant 1, \quad \mathbf{Y}_{t+1} = \mathbf{A}\mathbf{X}_t - \beta\mathbf{X}_{t-1}\mathbf{R}_t^{-1},$$

$$\mathbf{X}_{t+1}, \mathbf{R}_{t+1} = \mathrm{QR}(\mathbf{Y}_{t+1}),$$

so that defining $\mathbf{Z}_0 := \mathbf{X}_0$ and $\mathbf{Z}_t := \mathbf{X}_t \mathbf{R}_t \ldots \mathbf{R}_1$ for all $t \geqslant 1$, we have

$$\forall t \geqslant 1, \quad \mathbf{Z}_{t+1} = \mathbf{A}\mathbf{Z}_t - \beta\mathbf{Z}_{t-1},$$

$$\mathbf{Z}_1 = \frac{1}{2}\mathbf{A}\mathbf{Z}_0.$$

We can then write $\mathbf{Z}_t$ as

$$\mathbf{Z}_t = p_t(\mathbf{A})\mathbf{X}_0,$$

where $p_t$ is the polynomial defined in (26). Since $\mathbf{R}_t \ldots \mathbf{R}_1$ is non-singular, $\tan\theta_k(\mathbf{U}_k, \mathbf{X}_t)$ can conveniently be written as:

$$\tan\theta_k(\mathbf{U}_k, \mathbf{X}_t) = \left\|\left(\mathbf{U}_{-k}^\top \mathbf{X}_t\right)\left(\mathbf{U}_k^\top \mathbf{X}_t\right)^{-1}\right\|_2 = \left\|\left(\mathbf{U}_{-k}^\top \mathbf{Z}_t\right)\left(\mathbf{U}_k^\top \mathbf{Z}_t\right)^{-1}\right\|_2.$$

We deduce from it that for all $t \geqslant 0$,

$$\begin{aligned}
\tan\theta_k(\mathbf{U}_k, \mathbf{X}_t) &= \left\|\left(\mathbf{U}_{-k}^\top p_t(\mathbf{A})\mathbf{X}_0\right)\left(\mathbf{U}_k^\top p_t(\mathbf{A})\mathbf{X}_0\right)^{-1}\right\|_2 \\
&= \underbrace{\|p_t(\mathbf{\Lambda}_{-k})\|_2}_{=p_t(2\sqrt{\beta})} \left\|\left(\mathbf{U}_{-k}^\top \mathbf{X}_0\right)\left(\mathbf{U}_k^\top \mathbf{X}_0\right)^{-1}\right\|_2 \underbrace{\|p_t(\mathbf{\Lambda}_k)^{-1}\|_2}_{=1/p_t(\lambda_k)}, \\
&= \frac{2\sqrt{\beta}^t}{(\lambda_k^+)^t + (\lambda_k^-)^t} \tan\theta_k(\mathbf{U}_k, \mathbf{X}_0) \\
&\geqslant \left(\frac{\sqrt{\beta}}{\lambda_k^+}\right)^t \tan\theta_k(\mathbf{U}_k, \mathbf{X}_0),
\end{aligned}$$

where the last equality is from Lemma C.6, and $\lambda_k^\pm$ are defined in (29). Then, using (40) and Lemma C.11, we know that

$$\frac{\sqrt{\beta}}{\lambda_k^+} = 1 - \sqrt{\Delta}f(\Delta) \geqslant 1 - \alpha\sqrt{\Delta} > 0,$$

where $f$ is defined in Lemma C.11 and $\alpha = \sqrt{3} - \sqrt{2}$ if $\Delta \geqslant 1/2$, and $\alpha = \sqrt{2}$ otherwise. Thus, for all $t \geqslant 0$,

$$\tan \theta_k(\mathbf{U}_k, \mathbf{X}_t) \geqslant (1 - \alpha\sqrt{\Delta})^t \tan \theta_k(\mathbf{U}_k, \mathbf{X}_0).$$

We deduce from it that for all $t < T$, where

$$T = \frac{1}{-\log(1 - \alpha\sqrt{\Delta})} \log\left(\frac{\tan \theta_k(\mathbf{U}_k, \mathbf{X}_0)}{\varepsilon}\right) = \Omega\left(\frac{1}{\sqrt{\Delta}} \log\left(\frac{\tan \theta_k(\mathbf{U}_k, \mathbf{X}_0)}{\varepsilon}\right)\right),$$

we have $\tan \theta_k(\mathbf{U}_k, \mathbf{X}_t) > \varepsilon$. This concludes the proof. $\qquad\square$

**Theorem C.15** (Theorem 2.4). *Let $\varepsilon \in (0, 1)$. There exists $\mathbf{X}_0 \in \mathrm{St}(d, k)$ such that $\cos \theta_k(\mathbf{U}_k, \mathbf{X}_0) > 0$ and a sequence of perturbations $\{\mathbf{\Xi}_t\}_{t \geqslant 0}$ verifying*

$$\|\mathbf{U}_{-k}^\top \mathbf{\Xi}_t\|_2 \leqslant 8(\lambda_k - 2\sqrt{\beta})\varepsilon,$$
$$\|\mathbf{U}_k^\top \mathbf{\Xi}_t\|_2 = 0,$$

*such that the ANPM iterates $\{\mathbf{X}_t\}_{t \geqslant 0}$ defined by (1) with momentum parameter $\beta$ verify $\tan \theta_k(\mathbf{U}_k, \mathbf{X}_t) > \varepsilon$ for all $t \geqslant 0$.*

*Proof.* Let $\theta_0 := \arctan(2\varepsilon)$ and let $\mathbf{X}_0 \in \mathrm{St}(d, k)$ be the matrix whose $k - 1$ first columns are $\mathbf{v}_1, \ldots, \mathbf{v}_{k-1}$, and whose last column is $\cos \theta_0 \mathbf{v}_k + \sin \theta_0 \mathbf{v}_{k+1}$. Notice that since $\mathbf{U}_k$ is the matrix whose columns are $\mathbf{v}_1, \ldots, \mathbf{v}_k$, we have that $\mathbf{U}_k^\top \mathbf{X}_0 = \mathrm{diag}(1, \ldots, 1, \cos \theta_0)$, so that $\cos \theta_k(\mathbf{U}_k, \mathbf{X}_0) = \cos \theta_0$ and $\tan \theta_k(\mathbf{U}_k, \mathbf{X}_0) = \tan \theta_0 = 2\varepsilon > \varepsilon$.

We consider the perturbations $\mathbf{\Xi}_t$ defined as

$$\mathbf{\Xi}_t \equiv 8(\lambda_k - 2\sqrt{\beta})\varepsilon[\mathbf{0}, \ldots, \mathbf{0}, \mathbf{v}_{k+1}] \in \mathbb{R}^{d \times k}.$$

$\mathbf{\Xi}_t$ verifies $\|\mathbf{U}_{-k}^\top \mathbf{\Xi}_t\|_2 \leqslant 8(\lambda_k - 2\sqrt{\beta})\varepsilon$ and $\mathbf{U}_k^\top \mathbf{\Xi}_t = \mathbf{0}$.

Since the $k - 1$ first columns of $\mathbf{X}_0$ are aligned with the top-$k$ eigenvectors of $\mathbf{A}$ and the $k - 1$ first columns of $\mathbf{\Xi}_t$ are zero, the $k - 1$ first columns of $\mathbf{X}_t$ remain constantly equal to $\mathbf{v}_1, \ldots, \mathbf{v}_{k-1}$ throughout the iterations. Furthermore, the $k$-th columns of $\mathbf{X}_0$ and $\mathbf{\Xi}_t$ are contained in $\mathrm{Span}(\mathbf{v}_k, \mathbf{v}_{k+1})$, which is stable by multiplication with $\mathbf{A}$. Thus, we only need to analyze the evolution of the $k$ and $k + 1$-th components of the $k$-th column of $\mathbf{X}_t$, which we denote by $\mathbf{x}_t \in \mathbb{R}^2$. The dynamics then become:

$$\mathbf{y}_1 = \frac{1}{2}\tilde{\mathbf{A}}\mathbf{x}_0, \quad \mathbf{x}_1 = \mathbf{y}_1/\|\mathbf{y}_1\|_2,$$
$$\forall t \geqslant 1, \quad \mathbf{y}_{t+1} = \tilde{\mathbf{A}}\mathbf{x}_t - \beta\mathbf{x}_{t-1}/\|\mathbf{y}_t\|_2 + 8(\lambda_k - 2\sqrt{\beta})\varepsilon\tilde{\mathbf{v}}_2,$$
$$\mathbf{x}_{t+1} = \mathbf{y}_{t+1}/\|\mathbf{y}_{t+1}\|_2,$$

and we have that $\theta_k(\mathbf{U}_k, \mathbf{X}_t) = \theta_1(\tilde{\mathbf{v}}_1, \mathbf{x}_t) = \arccos(\langle \tilde{\mathbf{v}}_1, \mathbf{x}_t \rangle)$. Here $\tilde{\mathbf{A}} := \mathrm{diag}(\lambda_k, 2\sqrt{\beta})$. This is the ANPM dynamics in dimension 2 with perturbations $\tilde{\boldsymbol{\xi}}_t := 8(\lambda_k - 2\sqrt{\beta})\varepsilon\tilde{\mathbf{v}}_2$.

In this setting, the sequence $\{\mathbf{G}_t\}$ defined in (14) reduces to a scalar sequence $\{g_t\}$ defined by the following Riccati recurrence:

$$\forall t \geqslant 0, \quad g_{t+1} = \frac{1}{1 - \frac{\beta}{\lambda_k^2}g_t}, \quad g_0 = 2.$$

It is easy to prove that in this case, for all $t \geqslant 0$,

$$g_t = \lambda_k \frac{p_t(\lambda_k)}{p_{t+1}(\lambda_k)},$$

where $p_t$ is the polynomial defined in (26) (a simple way to prove it would be to notice that the sequence $m_t := \lambda_k^t/(g_{t-1} \cdot \ldots \cdot g_0)$ verifies the linear recurrence $m_{t+1} = \lambda_k m_t - \beta m_{t-1}$, with $m_0 = 1$ and $m_1 = \lambda_k/2$).

The noise condition (12) holds, since $\tilde{\mathbf{v}}_1^\top \tilde{\xi}_t = 0$, so Proposition C.3 and Lemmas C.4 and C.5 still hold. In particular, from Lemma C.5, denoting $h_t := \tan \theta_1(\tilde{\mathbf{v}}_1, \mathbf{x}_t)$, we have for all $t \geqslant 0$,

$$h_t = \frac{p_t(2\sqrt{\beta})}{p_t(\lambda_k)} h_0 + 8(\lambda_k - 2\sqrt{\beta})\varepsilon \sum_{s=0}^{t-1} \frac{q_s(2\sqrt{\beta})}{p_t(\lambda_k)} \frac{p_{t-1-s}(\lambda_k)}{\cos \theta_1(\tilde{\mathbf{v}}_1, \mathbf{x}_{t-1-s})},$$

where $q_t$ is the polynomial defined in (27). Then, using Lemma C.6 and the fact that $1/\cos\theta \geqslant 1$, we have for all $t \geqslant 0$,

$$h_t \geqslant \left(\frac{\sqrt{\beta}}{\lambda_k^+}\right)^t h_0 + 4\varepsilon \frac{(\lambda_k - 2\sqrt{\beta})}{\lambda_k^+} \sum_{s=0}^{t-1} (s+1) \left(\frac{\sqrt{\beta}}{\lambda_k^+}\right)^s$$

$$\overset{\lambda_k^+ \leqslant \lambda_k}{\geqslant} \gamma^t h_0 + 4\Delta\varepsilon \sum_{s=0}^{t-1} (s+1)\gamma^s$$

$$= \gamma^t \underbrace{h_0}_{=2\varepsilon} + 4\Delta\varepsilon \frac{1 - (t+1)\gamma^t + t\gamma^{t+1}}{(1-\gamma)^2},$$

where we defined $\gamma := \sqrt{\beta}/\lambda_k^+ \in (0,1)$. From (40), we have $1 - \gamma = \sqrt{\Delta}f(\Delta)$ where $f$ is defined and bounded in Lemma C.11, so that $1 - \gamma \leqslant \sqrt{\Delta}\sqrt{2}$. Thus, $\Delta \geqslant (1-\gamma)^2/2$ and we have for all $t \geqslant 0$,

$$h_t \geqslant 2\varepsilon \left(1 - t(1-\gamma)\gamma^t\right) =: \varphi(t).$$

$\varphi$ is a differentiable function over $\mathbb{R}_+$ with derivative

$$\varphi'(t) = 2\varepsilon(1-\gamma)(-1 + t\log(1/\gamma))\gamma^t.$$

Letting $t^*$ be defined as

$$t^* := \frac{1}{\log(1/\gamma)} > 0,$$

we have that $\varphi'(t) < 0$ for all $t \in [0, t^*)$ and $\varphi'(t) > 0$ for all $t > t^*$. Thus, $\varphi$ is decreasing over $[0, t^*]$ and increasing over $[t^*, +\infty)$, and is minimized at $t^*$. Thus, for all $t \geqslant 0$,

$$h_t \geqslant \varphi(t^*) = 2\varepsilon \left(1 - \frac{1-\gamma}{e\log(1/\gamma)}\right).$$

Since $\gamma \in (0,1)$, we have $\log(1/\gamma) \geqslant 1 - \gamma$ so that for all $t \geqslant 0$,

$$h_t \geqslant 2\varepsilon \left(1 - \frac{1}{e}\right) > \varepsilon,$$

which concludes the proof. $\qquad\square$

**Theorem C.16** (Theorem 2.5). *Let $\varepsilon \in (0,1)$. There exists $\mathbf{X}_0 \in \mathrm{St}(d,k)$ such that $\cos \theta_k(\mathbf{U}_k, \mathbf{X}_0) > 0$ and a sequence of perturbations $\{\Xi_t\}_{t \geqslant 0}$ verifying*

$$\|\mathbf{U}_{-k}^\top \Xi_t\|_2 = 0,$$
$$\|\mathbf{U}_k^\top \Xi_t\|_2 \leqslant (\lambda_k - 2\sqrt{\beta}) \cos \theta_k(\mathbf{U}_k, \mathbf{X}_t),$$

*such that the ANPM iterates $\{\mathbf{X}_t\}_{t \geqslant 0}$ defined by (1) with momentum parameter $\beta$ verify $\tan \theta_k(\mathbf{U}_k, \mathbf{X}_t) > \varepsilon$ for all $t \geqslant 0$.*

*Proof.* We define $\mathbf{X}_0$ similarly as in the proof of Theorem 2.4, with $\theta_0 := \arctan(2\varepsilon)$. $\mathbf{X}_0 \in \mathrm{St}(d,k)$ is the matrix whose $k-1$ first columns are $\mathbf{v}_1, \ldots, \mathbf{v}_{k-1}$, and whose last column is $\cos\theta_0 \mathbf{v}_k + \sin\theta_0 \mathbf{v}_{k+1}$. Then, we have $\cos \theta_k(\mathbf{U}_k, \mathbf{X}_0) = \cos\theta_0$ and $\tan \theta_k(\mathbf{U}_k, \mathbf{X}_0) = \tan\theta_0 = 2\varepsilon > \varepsilon$.

We define recursively the sequences $\{\mathbf{X}_t\}$ and $\{\boldsymbol{\Xi}_t\}$ starting from $\mathbf{X}_0$:

$$\boldsymbol{\Xi}_0 = -\frac{1}{2}(\lambda_k - 2\sqrt{\beta})\cos\theta_k(\mathbf{U}_k, \mathbf{X}_0)[\mathbf{0}, \ldots, \mathbf{0}, \mathbf{v}_k] \in \mathbb{R}^{d \times k},$$

$$\mathbf{X}_1, \mathbf{R}_1 = \mathrm{QR}\left(\frac{1}{2}\mathbf{A}\mathbf{X}_0 + \boldsymbol{\Xi}_0\right),$$

$$\forall t \geqslant 1, \quad \boldsymbol{\Xi}_t = -(\lambda_k - 2\sqrt{\beta})\cos\theta_k(\mathbf{U}_k, \mathbf{X}_t)[\mathbf{0}, \ldots, \mathbf{0}, \mathbf{v}_k] \in \mathbb{R}^{d \times k},$$

$$\mathbf{Y}_{t+1} = \mathbf{A}\mathbf{X}_t - \beta\mathbf{X}_{t-1}\mathbf{R}_t^{-1} + \boldsymbol{\Xi}_t,$$

$$\mathbf{X}_{t+1}, \mathbf{R}_{t+1} = \mathrm{QR}(\mathbf{Y}_{t+1}).$$

Then, $\{\mathbf{X}_t\}$ follows the ANPM dynamics with perturbations $\{\boldsymbol{\Xi}_t\}$. Notice in particular that for all $t \geqslant 0$,

$$\|\mathbf{U}_k^\top \boldsymbol{\Xi}_t\|_2 \leqslant (\lambda_k - 2\sqrt{\beta})\cos\theta_k(\mathbf{U}_k, \mathbf{X}_t),$$

$$\|\mathbf{U}_{-k}^\top \boldsymbol{\Xi}_t\|_2 = 0.$$

The same argument stated for the proof of Theorem 2.4 holds: since the $k - 1$ first columns of $\mathbf{X}_0$ are aligned with the top-$k$ eigenvectors of $\mathbf{A}$ and the $k - 1$ first columns of $\boldsymbol{\Xi}_t$ are zero, the $k - 1$ first columns of $\mathbf{X}_t$ remain constantly equal to $\mathbf{v}_1, \ldots, \mathbf{v}_{k-1}$ throughout the iterations. Furthermore, the $k$-th columns of $\mathbf{X}_0$ and $\boldsymbol{\Xi}_t$ are contained in $\mathrm{Span}(\mathbf{v}_k, \mathbf{v}_{k+1})$, which is stable by multiplication with $\mathbf{A}$. Thus, we only need to analyze the evolution of the $k$ and $k+1$-th components of the $k$-th column of $\mathbf{X}_t$, which we denote by $\mathbf{x}_t \in \mathbb{R}^2$. The dynamics then become:

$$\mathbf{y}_1 = \frac{1}{2}\tilde{\mathbf{A}}\mathbf{x}_0 - \frac{1}{2}(\lambda_k - 2\sqrt{\beta})\cos\theta_k(\mathbf{U}_k, \mathbf{X}_0)\tilde{\mathbf{v}}_1,$$

$$\mathbf{x}_1 = \mathbf{y}_1/\|\mathbf{y}_1\|_2,$$

$$\forall t \geqslant 1, \quad \mathbf{y}_{t+1} = \tilde{\mathbf{A}}\mathbf{x}_t - \beta\mathbf{x}_{t-1}/\|\mathbf{y}_t\|_2 - (\lambda_k - 2\sqrt{\beta})\cos\theta_k(\mathbf{U}_k, \mathbf{X}_t)\tilde{\mathbf{v}}_1,$$

$$\mathbf{x}_{t+1} = \mathbf{y}_{t+1}/\|\mathbf{y}_{t+1}\|_2,$$

and we have that $\theta_k(\mathbf{U}_k, \mathbf{X}_t) = \theta_1(\tilde{\mathbf{v}}_1, \mathbf{x}_t) = \arccos(\langle\tilde{\mathbf{v}}_1, \mathbf{x}_t\rangle)$. Here $\tilde{\mathbf{A}} := \mathrm{diag}(\lambda_k, 2\sqrt{\beta})$. Then, $\cos\theta_k(\mathbf{U}_k, \mathbf{X}_t) = \tilde{\mathbf{v}}_1^\top\mathbf{x}_t$, and the dynamics of $\mathbf{x}_t$ can be rewritten as:

$$\mathbf{y}_1 = \frac{1}{2}2\sqrt{\beta}\mathbf{x}_0, \quad \mathbf{x}_1 = \mathbf{y}_1/\|\mathbf{y}_1\|_2,$$

$$\forall t \geqslant 1, \quad \mathbf{y}_{t+1} = 2\sqrt{\beta}\mathbf{x}_t - \beta\mathbf{x}_{t-1}/\|\mathbf{y}_t\|_2,$$

$$\mathbf{x}_{t+1} = \mathbf{y}_{t+1}/\|\mathbf{y}_{t+1}\|_2,$$

since $\tilde{\mathbf{A}} - (\lambda_k - 2\sqrt{\beta})\tilde{\mathbf{v}}_1\tilde{\mathbf{v}}_1^\top = 2\sqrt{\beta}\mathbf{I}_2$. From this, we deduce that for all $t \geqslant 0$, $\mathbf{x}_t$ is a unit norm vector in $\mathrm{Span}(\mathbf{x}_0)$, so that $\theta_1(\tilde{\mathbf{v}}_1, \mathbf{x}_t) = \theta_1(\tilde{\mathbf{v}}_1, \mathbf{x}_0) = \theta_k(\mathbf{U}_k, \mathbf{X}_0)$ remains constant for all $t \geqslant 0$. Thus, for all $t \geqslant 0$,

$$\tan\theta_k(\mathbf{U}_k, \mathbf{X}_t) = 2\varepsilon > \varepsilon.$$

$\square$

# D. Proof for Section 3

## D.1. Proof of Theorem 3.3

We first restate Theorem 3.3 with an explicit condition on the number of gossip iterations per step.

**Theorem D.1.** *Let $\varepsilon \in (0, 1)$. Let $\{\mathbf{A}_i\}_{i=1}^n \in (\mathbb{R}^{d \times d})^n$ such that $\mathbf{A} := n^{-1}\sum_{i=1}^n \mathbf{A}_i \succeq \mathbf{0}$ is PSD with eigenvalues $\lambda_1 \geqslant \lambda_k > \lambda_{k+1} \geqslant \cdots \geqslant \lambda_d$ and let $\mathbf{U}_k \in \mathrm{St}(d, k)$ be its top-$k$ eigenvectors. Let $\mathbf{X}_0 \in \mathrm{St}(d, k)$ such that $\cos\theta_k(\mathbf{U}_k, \mathbf{X}_0) > 0$, let $\beta > 0$ such that $\lambda_k > 2\sqrt{\beta} \geqslant \lambda_{k+1}$, and let $\mathbf{W}$ be a gossip matrix. Then, running Algorithm 2 with $L$ gossip communications per step, where $L$ verifies*

$$L \geqslant \frac{c_1}{\sqrt{\gamma_\mathbf{W}}}\log\left(c_2\sqrt{nk}\frac{M}{\lambda_k}\frac{\lambda_k}{\lambda_k - 2\sqrt{\beta}}\frac{1}{\alpha_0}\frac{1}{\varepsilon}\right), \tag{53}$$

*returns for all $t \geqslant T$ and for all $i \in \{1, \ldots, n\}$ an estimate $\mathbf{X}_{i,t} \in \mathrm{St}(d, k)$ such that $\sin \theta_k(\mathbf{U}_k, \mathbf{X}_{i,t}) \leqslant 2\varepsilon$, where*

$$T = \mathcal{O}\left(\sqrt{\frac{\lambda_k}{\lambda_k - \lambda_{k+1}}} \log\left(\frac{\tan \theta_k(\mathbf{U}_k, \mathbf{X}_0)}{\varepsilon}\right)\right).$$

*Here $c_1 := 6$ and $c_2 := 11$ are universal constants, and $\alpha_0$ and $M$ are defined as:*

$$\alpha_0 := \frac{1}{\sqrt{1 + (\varepsilon/2 + \tan \theta_k(\mathbf{U}_k, \mathbf{X}_0))^2}},$$

$$M := \max_{i \in \{1, \ldots, n\}} \|\mathbf{A}_i\|_2.$$

The idea for this proof is to define an iterate $\bar{\mathbf{X}}_t$ which represents an "average" of the local iterates $\{\mathbf{X}_{i,t}\}_{i=1}^n$ at each step $t$, and to show that (i) $\bar{\mathbf{X}}_t$ follows the ANPM dynamics of (1) and thus enjoys the convergence guarantees of Theorem 2.2, and (ii) each local iterate $\mathbf{X}_{i,t}$ stays close to $\bar{\mathbf{X}}_t$ throughout the algorithm, so that the convergence of $\bar{\mathbf{X}}_t$ implies the convergence of each $\mathbf{X}_{i,t}$. Both of these properties hold under the assumption that the number of gossip iterations $L$ per step is sufficiently large. We first recall the dynamics of the local variables $\{\mathbf{X}_{i,t}, \mathbf{Y}_{i,t}, \mathbf{R}_{i,t}\}_{i=1}^n$ given by Algorithm 2 for all $t \geqslant 1$:

$$\forall i \in \{1, \ldots, n\}, \quad \mathbf{Y}_{i,t+1/2} = \mathbf{A}_i \mathbf{X}_{i,t} - \beta \mathbf{X}_{i,t-1} \mathbf{R}_{i,t}^{-1},$$

$$\mathbf{Y}_{i,t+1} = \mathrm{AccGossip}(\{\mathbf{Y}_{j,t+1/2}\}_{j=1}^n, \mathbf{W}, L, i),$$

$$\mathbf{X}_{i,t+1}, \mathbf{R}_{i,t+1} = \mathrm{QR}(\mathbf{Y}_{i,t+1}),$$

where $\mathrm{AccGossip}(\{\mathbf{Y}_{j,t+1/2}\}_{j=1}^n, \mathbf{W}, L, i)$ refers to the output of Algorithm 1 on node $i$ after $L$ iterations, with gossip matrix $\mathbf{W}$ and initialization $\{\mathbf{Y}_{j,t+1/2}\}_{j=1}^n$. We can then define the global quantities that will be the backbone of our analysis. These quantities will typically be represented by an overline symbol:

$$\bar{\mathbf{X}}_0 := \mathbf{X}_0, \qquad \boldsymbol{\Xi}_0 := \mathbf{0}$$

$$\bar{\mathbf{Y}}_1 := \frac{1}{n} \sum_{i=1}^n \frac{1}{2} \mathbf{A}_i \mathbf{X}_{i,0} = \frac{1}{2} \mathbf{A} \bar{\mathbf{X}}_0, \qquad \bar{\mathbf{X}}_1, \bar{\mathbf{R}}_1 := \mathrm{QR}(\bar{\mathbf{Y}}_1),$$

$$\forall t \geqslant 1, \quad \left\{ \begin{array}{l} \bar{\mathbf{Y}}_{t+1} := \frac{1}{n} \sum_{i=1}^n \mathbf{Y}_{i,t+1/2} = \frac{1}{n} \sum_{i=1}^n \left(\mathbf{A}_i \mathbf{X}_{i,t} - \beta \mathbf{X}_{i,t-1} \mathbf{R}_{i,t}^{-1}\right), \\ \bar{\mathbf{X}}_{t+1}, \bar{\mathbf{R}}_{t+1} := \mathrm{QR}(\bar{\mathbf{Y}}_{t+1}), \\ \boldsymbol{\Xi}_t := \bar{\mathbf{Y}}_{t+1} - \left(\mathbf{A} \bar{\mathbf{X}}_t - \beta \bar{\mathbf{X}}_{t-1} \bar{\mathbf{R}}_t^{-1}\right). \end{array} \right. \tag{54}$$

Then, $(\bar{\mathbf{X}}_t, \bar{\mathbf{R}}_t, \bar{\mathbf{Y}}_t)$ follows the ANPM dynamics with noise $\boldsymbol{\Xi}_t$:

$$\bar{\mathbf{Y}}_1 = \frac{1}{2} \mathbf{A} \bar{\mathbf{X}}_0 + \boldsymbol{\Xi}_0, \quad \bar{\mathbf{X}}_1, \bar{\mathbf{R}}_1 = \mathrm{QR}(\bar{\mathbf{Y}}_1),$$

$$\forall t \geqslant 1, \quad \left\{ \begin{array}{l} \bar{\mathbf{Y}}_{t+1} = \mathbf{A} \bar{\mathbf{X}}_t - \beta \bar{\mathbf{X}}_{t-1} \bar{\mathbf{R}}_t^{-1} + \boldsymbol{\Xi}_t, \\ \bar{\mathbf{X}}_{t+1}, \bar{\mathbf{R}}_{t+1} := \mathrm{QR}(\bar{\mathbf{Y}}_{t+1}). \end{array} \right. \tag{55}$$

Furthermore, from Proposition 3.2, we have the following upper bound on the approximation error $\|\mathbf{Y}_{i,t} - \bar{\mathbf{Y}}_t\|_F$ after $L$ gossip iterations at each step $t$:

$$\forall i \in \{1, \ldots, n\}, \quad \|\mathbf{Y}_{i,t} - \bar{\mathbf{Y}}_t\|_F \leqslant (1 - \sqrt{\gamma_{\mathbf{W}}})^L \sqrt{n} \max_{j=1, \ldots, n} \|\mathbf{Y}_{j,t-1/2} - \bar{\mathbf{Y}}_t\|_F. \tag{56}$$

Accordingly with the notations used for the proof of Theorem 2.2 in Appendix C.2, we introduce the sequence $\{\mathbf{G}_t\}$ defined as:

$$\mathbf{G}_0 := \frac{1}{2} \mathbf{I}_k, \qquad \mathbf{G}_{t+1} = (\mathbf{I}_k - \beta \boldsymbol{\Lambda}_k^{-1} \mathbf{G}_t \boldsymbol{\Lambda}_k^{-1} + \mathbf{E}_{t+1})^{-1}, \qquad \forall t \geqslant 0,$$

$$\text{where} \quad \mathbf{E}_t := \boldsymbol{\Lambda}_k^{-1}(\mathbf{U}_k^\top \boldsymbol{\Xi}_t)(\mathbf{U}_k^\top \bar{\mathbf{X}}_t)^{-1}, \qquad \forall t \geqslant 0.$$

To prove Theorem D.1, we use the following proposition, which shows that $\boldsymbol{\Xi}_t$ satisfies the noise conditions (3) and (4) under the assumption that $L$ satisfies (53). This guarantees that $\{\mathbf{G}_t\}$ is well-defined, as shown in Proposition C.3, and will allow us to apply Theorem 2.2 to $\bar{\mathbf{X}}_t$. This proposition also shows that each local iterate $\mathbf{X}_{i,t}$ stays close to $\bar{\mathbf{X}}_t$, which will allow us to derive the convergence of each $\mathbf{X}_{i,t}$ from the convergence of $\bar{\mathbf{X}}_t$.

**Proposition D.2.** *Under the assumptions of Theorem D.1, for all $t \geqslant 1$, we have that*

$$(\mathcal{P}_t): \begin{cases} \max_{i=1,\ldots,n} \|\mathbf{Y}_{i,t} - \bar{\mathbf{Y}}_t\|_{\mathrm{F}} \leqslant c_3 \frac{\lambda_k^2 \alpha_0^5}{M} \frac{\lambda_k - 2\sqrt{\beta}}{\lambda_k} \varepsilon, \\ \max_{i=1,\ldots,n} \|\mathbf{X}_{i,t} - \bar{\mathbf{X}}_t\|_{\mathrm{F}} \leqslant \frac{c_4}{M} (\lambda_k - 2\sqrt{\beta}) \alpha_0^2 \varepsilon, \\ \|\boldsymbol{\Xi}_t\|_2 \leqslant c(\lambda_k - 2\sqrt{\beta}) \cos \theta_k (\mathbf{U}_k, \bar{\mathbf{X}}_t) \varepsilon. \end{cases}$$

*where $c := 1/32$ is the universal constant from Theorem 2.2, and $c_3 := 1/1250$ and $c_4 := 1/200$ are universal constants.*

The proof of this proposition relies on multiple technical lemmas, which relate various bounds on the norms of quantities involved in the algorithm.

**Lemma D.3.** *Let $t \geqslant 1$ and assume that*

$$\forall s \in \{1, \ldots, t-1\}, \quad \|\boldsymbol{\Xi}_s\|_2 \leqslant c(\lambda_k - 2\sqrt{\beta}) \cos \theta_k (\mathbf{U}_k, \bar{\mathbf{X}}_s) \varepsilon.$$

*Then, for all $s \in \{0, \ldots, t\}$, we have that*

$$\cos \theta_k (\mathbf{U}_k, \bar{\mathbf{X}}_s) \geqslant \alpha_0. \tag{57}$$

*Proof.* Under the hypothesis of the lemma, $\boldsymbol{\Xi}_s$ satisfies the noise conditions (3) and (4) of Theorem 2.2 for all $s \in \{0, \ldots, t-1\}$ (recall that $\boldsymbol{\Xi}_0$ is defined to be $\mathbf{0}$). Thus, we can apply Lemma C.10 to the sequence $\{\bar{\mathbf{X}}_s\}_{s=0}^t$ defined by (55), which gives for all $s \in \{0, \ldots, t\}$,

$$\tan \theta_k (\mathbf{U}_k, \bar{\mathbf{X}}_s) \leqslant (1 - \sqrt{\Delta}/2)^s \tan \theta_k (\mathbf{U}_k, \bar{\mathbf{X}}_0) + \frac{\varepsilon}{2}$$
$$\leqslant \tan \theta_k (\mathbf{U}_k, \bar{\mathbf{X}}_0) + \frac{\varepsilon}{2},$$

where $\Delta = 1 - 2\sqrt{\beta}/\lambda_k \in (0, 1)$. Then, using the fact that $\cos \theta = 1/\sqrt{1 + \tan^2 \theta}$ for all $\theta \in [0, \pi/2)$, we have for all $s \in \{0, \ldots, t\}$,

$$\cos \theta_k (\mathbf{U}_k, \bar{\mathbf{X}}_s) \geqslant \frac{1}{\sqrt{1 + \left(\tan \theta_k (\mathbf{U}_k, \bar{\mathbf{X}}_0) + \frac{\varepsilon}{2}\right)^2}} = \alpha_0,$$

which concludes the proof. $\qquad\square$

**Lemma D.4.** *Let $t \geqslant 1$ and assume that*

$$\forall s \in \{1, \ldots, t-1\}, \quad \|\boldsymbol{\Xi}_s\|_2 \leqslant c(\lambda_k - 2\sqrt{\beta}) \cos \theta_k (\mathbf{U}_k, \bar{\mathbf{X}}_s) \varepsilon.$$

*Then,*

$$\|\bar{\mathbf{Y}}_t^\dagger\|_2 = \|\bar{\mathbf{R}}_t^{-1}\|_2 \leqslant \frac{1}{\lambda_k \alpha_0} \frac{1}{1/2 - c}. \tag{58}$$

*Proof.* First, from Lemma D.3, we have that $\cos \theta_k (\mathbf{U}_k, \bar{\mathbf{X}}_{t-1}) \geqslant \alpha_0$. Furthermore, from the hypothesis of the lemma, we have that the noise condition (12) is satisfied, so that Proposition C.3 holds. In particular, we have the following bound on $\|\mathbf{G}_{t-1}\|_2$:

$$\|\mathbf{G}_{t-1}\|_2 \leqslant \frac{1}{1/2 - c}, \tag{59}$$

and the relationship (19):

$$\mathbf{U}_k^\top \bar{\mathbf{Y}}_t = \boldsymbol{\Lambda}_k \mathbf{G}_{t-1}^{-1} \mathbf{U}_k^\top \bar{\mathbf{X}}_{t-1}.$$

To prove the upper bound on $\|\bar{\mathbf{Y}}_t^\dagger\|_2$, we prove a positive lower bound on $\sigma_{\min}(\bar{\mathbf{Y}}_t)$. Because of Theorem A.5, we have
$$\sigma_{\min}(\bar{\mathbf{Y}}_t) = \sigma_{\min}\left(\begin{bmatrix} \mathbf{U}_k^\top \bar{\mathbf{Y}}_t \\ \mathbf{U}_{-k}^\top \bar{\mathbf{Y}}_t \end{bmatrix}\right) \geqslant \sigma_{\min}(\mathbf{U}_k^\top \bar{\mathbf{Y}}_t). \text{ We can then lower bound } \sigma_{\min}(\mathbf{U}_k^\top \bar{\mathbf{Y}}_t) \text{ as follows:}$$

$$\sigma_{\min}(\mathbf{U}_k^\top \bar{\mathbf{Y}}_t) \geqslant \sigma_{\min}(\mathbf{\Lambda}_k \mathbf{G}_{t-1}^{-1} \mathbf{U}_k^\top \bar{\mathbf{X}}_{t-1}) \geqslant \frac{\lambda_k \cos\theta_k(\mathbf{U}_k, \bar{\mathbf{X}}_{t-1})}{\|\mathbf{G}_{t-1}\|_2} \overset{(57),(59)}{\geqslant} \lambda_k \alpha_0(1/2 - c) > 0. \tag{60}$$

Since $\bar{\mathbf{R}}_t$ is the R-factor of $\bar{\mathbf{Y}}_t$, we immediately have that

$$\|\bar{\mathbf{Y}}_t^\dagger\|_2 = \|\bar{\mathbf{R}}_t^{-1}\|_2 \leqslant \frac{1}{\lambda_k \alpha_0} \frac{1}{1/2 - c}.$$

□

**Lemma D.5.** *Let $t \geqslant 1$ and assume that for all $s \in \{1, \ldots, t-1\}$,*

$$\|\mathbf{\Xi}_s\|_2 \leqslant c(\lambda_k - 2\sqrt{\beta})\cos\theta_k(\mathbf{U}_k, \bar{\mathbf{X}}_s)\varepsilon,$$

*and that for all $i \in \{1, \ldots, n\}$,*

$$\|\mathbf{Y}_{i,t} - \bar{\mathbf{Y}}_t\|_{\mathrm{F}} \leqslant c_3 \frac{\lambda_k^2 \alpha_0^5}{M} \frac{\lambda_k - 2\sqrt{\beta}}{\lambda_k} \varepsilon.$$

*Then, for all $i \in \{1, \ldots, n\}$,*

$$\|\mathbf{R}_{i,t}^{-1}\|_2 \leqslant \frac{1}{c_5} \frac{1}{\alpha_0 \lambda_k}, \tag{61}$$

*where $c_5 := \frac{1}{2} - c - c_3$.*

*Proof.* Let $i \in \{1, \ldots, n\}$. $\mathbf{R}_{i,t}$ is the R-factor of $\mathbf{Y}_{i,t}$. We will thus prove the lemma by lower bounding $\sigma_{\min}(\mathbf{Y}_{i,t})$. Using Theorem A.4, we have

$$\sigma_{\min}(\mathbf{Y}_{i,t}) \geqslant \sigma_{\min}(\bar{\mathbf{Y}}_t) - \|\mathbf{Y}_{i,t} - \bar{\mathbf{Y}}_t\|_2 \geqslant \sigma_{\min}(\bar{\mathbf{Y}}_t) - \|\mathbf{Y}_{i,t} - \bar{\mathbf{Y}}_t\|_{\mathrm{F}}$$

The assumptions of Lemma D.4 are satisfied, so that we can use (58) to bound $\sigma_{\min}(\bar{\mathbf{Y}}_t)$. Thus, using both (58) and the assumption on $\|\mathbf{Y}_{i,t} - \bar{\mathbf{Y}}_t\|_{\mathrm{F}}$, we have

$$\sigma_{\min}(\mathbf{Y}_{i,t}) \geqslant \lambda_k \alpha_0(1/2 - c) - c_3 \frac{\lambda_k^2 \alpha_0^5}{M} \frac{\lambda_k - 2\sqrt{\beta}}{\lambda_k} \varepsilon$$
$$\geqslant c_5 \lambda_k \alpha_0,$$

where the last inequality is due to $\alpha_0 \leqslant 1$, $\frac{\lambda_k - 2\sqrt{\beta}}{\lambda_k} \leqslant 1$, $\varepsilon \leqslant 1$, and

$$\lambda_k \leqslant \lambda_1 = \|\mathbf{A}\|_2 = \left\|\frac{1}{n}\sum_{i=1}^n \mathbf{A}_i\right\|_2 \leqslant \frac{1}{n}\sum_{i=1}^n \|\mathbf{A}_i\|_2 \leqslant M. \tag{62}$$

We thus obtain

$$\|\mathbf{R}_{i,t}^{-1}\|_2 \leqslant \frac{1}{c_5 \lambda_k \alpha_0}.$$

□

**Lemma D.6.** *Let $t \geqslant 1$ and assume that for all $s \in \{1, \ldots, t-1\}$,*

$$\|\mathbf{\Xi}_s\|_2 \leqslant c(\lambda_k - 2\sqrt{\beta})\cos\theta_k(\mathbf{U}_k, \bar{\mathbf{X}}_s)\varepsilon,$$

*and that for all $i \in \{1, \ldots, n\}$,*

$$\|\mathbf{Y}_{i,t} - \bar{\mathbf{Y}}_t\|_{\mathrm{F}} \leqslant c_3 \frac{\lambda_k^2 \alpha_0^5}{M} \frac{\lambda_k - 2\sqrt{\beta}}{\lambda_k} \varepsilon.$$

*Then, for all $i \in \{1, \ldots, n\}$, the following inequalities hold:*

$$\|\bar{\mathbf{Y}}_t^\dagger\|_2 \|\mathbf{Y}_{i,t} - \bar{\mathbf{Y}}_t\|_{\mathrm{F}} \leqslant \frac{1}{2} < 1, \tag{63}$$

$$\frac{\sqrt{2}\|\bar{\mathbf{Y}}_t^\dagger\|_2 \|\mathbf{Y}_{i,t} - \bar{\mathbf{Y}}_t\|_{\mathrm{F}}}{1 - \|\bar{\mathbf{Y}}_t^\dagger\|_2 \|\mathbf{Y}_{i,t} - \bar{\mathbf{Y}}_t\|_{\mathrm{F}}} \leqslant c_4 \frac{\lambda_k^2 \alpha_0^4}{M} \frac{\lambda_k - 2\sqrt{\beta}}{\lambda_k} \varepsilon, \tag{64}$$

$$\|\mathbf{X}_{i,t} - \bar{\mathbf{X}}_t\|_{\mathrm{F}} \leqslant c_4 \frac{\alpha_0^2}{M} (\lambda_k - 2\sqrt{\beta})\varepsilon, \tag{65}$$

$$\|\mathbf{R}_{i,t} - \bar{\mathbf{R}}_t\|_2 \leqslant c_4 c_6 (\lambda_k - 2\sqrt{\beta})\alpha_0^3 \varepsilon, \tag{66}$$

*where $c_3$ and $c_4$ are the universal constants defined in Proposition D.2, $c_5$ is the universal constant defined in Lemma D.5, and $c_6 := 1 + \frac{1}{4c_5}$.*

*Proof.* Let $i \in \{1, \ldots, n\}$. Because of the assumption on the norm of $\boldsymbol{\Xi}_s$ for all $s \in \{1, \ldots, t-1\}$, we can apply Lemma D.4 to get the bound (58) on $\|\bar{\mathbf{Y}}_t^\dagger\|_2$. Then, using the assumption on $\|\mathbf{Y}_{i,t} - \bar{\mathbf{Y}}_t\|_{\mathrm{F}}$, we have

$$\|\bar{\mathbf{Y}}_t^\dagger\|_2 \|\mathbf{Y}_{i,t} - \bar{\mathbf{Y}}_t\|_{\mathrm{F}} \leqslant \frac{1}{\lambda_k \alpha_0} \frac{1}{1/2 - c} c_3 \frac{\lambda_k^2 \alpha_0^5}{M} \frac{\lambda_k - 2\sqrt{\beta}}{\lambda_k} \varepsilon$$

$$= \frac{c_3}{(1/2 - c)} \frac{\lambda_k \alpha_0^4}{M} \frac{\lambda_k - 2\sqrt{\beta}}{\lambda_k} \varepsilon \leqslant \frac{c_3}{(1/2 - c)} \leqslant \frac{1}{2},$$

where the second-to-last inequality is because $\alpha_0 \leqslant 1$, $\frac{\lambda_k - 2\sqrt{\beta}}{\lambda_k} \leqslant 1$, $\varepsilon \leqslant 1$, and $\lambda_k \leqslant M$ from (62). This proves (63).

Then, (64) can be obtained by lower bounding the denominator by $1/2$, and upper bounding the numerator using the assumption on $\|\mathbf{Y}_{i,t} - \bar{\mathbf{Y}}_t\|_{\mathrm{F}}$ and the bound (58) on $\|\bar{\mathbf{Y}}_t^\dagger\|_2$:

$$\frac{\sqrt{2}\|\bar{\mathbf{Y}}_t^\dagger\|_2 \|\mathbf{Y}_{i,t} - \bar{\mathbf{Y}}_t\|_{\mathrm{F}}}{1 - \|\bar{\mathbf{Y}}_t^\dagger\|_2 \|\mathbf{Y}_{i,t} - \bar{\mathbf{Y}}_t\|_{\mathrm{F}}} \leqslant 2\sqrt{2}\|\bar{\mathbf{Y}}_t^\dagger\|_2 \|\mathbf{Y}_{i,t} - \bar{\mathbf{Y}}_t\|_{\mathrm{F}}$$

$$\leqslant 2\sqrt{2} \frac{1}{\lambda_k \alpha_0} \frac{1}{1/2 - c} c_3 \frac{\lambda_k^2 \alpha_0^5}{M} \frac{\lambda_k - 2\sqrt{\beta}}{\lambda_k} \varepsilon$$

$$\leqslant c_4 \frac{\lambda_k \alpha_0^4}{M} \frac{\lambda_k - 2\sqrt{\beta}}{\lambda_k} \varepsilon,$$

where the last inequality holds since $2\sqrt{2}\frac{c_3}{(1/2-c)} \leqslant c_4$.

The two last bounds of the lemma are proved using the perturbation bounds in Theorems A.2 and A.3, which are both applicable because of (63). First, from Theorem A.2, we have

$$\|\mathbf{X}_{i,t} - \bar{\mathbf{X}}_t\|_{\mathrm{F}} \leqslant \frac{\sqrt{2}\|\bar{\mathbf{Y}}_t^\dagger\|_2 \|\mathbf{Y}_{i,t} - \bar{\mathbf{Y}}_t\|_{\mathrm{F}}}{1 - \|\bar{\mathbf{Y}}_t^\dagger\|_2 \|\mathbf{Y}_{i,t} - \bar{\mathbf{Y}}_t\|_{\mathrm{F}}} \leqslant c_4 \frac{\lambda_k \alpha_0^4}{M} \frac{\lambda_k - 2\sqrt{\beta}}{\lambda_k} \varepsilon$$

$$\leqslant c_4 \frac{\alpha_0^2}{M} (\lambda_k - 2\sqrt{\beta})\varepsilon,$$

where we used in the last inequality $\alpha_0 \leqslant 1$. This proves (65). Then, from Theorem A.3, we have

$$\|\mathbf{R}_{i,t} - \bar{\mathbf{R}}_t\|_2 \leqslant \frac{\sqrt{2}\|\bar{\mathbf{Y}}_t^\dagger\|_2 \|\mathbf{Y}_{i,t} - \bar{\mathbf{Y}}_t\|_{\mathrm{F}}}{1 - \|\bar{\mathbf{Y}}_t^\dagger\|_2 \|\mathbf{Y}_{i,t} - \bar{\mathbf{Y}}_t\|_{\mathrm{F}}} \|\bar{\mathbf{R}}_t\|_2. \tag{67}$$

We thus need to upper bound $\|\bar{\mathbf{R}}_t\|_2$ to conclude. Since $\bar{\mathbf{R}}_t$ is the R-factor of $\bar{\mathbf{Y}}_t$, we have $\|\bar{\mathbf{R}}_t\|_2 = \|\bar{\mathbf{Y}}_t\|_2$. Then, from the definition of $\bar{\mathbf{Y}}_t$ and the triangle inequality, we have

$$\|\bar{\mathbf{Y}}_t\|_2 = \left\| \frac{1}{n} \sum_{i=1}^{n} \left( \mathbf{A}_i \mathbf{X}_{i,t} - \beta \mathbf{X}_{i,t-1} \mathbf{R}_{i,t}^{-1} \right) \right\|_2 \leqslant \frac{1}{n} \sum_{i=1}^{n} \left( \|\mathbf{A}_i\|_2 \|\mathbf{X}_{i,t}\|_2 + \beta \|\mathbf{X}_{i,t-1}\|_2 \|\mathbf{R}_{i,t}^{-1}\|_2 \right)$$

$$\|\bar{\mathbf{R}}_t\|_2 \leqslant M + \frac{\beta}{c_5 \lambda_k \alpha_0} \leqslant c_6 \frac{M}{\alpha_0}, \tag{68}$$

where the second-to-last inequality is due to the bound (61) on $\|\mathbf{R}_{i,t}^{-1}\|_2$ from Lemma D.5, and the last inequality is due to $\alpha_0 \leqslant 1$ and $\beta/\lambda_k \leqslant \lambda_k/4 \leqslant M/4$. Plugging (68) and (64) into (67) concludes the proof of (66):

$$\|\mathbf{R}_{i,t} - \bar{\mathbf{R}}_t\|_2 \leqslant c_4 \frac{\lambda_k \alpha_0^4}{M} \frac{\lambda_k - 2\sqrt{\beta}}{\lambda_k} \varepsilon c_6 \frac{M}{\alpha_0} = c_4 c_6 (\lambda_k - 2\sqrt{\beta}) \alpha_0^3 \varepsilon.$$

$\square$

**Lemma D.7.** *Let $t \geqslant 1$ and assume that for all $s \in \{1, \ldots, t-1\}$,*

$$\|\mathbf{\Xi}_s\|_2 \leqslant c(\lambda_k - 2\sqrt{\beta}) \cos \theta_k (\mathbf{U}_k, \bar{\mathbf{X}}_s) \varepsilon,$$

*and that for all $i \in \{1, \ldots, n\}$,*

$$\|\mathbf{Y}_{i,t} - \bar{\mathbf{Y}}_t\|_{\mathrm{F}} \leqslant c_3 \frac{\lambda_k^2 \alpha_0^5}{M} \frac{\lambda_k - 2\sqrt{\beta}}{\lambda_k} \varepsilon.$$

*Then, for all $i \in \{1, \ldots, n\}$, the following bound holds:*

$$\|\mathbf{X}_{i,t-1} \mathbf{R}_{i,t}^{-1} - \bar{\mathbf{X}}_{t-1} \bar{\mathbf{R}}_t^{-1}\|_{\mathrm{F}} \leqslant c_7 \frac{1}{\lambda_k^2} (\lambda_k - 2\sqrt{\beta}) \alpha_0 \varepsilon, \tag{69}$$

*where $c_7 := \frac{c_4}{c_5} \left( \frac{c_6}{1/2 - c} + 1 \right)$.*

*Proof.* Let $i \in \{1, \ldots, n\}$. We decompose $\mathbf{X}_{i,t-1} \mathbf{R}_{i,t}^{-1} - \bar{\mathbf{X}}_{t-1} \bar{\mathbf{R}}_t^{-1}$ into the following terms:

$$\begin{aligned}
\mathbf{X}_{i,t-1} \mathbf{R}_{i,t}^{-1} - \bar{\mathbf{X}}_{t-1} \bar{\mathbf{R}}_t^{-1} &= \mathbf{X}_{i,t-1} \mathbf{R}_{i,t}^{-1} - \bar{\mathbf{X}}_{t-1} \mathbf{R}_{i,t}^{-1} + \bar{\mathbf{X}}_{t-1} \mathbf{R}_{i,t}^{-1} - \bar{\mathbf{X}}_{t-1} \bar{\mathbf{R}}_t^{-1} \\
&= (\mathbf{X}_{i,t-1} - \bar{\mathbf{X}}_{t-1}) \mathbf{R}_{i,t}^{-1} + \bar{\mathbf{X}}_{t-1} (\mathbf{R}_{i,t}^{-1} - \bar{\mathbf{R}}_t^{-1}) \\
&= (\mathbf{X}_{i,t-1} - \bar{\mathbf{X}}_{t-1}) \mathbf{R}_{i,t}^{-1} + \bar{\mathbf{X}}_{t-1} \bar{\mathbf{R}}_t^{-1} (\bar{\mathbf{R}}_t - \mathbf{R}_{i,t}) \mathbf{R}_{i,t}^{-1}.
\end{aligned}$$

We can thus bound its norm as

$$\|\mathbf{X}_{i,t-1} \mathbf{R}_{i,t}^{-1} - \bar{\mathbf{X}}_{t-1} \bar{\mathbf{R}}_t^{-1}\|_{\mathrm{F}} \leqslant \|\mathbf{R}_{i,t}^{-1}\|_2 \|\mathbf{X}_{i,t-1} - \bar{\mathbf{X}}_{t-1}\|_{\mathrm{F}} + \|\bar{\mathbf{X}}_{t-1}\|_2 \|\bar{\mathbf{R}}_t^{-1}\|_2 \|\bar{\mathbf{R}}_t - \mathbf{R}_{i,t}\|_{\mathrm{F}} \|\mathbf{R}_{i,t}^{-1}\|_2.$$

Each of these factors have been upper bounded in previous lemmas, whose assumptions are satisfied. First, from Lemma D.5, we have the bound (61) on $\|\mathbf{R}_{i,t}^{-1}\|_2$. From Lemma D.4, we have the bound (58) on $\|\bar{\mathbf{R}}_t^{-1}\|_2$. Then, from (65) in Lemma D.6, we have the bound on $\|\mathbf{X}_{i,t-1} - \bar{\mathbf{X}}_{t-1}\|_{\mathrm{F}}$ and from (66) in Lemma D.6, we have the bound on $\|\bar{\mathbf{R}}_t - \mathbf{R}_{i,t}\|_{\mathrm{F}}$. Applying all these bounds on the above inequality yields the wanted result:

$$\begin{aligned}
\|\mathbf{X}_{i,t-1} \mathbf{R}_{i,t}^{-1} - \bar{\mathbf{X}}_{t-1} \bar{\mathbf{R}}_t^{-1}\|_{\mathrm{F}} &\leqslant \frac{1}{c_5} \frac{1}{\alpha_0 \lambda_k} c_4 \frac{\alpha_0^2}{M} (\lambda_k - 2\sqrt{\beta}) \varepsilon + \frac{1}{\lambda_k \alpha_0} \frac{1}{1/2 - c} c_4 c_6 (\lambda_k - 2\sqrt{\beta}) \alpha_0^3 \varepsilon \frac{1}{c_5} \frac{1}{\alpha_0 \lambda_k} \\
&\leqslant c_7 \frac{1}{\lambda_k^2} (\lambda_k - 2\sqrt{\beta}) \alpha_0 \varepsilon,
\end{aligned}$$

where the last inequality is due to $\lambda_k \leqslant M$.

$\square$

We can now prove Proposition D.2.

*Proof of Proposition D.2.* We will prove this result by induction. We recall the definition of the proposition $(\mathcal{P}_t)$ for all $t \geqslant 1$:

$$(\mathcal{P}_t): \begin{cases} \max_{i=1,\ldots,n} \|\mathbf{Y}_{i,t} - \bar{\mathbf{Y}}_t\|_{\mathrm{F}} \leqslant c_3 \frac{\lambda_k^2 \alpha_0^5}{M} \frac{\lambda_k - 2\sqrt{\beta}}{\lambda_k} \varepsilon, \\ \max_{i=1,\ldots,n} \|\mathbf{X}_{i,t} - \bar{\mathbf{X}}_t\|_{\mathrm{F}} \leqslant \frac{c_4}{M}(\lambda_k - 2\sqrt{\beta})\alpha_0^2 \varepsilon, \\ \|\mathbf{\Xi}_t\|_2 \leqslant c(\lambda_k - 2\sqrt{\beta}) \cos\theta_k(\mathbf{U}_k, \bar{\mathbf{X}}_t)\varepsilon. \end{cases}$$

**Base case:** We first show $(\mathcal{P}_1)$. $\{\mathbf{Y}_{i,1}\}_{i=1}^n$ is the output of Algorithm 1 after $L$ gossip iterations on the initial values $\{\frac{1}{2}\mathbf{A}_i\mathbf{X}_{i,0}\}_{i=1}^n$. Thus, from Proposition 3.2, we have that for all $i \in \{1, \ldots, n\}$,

$$\|\mathbf{Y}_{i,1} - \bar{\mathbf{Y}}_1\|_{\mathrm{F}} \leqslant (1 - \sqrt{\gamma_{\mathbf{W}}})^L \sqrt{n} \max_{i=1,\ldots,n} \left\| \frac{1}{2}\mathbf{A}_i\mathbf{X}_{i,0} - \frac{1}{2}\mathbf{A}\mathbf{X}_0 \right\|_{\mathrm{F}}$$

$$\leqslant (1 - \sqrt{\gamma_{\mathbf{W}}})^L \sqrt{n} \max_{i=1,\ldots,n} \frac{1}{2} \left( \underbrace{\|\mathbf{A}_i\|_2}_{\leqslant M} \underbrace{\|\mathbf{X}_{i,0}\|_{\mathrm{F}}}_{=\sqrt{k}} + \underbrace{\|\mathbf{A}\|_2}_{\leqslant n^{-1}\sum_i \|\mathbf{A}_i\|_2 \leqslant M} \underbrace{\|\mathbf{X}_0\|_{\mathrm{F}}}_{=\sqrt{k}} \right)$$

$$\leqslant (1 - \sqrt{\gamma_{\mathbf{W}}})^L \sqrt{nk}M.$$

From the assumption on the number of gossip iterations $L$ in (53), since $c_1 \geqslant 5$, $c_2^5 \geqslant 1/c_3$ and $\frac{1}{c_3}\sqrt{nk}\frac{\lambda_k}{\lambda_k - 2\sqrt{\beta}}\frac{1}{\varepsilon} \geqslant 1$, we have

$$L \geqslant \frac{1}{-\log(1 - \sqrt{\gamma_{\mathbf{W}}})} \log\left( \frac{1}{c_3}\sqrt{nk}\frac{M^2}{\lambda_k^2}\frac{\lambda_k}{\lambda_k - 2\sqrt{\beta}}\frac{1}{\alpha_0^5}\frac{1}{\varepsilon} \right)$$

so that

$$\|\mathbf{Y}_{i,1} - \bar{\mathbf{Y}}_1\|_{\mathrm{F}} \leqslant c_3 \frac{\lambda_k^2 \alpha_0^5}{M} \frac{\lambda_k - 2\sqrt{\beta}}{\lambda_k} \varepsilon. \tag{70}$$

This proves the first point of $(\mathcal{P}_1)$. The second point of is the bound (65) from Lemma D.6, whose assumptions are satisfied because of the bound on $\|\mathbf{Y}_{i,1} - \bar{\mathbf{Y}}_1\|_{\mathrm{F}}$ in (70).

Finally, we control the norm of $\mathbf{\Xi}_1$ to prove the last point of $(\mathcal{P}_1)$. From the definition of $\mathbf{\Xi}_1$ in (54), we have

$$\mathbf{\Xi}_1 = \bar{\mathbf{Y}}_2 - \left(\mathbf{A}\bar{\mathbf{X}}_1 - \beta\bar{\mathbf{X}}_0\bar{\mathbf{R}}_1^{-1}\right) = \frac{1}{n}\sum_{i=1}^n \left( \left(\mathbf{A}_i\mathbf{X}_{i,1} - \beta\mathbf{X}_{i,0}\mathbf{R}_{i,1}^{-1}\right) - \left(\mathbf{A}_i\bar{\mathbf{X}}_1 - \beta\bar{\mathbf{X}}_0\bar{\mathbf{R}}_1^{-1}\right) \right),$$

$$\|\mathbf{\Xi}_1\|_2 \leqslant \|\mathbf{\Xi}_1\|_{\mathrm{F}} \leqslant \frac{1}{n}\sum_{i=1}^n \|\mathbf{A}_i\|_2\|\mathbf{X}_{i,1} - \bar{\mathbf{X}}_1\|_{\mathrm{F}} + \frac{\beta}{n}\sum_{i=1}^n \|\mathbf{X}_{i,0}\mathbf{R}_{i,1}^{-1} - \bar{\mathbf{X}}_0\bar{\mathbf{R}}_1^{-1}\|_{\mathrm{F}}. \tag{71}$$

$\|\mathbf{X}_{i,1} - \bar{\mathbf{X}}_1\|_{\mathrm{F}}$ is upper-bounded in (65) from Lemma D.6, and $\|\mathbf{X}_{i,0}\mathbf{R}_{i,1}^{-1} - \bar{\mathbf{X}}_0\bar{\mathbf{R}}_1^{-1}\|_{\mathrm{F}}$ is upper-bounded in (69) from Lemma D.7. Both of these lemmas have their assumptions satisfied because of (70). Plugging these two bounds into (71) yields

$$\|\mathbf{\Xi}_1\|_2 \leqslant M\frac{c_4}{M}(\lambda_k - 2\sqrt{\beta})\alpha_0^2\varepsilon + \beta c_7 \frac{1}{\lambda_k^2}(\lambda_k - 2\sqrt{\beta})\alpha_0\varepsilon$$

$$\leqslant c(\lambda_k - 2\sqrt{\beta})\alpha_0\varepsilon,$$

where the last inequality is due to $\beta < \lambda_k^2/4$ and $c_4 + c_7/4 \leqslant c$. This concludes the proof of $(\mathcal{P}_1)$.

**Induction:** Let $t \geqslant 2$ and assume that $(\mathcal{P}_s)$ holds for all $s \in \{1, \ldots, t-1\}$. We will show that $(\mathcal{P}_t)$ also holds. First, from the induction hypothesis, we have that for all $s \in \{1, \ldots, t-1\}$,

$$\|\mathbf{\Xi}_s\|_2 \leqslant c(\lambda_k - 2\sqrt{\beta}) \cos\theta_k(\mathbf{U}_k, \bar{\mathbf{X}}_s)\varepsilon.$$

We can prove the upper-bound on $\|\mathbf{Y}_{i,t} - \bar{\mathbf{Y}}_t\|_F$ in the first point of $(\mathcal{P}_t)$ using a similar reasoning as in the base case. Since $\{\mathbf{Y}_{i,t}\}_{i=1}^n$ is the output of Algorithm 1 after $L$ gossip iterations on the initial values $\{\mathbf{A}_i\mathbf{X}_{i,t-1} - \beta\mathbf{X}_{i,t-2}\mathbf{R}_{i,t-1}^{-1}\}_{i=1}^n$, we have from Proposition 3.2 that for all $i \in \{1,\dots,n\}$,

$$\|\mathbf{Y}_{i,t} - \bar{\mathbf{Y}}_t\|_F \leqslant (1-\sqrt{\gamma\mathbf{W}})^L \sqrt{n} \max_{j=1,\dots,n} \left\| \mathbf{A}_i\mathbf{X}_{j,t-1} - \beta\mathbf{X}_{j,t-2}\mathbf{R}_{j,t-1}^{-1} - \frac{1}{n}\sum_{m=1}^n \left(\mathbf{A}_m\mathbf{X}_{m,t-1} - \beta\mathbf{X}_{m,t-2}\mathbf{R}_{m,t-1}^{-1}\right) \right\|_F$$

$$\leqslant (1-\sqrt{\gamma\mathbf{W}})^L \sqrt{n} \max_{j=1,\dots,n} \left( \|\mathbf{A}_i\|_2\|\mathbf{X}_{j,t-1}\|_F + \beta\|\mathbf{X}_{j,t-2}\|_F\|\mathbf{R}_{j,t-1}^{-1}\|_2 + \right.$$

$$\left. \frac{1}{n}\sum_{m=1}^n \left(\|\mathbf{A}_m\|_2\|\mathbf{X}_{m,t-1}\|_F + \beta\|\mathbf{X}_{m,t-2}\|_F\|\mathbf{R}_{m,t-1}^{-1}\|_2\right) \right)$$

$$\leqslant (1-\sqrt{\gamma\mathbf{W}})^L \sqrt{nk} \left(2M + 2\beta\frac{1}{c_5\lambda_k\alpha_0}\right)$$

$$\leqslant (1-\sqrt{\gamma\mathbf{W}})^L \sqrt{nk}\frac{c_8 M}{\alpha_0},$$

where the second-to-last inequality is due to the bounds (61) on $\|\mathbf{R}_{m,t-1}^{-1}\|_2$ in Lemma D.5, whose assumptions are verified because of the induction hypothesis, and where the last inequality is due to $\beta/\lambda_k < \lambda_k/4 \leqslant M/4$ and $c_8 := 2 + \frac{1}{2c_5}$. Using the assumption on the number of gossip iterations $L$ in (53), since $c_1 = 6$, $\lambda_k/(\lambda_k - 2\sqrt{\beta}) \geqslant 1$, $M/\lambda_k \geqslant 1$, $\alpha_0 \leqslant 1$, $\varepsilon \leqslant 1$, $\sqrt{nk} \geqslant 1$ and $c_2^6 \geqslant c_8/c_3$, we have

$$L \geqslant \frac{1}{-\log(1-\sqrt{\gamma\mathbf{W}})}\log\left(\frac{c_8}{c_3}\sqrt{nk}\frac{M^2}{\lambda_k^2}\frac{\lambda_k}{\lambda_k - 2\sqrt{\beta}}\frac{1}{\alpha_0^6}\frac{1}{\varepsilon}\right),$$

so that we can upper bound $(1-\sqrt{\gamma\mathbf{W}})^L$ and obtain the wanted bound on $\|\mathbf{Y}_{i,t} - \bar{\mathbf{Y}}_t\|_F$:

$$\|\mathbf{Y}_{i,t} - \bar{\mathbf{Y}}_t\|_F \leqslant c_3\frac{\lambda_k^2\alpha_0^5}{M}\frac{\lambda_k - 2\sqrt{\beta}}{\lambda_k}\varepsilon.$$

Then, just like in the base case, the second point of $(\mathcal{P}_t)$ is (65) from Lemma D.6, whose assumptions are satisfied because of the bound we just proved on $\|\mathbf{Y}_{i,t} - \bar{\mathbf{Y}}_t\|_F$ and because of the bounds on $\boldsymbol{\Xi}_s$ for $s \in \{1,\dots,t-1\}$ from the induction hypothesis.

Finally, we control the norm of $\boldsymbol{\Xi}_t$ to prove the last point of $(\mathcal{P}_t)$. We proceed similarly as in the base case: from the definition of $\boldsymbol{\Xi}_t$ in (54), we have

$$\boldsymbol{\Xi}_t = \bar{\mathbf{Y}}_{t+1} - \left(\mathbf{A}\bar{\mathbf{X}}_t - \beta\bar{\mathbf{X}}_{t-1}\bar{\mathbf{R}}_t^{-1}\right) = \frac{1}{n}\sum_{i=1}^n \left(\left(\mathbf{A}_i\mathbf{X}_{i,t} - \beta\mathbf{X}_{i,t-1}\mathbf{R}_{i,t}^{-1}\right) - \left(\mathbf{A}_i\bar{\mathbf{X}}_t - \beta\bar{\mathbf{X}}_{t-1}\bar{\mathbf{R}}_t^{-1}\right)\right),$$

$$\|\boldsymbol{\Xi}_t\|_2 \leqslant \|\boldsymbol{\Xi}_t\|_F \leqslant \frac{1}{n}\sum_{i=1}^n \|\mathbf{A}_i\|_2\|\mathbf{X}_{i,t} - \bar{\mathbf{X}}_t\|_F + \frac{\beta}{n}\sum_{i=1}^n \|\mathbf{X}_{i,t-1}\mathbf{R}_{i,t}^{-1} - \bar{\mathbf{X}}_{t-1}\bar{\mathbf{R}}_t^{-1}\|_F.$$

Then, using the bounds (65) on $\|\mathbf{X}_{i,t} - \bar{\mathbf{X}}_t\|_F$ from Lemma D.6 and (69) on $\|\mathbf{X}_{i,t-1}\mathbf{R}_{i,t}^{-1} - \bar{\mathbf{X}}_{t-1}\bar{\mathbf{R}}_t^{-1}\|_F$ from Lemma D.7, whose assumptions are satisfied because of the bound we just proved on $\|\mathbf{Y}_{i,t} - \bar{\mathbf{Y}}_t\|_F$ and because of the bounds on $\boldsymbol{\Xi}_s$ for $s \in \{1,\dots,t-1\}$ from the induction hypothesis, we have

$$\|\boldsymbol{\Xi}_t\|_2 \leqslant M\frac{c_4}{M}(\lambda_k - 2\sqrt{\beta})\alpha_0^2\varepsilon + \beta c_7\frac{1}{\lambda_k^2}(\lambda_k - 2\sqrt{\beta})\alpha_0\varepsilon$$

$$\leqslant c(\lambda_k - 2\sqrt{\beta})\alpha_0\varepsilon,$$

where the last inequality is due to $\beta < \lambda_k^2/4$ and $c_4 + c_7/4 \leqslant c$. This concludes the proof of $(\mathcal{P}_t)$ and thus the induction.

$\square$

We can now prove Theorem D.1.

*Proof of Theorem D.1.* According to Proposition D.2, we have that for all $t \geqslant 0$,

$$\|\mathbf{\Xi}_t\|_2 \leqslant c(\lambda_k - 2\sqrt{\beta})\cos\theta_k(\mathbf{U}_k, \bar{\mathbf{X}}_t)\varepsilon.$$

As such, the ANPM dynamics (55) followed by $\{\bar{\mathbf{X}}_t\}_{t\geqslant 0}$ satisfy the noise conditions in Theorem 2.2. Thus, for all $t \geqslant T$, we have $\sin\theta_k(\mathbf{U}_k, \bar{\mathbf{X}}_t) \leqslant \varepsilon$, where $T$ is such that

$$T = \mathcal{O}\left(\sqrt{\frac{\lambda_k}{\lambda_k - 2\sqrt{\beta}}}\log\left(\frac{\tan\theta_k(\mathbf{U}_k, \mathbf{X}_0)}{\varepsilon}\right)\right).$$

Furthermore, according to Proposition D.2, we have that for all $t \geqslant T$, and for all $i \in \{1, \ldots, n\}$,

$$\|\mathbf{X}_{i,t} - \bar{\mathbf{X}}_t\|_{\mathrm{F}} \leqslant c_4\frac{1}{M}(\lambda_k - 2\sqrt{\beta})\alpha_0^2\varepsilon \leqslant \frac{\lambda_k - 2\sqrt{\beta}}{\lambda_k}\varepsilon$$
$$\|\mathbf{X}_{i,t} - \bar{\mathbf{X}}_t\|_2 \leqslant \varepsilon,$$

where the second inequality is from $c_4 \leqslant 1$, $\lambda_k \leqslant M$ and $\alpha_0 \leqslant 1$. Then, using Proposition A.1, we have for all $t \geqslant T$,

$$\sin\theta_k(\mathbf{U}_k, \mathbf{X}_{i,t}) = \|\mathbf{U}_{-k}^\top\mathbf{X}_{i,t}\|_2 \leqslant \|\mathbf{U}_{-k}^\top\bar{\mathbf{X}}_t\|_2 + \|\mathbf{U}_{-k}^\top(\mathbf{X}_{i,t} - \bar{\mathbf{X}}_t)\|_2$$
$$= \sin\theta_k(\mathbf{U}_k, \bar{\mathbf{X}}_t) + \|\mathbf{X}_{i,t} - \bar{\mathbf{X}}_t\|_2 \leqslant 2\varepsilon.$$

This concludes the proof. □

## E. Experimental Details

### E.1. Experimental Details for ANPM

For all of the experiments shown in Figure 1, we generate a single orthogonal basis of eigenvectors $\mathbf{U} \in \mathrm{St}(d, d)$ (by taking the Q-factor of a $d \times d$ matrix with i.i.d. standard normal entries). The matrix $\mathbf{A}$ is then constructed as

$$\mathbf{A} = \mathbf{U}\mathrm{diag}(\lambda_1, \ldots \lambda_1, \lambda_k, \lambda_{k+1}, \lambda_d, \ldots, \lambda_d)\mathbf{U}^\top,$$

where $\lambda_1 = \lambda_k = 1$, $\lambda_d = 0.5$ and $\lambda_{k+1}$ is varied to obtain different eigengaps $\Delta_k = 1 - \lambda_{k+1}$. In our experiments, we set $d = 1000$ and $k = 10$. $\mathbf{X}_0$ is generated as the Q-factor of a $d \times k$ matrix with i.i.d. standard normal entries.

Note that the adaptive tuning heuristic $\beta_t$ requires to run ANPM on $k + 1$ columns instead of $k$. In that case, for each plot, the initialization $\mathbf{X}_0$ is chosen so that its first $k$ columns are the same as the initialization used for the other methods. Furthermore, the quantity plotted corresponds to $\sin$ of the $k$-th principal angle between $\mathbf{U}_k$ and the $k$ first columns of $\mathbf{X}_t$, so that the comparison with other methods is fair. Indeed, from (Balcan et al., 2016; Xu, 2023), we expect the convergence of ANPM to the top-$k$ eigenspace to improve when running it on more than $k$ columns.

We sample the noise matrices $\mathbf{\Xi}_t$ from a so-called "adversarial" distribution, inspired by the adversarial examples used in the proofs in Appendix C.4. The adversarial examples from Appendix C.4 were designed to be in a single direction corresponding to the eigenvectors of $\mathbf{A}$, hindering as much as possible the convergence of NPM/ANPM. In our experiments, given a noise norm $\xi$, we sample $\mathbf{\Xi}_t$ as

$$\mathbf{\Xi}_t = -\xi\frac{\mathbf{U}\mathbf{V}}{\|\mathbf{V}\|_2}, \qquad \mathbf{V} = [|v_{i,j}|] \in \mathbb{R}^{d\times k}, \qquad v_{i,j} \overset{\text{i.i.d}}{\sim} \mathcal{N}(0, 1).$$

This ensures that 1) $\mathbf{\Xi}_t$ is of spectral norm $\xi$ and 2) all the columns $\boldsymbol{\xi}_i$ of $\mathbf{\Xi}_t$ verify $\langle\boldsymbol{\xi}_i, \boldsymbol{u}_j\rangle \leqslant 0$ for all $j \in \{1, \ldots, d\}$, which tends to hinder the convergence of ANPM in a similar way as the adversarial examples from Appendix C.4.

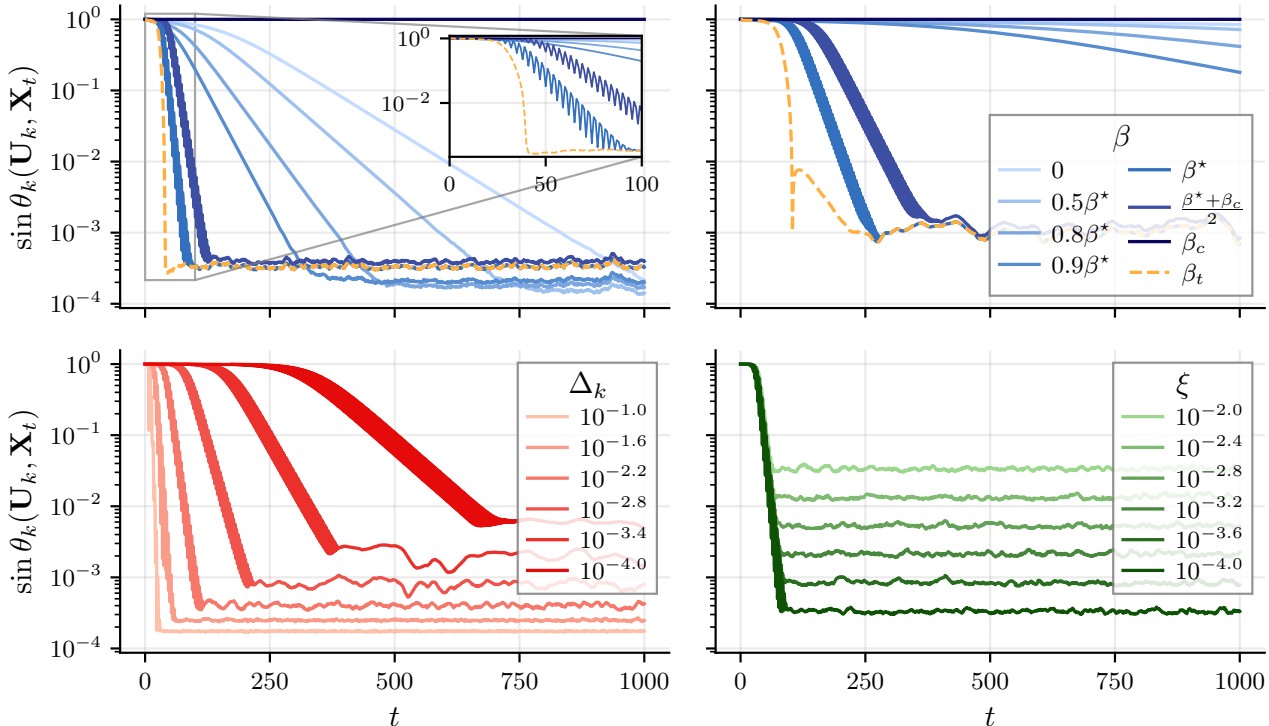

*Figure 3.* Experimental results for (A)NPM with stochastic mean-centered noise. From left to right: fixed noise norm and large gap, varying momentum; fixed noise norm and small gap, varying momentum; optimal momentum and fixed noise norm, varying gap; optimal momentum and fixed gap, varying noise norm.

## E.2. Additional experiments for ANPM

### E.2.1. ANPM WITH MEAN-CENTERED NOISE

In the same setting as described in the previous section, we perform experiments where $\boldsymbol{\Xi}_t$ is sampled from a centered distribution. We argue that mean-centered noise models behave differently from the adversarial noise used in Appendix E.1. Indeed, we show in Figure 3 the results of the same experiments as in Figure 1, but where $\boldsymbol{\Xi}_t$ is sampled uniformly on the sphere of radius $\xi$ for the spectral norm:

$$\boldsymbol{\Xi}_t = \xi \frac{\mathbf{V}}{\|\mathbf{V}\|_2}, \qquad \mathbf{V} = [v_{i,j}] \in \mathbb{R}^{d \times k}, \qquad v_{i,j} \overset{\text{i.i.d}}{\sim} \mathcal{N}(0,1).$$

We observe notably that 1) ANPM's evolution is still separated into a transient exponentially decaying regime followed by a stationary regime, 2) the level of the stationary regime (i.e. the final precision reached) seems to depend on the momentum parameter $\beta$ (which was not the case with adversarial noise), and 3) the level of the stationary regime is not proportional to the gap $\Delta_k$. These two last points suggest that the behavior of ANPM with mean-centered noise is different from the one with adversarial noise.

### E.2.2. LARGE-SCALE EXPERIMENTS FOR ANPM WITH GAUSSIAN NOISE

We show in the next experiment that ANPM can be used for very large-scale matrices with favorable structure. We perform spectral clustering on the Amazon0302 graph dataset (Leskovec et al., 2007), which consists of $d = 262111$ nodes and $s = 1234877$ edges. The resulting matrix is very large, of size $d \times d$, but sparse with $s + d \ll d^2$ non-zero entries. This allows for efficient matrix-vector products even though storing in memory the matrix in dense format would be impossible for many devices. We consider i.i.d centered Gaussian noise $[\boldsymbol{\Xi}_t]_{i,j} \overset{\text{i.i.d}}{\sim} \mathcal{N}(0, \sigma^2)$ for two different values of $\sigma$, and compare ANPM using the tuning heuristic $\beta_t$ described in (5), and non-accelerated NPM. We let $k = 30$, and show the results in terms of reconstruction error $\|(\mathbf{I}_d - \mathbf{X}_t \mathbf{X}_t^\top)\mathbf{A}\|_2$ rather than using the approximation error $\sin \theta_k(\mathbf{U}_k, \mathbf{X}_t)$, since we do not

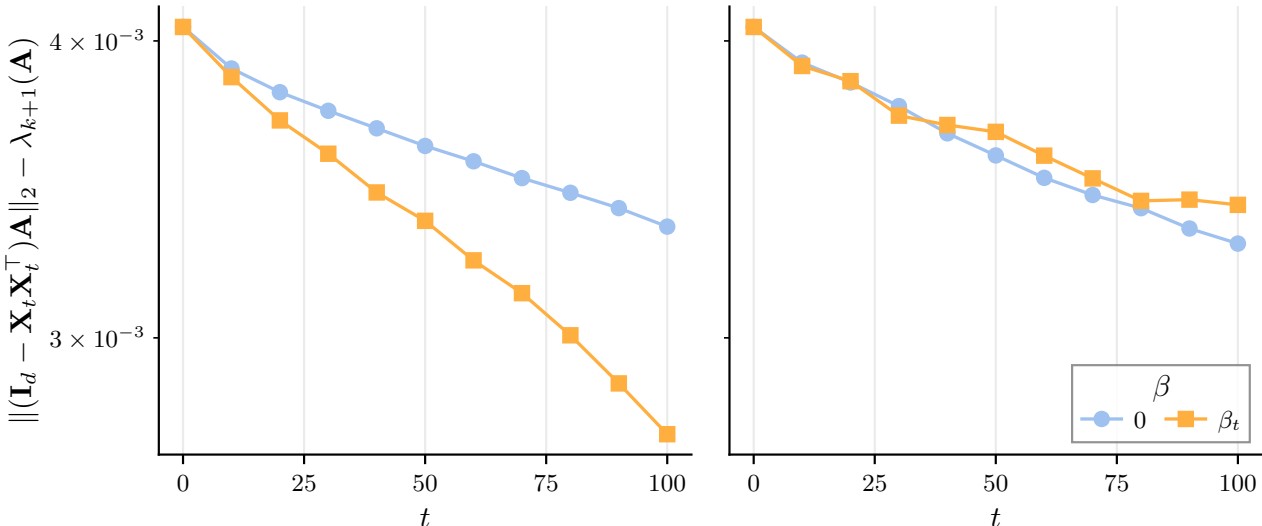

*Figure 4.* Experimental results for ANPM with Gaussian noise on the Amazon0302 dataset. **(Left)** $\sigma = 10^{-3}$. **(Right)** $\sigma = 2 \times 10^{-3}$.

have access with high precision to $\mathbf{U}_k$. The results are given in Figure 4. We observe that for smaller noises, ANPM with the tuning heuristic $\beta_t$ converges faster than NPM, while for larger noises, ANPM performs similarly to NPM.

### E.3. Experimental Details for ADePM

#### E.3.1. DETAILS FOR FED-HEART-DISEASE

Fed-Heart-Disease is a dataset from the FLamby collection (Ogier du Terrail et al., 2022), which consists of tabular data of dimension $d = 13$ partitioned into $n = 4$ hospitals. We perform decentralized PCA on rescaled local covariance matrices. Letting $\boldsymbol{\Phi}_i \in \mathbb{R}^{m_i \times d}$ be the local data matrix of agent $i$ for all $i = 1, \ldots, 4$, we let

$$\mathbf{A}_i := \frac{n}{m} \boldsymbol{\Phi}_i^\top \boldsymbol{\Phi}_i, \qquad \mathbf{A} := \frac{1}{n} \sum_{i=1}^n \mathbf{A}_i = \frac{1}{m} \sum_{i=1}^n \boldsymbol{\Phi}_i^\top \boldsymbol{\Phi}_i,$$

where $m = \sum_{i=1}^n m_i = 486$ is the total number of samples. This way, $\mathbf{A}$ is the empirical covariance matrix of the full dataset. We choose a ring graph topology for our communication network, with a gossip matrix $\mathbf{W}$ defined as

$$\mathbf{W} := \begin{bmatrix} 1/2 & 1/4 & 0 & 1/4 \\ 1/4 & 1/2 & 1/4 & 0 \\ 0 & 1/4 & 1/2 & 1/4 \\ 1/4 & 0 & 1/4 & 1/2 \end{bmatrix}.$$

#### E.3.2. DETAILS FOR EGO-FACEBOOK

Ego-Facebook (Leskovec & Mcauley, 2012) is a graph dataset consisting of 4039 Facebook users, connected by edges representing friendships. We pick a subset of size $n = d = 50$ of this graph by selecting the nodes labeled 0 through 49. This subset constitutes a connected graph $G$. The goal of spectral clustering is to find a low-dimensional embedding of the nodes of $G$ by using the bottom eigenvectors of its normalized Laplacian matrix, defined as

$$\mathbf{L}_{\mathrm{norm}} = \mathbf{I}_n - \mathbf{D}^{-1/2} \mathbf{S} \mathbf{D}^{-1/2},$$

where $\mathbf{S}$ is the adjacency matrix of $G$, such that $s_{i,j} = s_{j,i} = 1$ if there is an edge between nodes $i$ and $j$ and 0 otherwise, and where $\mathbf{D}$ is the diagonal degree matrix of $G$, such that $d_{i,i} = \sum_{j=1}^n s_{i,j}$. This matrix has eigenvalues in $[0, 2]$, with 0 being the smallest eigenvalue. Thus, finding the bottom $k$ eigenvectors of $\mathbf{L}_{\mathrm{norm}}$ is equivalent to finding the top $k$ eigenvectors of the PSD matrix

$$\mathbf{A} := 2\mathbf{I}_n - \mathbf{L}_{\mathrm{norm}} = \mathbf{I}_n + \mathbf{D}^{-1/2} \mathbf{S} \mathbf{D}^{-1/2} \succeq \mathbf{0}.$$

The agents $i \in \{1, \dots, n\}$ can construct matrices $\mathbf{A}_i$ such that $\mathbf{A} = n^{-1} \sum_{i=1}^{n} \mathbf{A}_i$ by using only knowledge of their neighbors and their degrees. Letting $d_i$ be the degree of node $i$, we let $\mathbf{A}_i$ be the matrix such that for all neighbor $j$ of $i$ in $G$,

$$[\mathbf{A}_i]_{i,i} = 1, \quad [\mathbf{A}_i]_{i,j} = [\mathbf{A}_i]_{j,i} = \frac{1}{2\sqrt{d_i d_j}},$$

and all other entries of $\mathbf{A}_i$ are 0. We can then use the decentralized PCA algorithms on the family of matrices $\{\mathbf{A}_i\}_{i=1}^{n}$ to find the bottom $k$ eigenvectors of $\mathbf{L}_{\mathrm{norm}}$.

To reduce the communication costs of our experiment, we suppose that the communication network $G'$ is given by the graph $G$, to which we add $\sim n \log(n)$ edges uniformly at random among the pairs of nodes that are not already connected in $G$, in order to increase the connectivity of the graph. We then define the gossip matrix $\mathbf{W}$ with the Metropolis-Hastings weights on the communication graph $G'$: letting $\mathcal{N}_i$ be the set of neighbors of node $i$ in $G'$, and $d'_i$ its degree in $G'$, we let

$$w_{i,j} := \begin{cases} \frac{1}{1+\max(d'_i, d'_j)} & \text{if } j \in \mathcal{N}_i, \\ 1 - \sum_{s \in \mathcal{N}_i} w_{i,s} & \text{if } i = j, \\ 0 & \text{otherwise.} \end{cases}$$

### E.3.3. DETAILS FOR DIGITS

The digits dataset (Alpaydin & Kaynak, 1998) is a dataset of 1797 grayscale $8 \times 8$ images of handwritten digits. We consider two different ways of splitting the dataset between $n = 10$ agents. In the homogeneous split, the data is distributed uniformly at random among the agents, so that the agents have similar local covariance matrices. In the heterogeneous split, the data is distributed between the agents according to their labels, so that each agent has data of a single digit and thus very different local covariance matrices. In both cases, we perform decentralized PCA on rescaled local covariance matrices, as in the case of Fed-Heart-Disease. The communication network is a ring graph of size 10, with a similar gossip matrix $\mathbf{W}$ as the one defined for Fed-Heart-Disease (i.e. a circulant matrix with coefficients $(1/2, 1/4, 0, \dots, 0, 1/4)$).

### E.3.4. ADAPTING THE TUNING HEURISTIC (5) TO ADePM

We adapt the tuning heuristic (5) for ANPM to ADePM as follows. Recall that the tuning heuristic for ANPM is, given an iterate $\mathbf{X}_t$ with $k+1$ columns,

$$\beta_t = \min_{j=1,\dots,k+1} \left[ \mathbf{X}_t^\top (\mathbf{A}\mathbf{X}_t + \mathbf{\Xi}_t) \right]_{j,j}^2 / 4.$$

For ADePM, each agent $i$ approximates $\beta_t$ locally as

$$\beta_{i,t} = \min_{j=1,\dots,k+1} \left[ \mathbf{X}_{i,t}^\top (\mathbf{Y}_{i,t-1} + \beta_{i,t-1} \mathbf{X}_{i,t-2} \mathbf{R}_{i,t-1}^{-1}) \right]_{j,j}^2 / 4,$$

since

$$\mathbf{Y}_{i,t-1} + \beta_{i,t-1} \mathbf{X}_{i,t-2} \mathbf{R}_{i,t-1}^{-1} = \mathbf{A}\bar{\mathbf{X}}_{t-1} + (-\beta_{i,t-1} \bar{\mathbf{X}}_{t-2} \bar{\mathbf{R}}_{t-1}^{-1} + \beta_{i,t-1} \mathbf{X}_{i,t-2} \mathbf{R}_{i,t-1}^{-1}) + \mathbf{\Xi}_t + (\mathbf{Y}_{i,t-1} - \bar{\mathbf{Y}}_{t-1})$$

is a good approximation of $\mathbf{A}\mathbf{X}_{i,t-1}$ for large gossip communications $L$. Here, we reused the notations from Appendix D. This method of choosing $\beta_{i,t}$ does not require any additional communication between the agents compared to vanilla ADePM.

