# OpenReview forum: "Improved Analysis of the Accelerated Noisy Power Method with Applications to Decentralized PCA"
_ICML.cc/2026/Conference — ICML 2026 regular_

### Official Review · Reviewer_YFyf · 2026-03-09

**Soundness:** 3
**Presentation:** 3
**Significance:** 3
**Originality:** 3
**Overall Recommendation:** 5
**Confidence:** 4

**Summary:**

This paper studies the accelerated power method under the noisy setting for the PCA problem. The limitation in current literature is that the noise is required to scales as $\epsilon^{\mu_k}$ where $\mu_k$ is some quantity related to the eigengap, and $\epsilon$ the target acccuracy. Based on the Accelerated Noisy Power Method (ANPM), the paper uses a novel technique that tracks the evolution of some coupling matrix that shows the interaction between the top-$k$ subspace and its orthogonal space. Via this approach, the paper shows the noise dependency that scale as $\epsilon$. Moreover, the paper provided a lower bound for ANPM that no relaxed noise condition and better time complexity can be achieved. The paper also included extension to decentralized PCA, and experimental result that validates the analysis.

**Compliance With Llm Reviewing Policy:**

Affirmed.

**Final Justification:**

The rebuttal addressed my main concern, which made me happy to raise the score.

**Key Questions For Authors:**

My main question is about how the plateau of the curve observed in experiment can be explained by the theory in the paper.

**Limitations:**

yes

**Strengths And Weaknesses:**

**Strength**
1. The improved condition on the noise scale is quite significant. In particular, the exponent dependency on the eigengap is removed.
2. The paper provided comprehensive lower bounds on the ANPM algorithm, justifying the optimality of the proved rates.
3. The paper provided extension to the decentralized PCA problem.

**Weaknesses**
1. The provided lower bound is only for the ANPM algorithm, thus only justifying the optimal rate for the ANPM algorithm, but gives no information on the hardness of the problem itself.
2. The paper assumes no structural information about the noise/perturbation. While this is general and can apply to any settings, assuming practical structures can give more information to the analysis and derive results that are closer to what we see in practice (e.g. the noise assumption that depend on the projection onto different subspace is a strong requirement, though inherited from previous work). The paper need to discuss whether extension to the case where the noise is, e.g. random and have zero-mean, as in the case of stochastic PCA, can potentially improve the rate.
3. In the experimental result, it seems that the accelerated algorithms plateaus, which does not happen in the non-acceleraetd case. The paper should justify why this happens.

---

> ### Author Rebuttal · Authors · 2026-03-31
>
> We thank the reviewer for the constructive comments and positive feedback on our paper. We address the reviewer's questions as follows:
>
> ## Plateaus in the experiments
>
> We first note that the plateaus that are observed in the experiments are not related to acceleration, and are not an inherent weakness of our acceleration method. In fact, in the two leftmost plots of Figure 1, the lines in dark blue corresponding to $\beta = 0$ (non-accelerated noisy power method (NPM)) plateau just like the ones corresponding to $\beta > 0$ (accelerated noisy power method (ANPM)). In Figure 2, DePM (dark purple, non-accelerated algorithm) plateaus, just like the lines corresponding to our accelerated algorithm ADePM.
>
> These plateaus were predicted by our theoretical results in the case of ANPM and those of [1] for the non-accelerated NPM. In both cases, we have upper bounds of form $\sin \theta_k(\mathbf{U}\_k,\mathbf{X}\_t) \leq O(\varepsilon) + h_0\gamma^t$. The term $O(\varepsilon)$ is due to the noise $\mathbf{U}\_{-k}^\top\boldsymbol{\Xi}\_t$. The transient (decreasing) regime that is observed in the first iterations for all methods corresponds to the geometric decay of the $h_0\gamma^t$ term, while the plateau in the latter iterations corresponds to the constant $O(\varepsilon)$ term. Generally, for ANPM and NPM, one cannot expect to reach an accuracy lower than $O(\varepsilon)$ if $||\mathbf{U}\_{-k}^\top\boldsymbol{\Xi}\_t||\_2$ is itself of order $O(\varepsilon)$.
>
> DeEPCA, the algorithm that does not plateau in Figure 2, does not suffer from noise of constant order ${O}(\varepsilon)$, while DePM and ADePM do. This is because DeEPCA uses a subspace tracking technique, that allows the norm of the error $\mathbf{U}_{-k}^\top\boldsymbol{\Xi}_t$ to decay geometrically instead of staying constant, ensuring that the method converges to zero error. This subspace tracking technique is a matter of communication, and is orthogonal to our acceleration scheme. We believe that an alternate version of ADePM could be made, using this same subspace tracking technique, that would converge to zero just like DeEPCA, but at an accelerated rate.
>
> ## Hardness of PCA with noisy matrix-vector products
>
> For any method that only has access at each iteration $t$ to $\mathbf{A}\mathbf{X}\_t + \boldsymbol{\Xi}\_t$ , there are instances where $||\mathbf{U}\_k^\top\boldsymbol{\Xi}\_t||\_2 \leq \lambda_k - \lambda_{k+1}$ such that estimating $\mathbf{U}\_k$ is impossible. Indeed, let $\mathbf{A} = \mathrm{diag}(\lambda_k, \dots, \lambda_k, \lambda_{k+1},\dots,\lambda_{k+1})$ and $\mathbf{B} = \lambda_{k+1} \mathbf{I}\_d$. Now, suppose that we only have access to matrix-vector products with $\mathbf{B}$. Then, this corresponds to having only access to $\mathbf{A}\mathbf{X}\_t + \boldsymbol{\Xi}\_t$, where $\boldsymbol{\Xi}\_t = (\mathbf{B} - \mathbf{A})\mathbf{X}\_t$ and we have $||\mathbf{U}\_k^\top\boldsymbol{\Xi}\_t||\_2 \leq \lambda_k - \lambda_{k+1}$. However, since $\mathbf{B}$ is proportional to the identity matrix, it holds no information, and no algorithm can estimate $\mathbf{U}_k$ from it.
>
> This simple argument confirms that the dependency of the condition on $||\mathbf{U}\_k^\top\boldsymbol{\Xi}\_t||\_2$ on the gap $\lambda_k - \lambda_{k+1}$ is fundamental, and cannot be improved upon. Using similar arguments on the same matrix $\mathbf{A}$ and letting $\mathbf{B} = \lambda_k \mathbf{I}\_d$, we can see that the dependency of $||\mathbf{U}\_{-k}^\top\boldsymbol{\Xi}\_t||\_2$ on $\lambda_k - \lambda_{k+1}$ also cannot be improved.
>
> ## Eventual improvements with random centered noise
>
> When the noise is random and centered, the convergence rate can not be improved in comparison to our main result. Indeed, under the assumptions of Theorem 2.2, for $\beta = \lambda_{k+1}^2/4$, ANPM already achieves the optimal worst-case convergence rate of block Krylov methods. Notice that the case $\boldsymbol{\Xi}_t = \mathbf{0}$ corresponds to an instance with random, zero-mean noise, and it is known that in this case, the best convergence rate attainable is the same as ANPM.
>
> However, we believe that more favorable conditions on the noise's magnitude can be obtained when the noise is random centered. This is evidenced by the experiments made on stochastic centered noise in Figure 3 in the appendix (page 42). Theorem 3 of [2] provides convergence guarantees for ANPM in a particular case of random centered noise, and gives an upper bound of the "noise magnitude" (here the standard deviation of the noise) in $(\lambda_1 - \lambda_2)^{3/4}$, which is in line with the observed improved dependency on the eigengap for our experiments with random centered noise in Figure 3.
>
> [1] Hardt, Price. The Noisy Power Method: A Meta Algorithm with Applications. NeurIPS 2014.
>
> [2] Xu, He, De Sa, Mitliagkas, Re. Accelerated Stochastic Power Iteration. AISTATS 2018.

---

> > ### Author Rebuttal · Reviewer_YFyf · 2026-04-03
> >
> > My concerns are addressed, and I would like to raise the score.

---

### Official Review · Reviewer_6JfM · 2026-03-10

**Soundness:** 3
**Presentation:** 3
**Significance:** 3
**Originality:** 3
**Overall Recommendation:** 5
**Confidence:** 3

**Summary:**

This paper studies accelerated noisy power method and provides an improved analysis which preserves the accelerated convergence rate with milder conditions. The authors also extend their method to decentralized PCA and provide accelerated convergence.

**Compliance With Llm Reviewing Policy:**

Affirmed.

**Final Justification:**

The rebuttal addressed my concerns.

**Key Questions For Authors:**

see weaknesses

**Limitations:**

yes

**Strengths And Weaknesses:**

Strengths:

1. The paper is well written and easy to follow.
2. The theoretical contribution is solid. The paper presents an improved analysis that preserves the accelerated convergence rate under milder conditions. The authors also demonstrate the tightness of the noise conditions.
3. To the best of my knowledge, the paper provide the first accelerate convergence rate for decentralized PCA.

Weaknesses:
I do not see any major weaknesses in the paper, but I would appreciate some discussion on the following points:
1. Can rigorous theoretical guarantees be established for the adaptive choice of $\beta$ in (5)? While the adaptive $\beta$ appears to be more practical since it is hard to find the optimal $\beta$, the current analysis does not provide any formal guarantees regarding its convergence behavior.
2. Are the convergence results and the communication cost for the decentralized PCA tight? The authors could further clarify whether the derived convergence rates and the communication cost of AdePM are optimal.

---

> ### Author Rebuttal · Authors · 2026-03-31
>
> We thank the reviewer for the constructive comments and positive feedback on our paper. We address the reviewer's questions as follows:
>
> ## Theoretical guarantees on the adaptive choice of $\beta_t$
>
> Understanding the behavior of ANPM when $\beta_t$ is adaptively tuned as in (5) poses an interesting and challenging problem. Theoretical guarantees in this setting are not well-established in the literature, even in the noiseless case. A careful study of the noiseless accelerated power method with this tuning heuristic could yield valuable insights and may well warrant further investigation. We emphasize that tackling this problem would likely be highly nontrivial.
>
> ## Are the convergence rate and communication cost of AdePM optimal?
>
> Defining optimal convergence rates and communication costs for the problem of decentralized PCA requires defining a class of algorithms over which we desire to compute these lower bounds. It is not clear in this case what is the right class of algorithms to define, which would allow us to explicitly compute these lower bounds and which would include algorithms such as DePM and ADePM.
>
> To simplify the discussion, consider the case $k=1$. Then, we can define a class of algorithms analogous to the one defined in [1] for decentralized convex optimization. At each local computation step, agent $i$ updates its local memory $M_{i,t}$ by adding to it $\mathrm{Span}(\mathbf{x}, \mathbf{A}\_i\mathbf{x} : \mathbf{x} \in M_{i,t})$ and at each communication step, agent $i$ updates its local memory by adding to it $\mathrm{Span}(\cup_{j \sim i}M_{j,t})$. The output of agent $i$ at step $T$ is then a unit vector in $M_{i,T}$. DePM, DeEPCA and ADePM all belong to this class of algorithms.
>
> For this class of algorithms, we can see that in terms of number of local computations, ADePM is worst-case optimal over the set of matrices $\mathbf{A}$ with top-2 eigenvalues $\lambda_1 > \lambda_2$. Indeed, in the case where for all $i$, $\mathbf{A}\_i = \mathbf{A}$, then the aforementioned class of algorithm just comes down to Krylov subspace methods, for which the optimal convergence rate is in $\tilde{O}(\sqrt{\lambda_1/(\lambda_1 - \lambda_2)})$, which is achieved by ADePM.
>
> It is harder to derive exactly the optimal number of communications for this class of algorithms, however we believe that ADePM is not optimal in terms of communications. Indeed, we believe that an alternative version of ADePM which uses the same subspace tracking technique as DeEPCA [2] would have an accelerated convergence rate, but a number of communications that is independent of $\varepsilon$ just like DeEPCA. This algorithm would then be more communication-efficient than ADePM. It is not clear whether this algorithm would then be optimal within the class of algorithms we just defined. We emphasize also that analyzing such an algorithm would be challenging.
>
> [1] Scaman, Bach, Bubeck, Lee, Massoulié. Optimal Algorithms for Smooth and Strongly Convex Distributed Optimization in Networks. ICML 2017.
>
> [2] Ye, Zhang. DeEPCA: Decentralized Exact PCA with Linear Convergence Rate. JMLR 2021.

---

> > ### Author Rebuttal · Reviewer_6JfM · 2026-04-02
> >
> > The authors have addressed my concerns.

---

### Official Review · Reviewer_59aW · 2026-03-10

**Soundness:** 4
**Presentation:** 3
**Significance:** 3
**Originality:** 2
**Overall Recommendation:** 5
**Confidence:** 2

**Summary:**

In principal component analysis (PCA), the goal is, given a positive semidefinite matrix $A$, to estimate the subspace spanned by the eigenvectors corresponding to its top $k$ eigenvalues. A popular method for this problem that uses queries of the form $Ax$ is the power method. In many practical settings, however, one does not have exact access to such queries but instead obtains estimates with bounded error. Algorithms such as the noisy power method have been studied in this setting, showing that their convergence guarantees are essentially the same as in the noiseless case. The number of iterations required for the noisy power method to converge is inversely proportional to $\lambda_k -\lambda_{k+1},$ which can be problematic when the gap between these eigenvalues is small. To mitigate this dependence, accelerated variants of the power method have been proposed, achieving convergence in a number of iterations proportional to $
\frac{1}{\sqrt{\lambda_k - \lambda_{k+1}}}.$ However, existing analyses guarantee robustness only when the noise norm is exponentially small in $\frac{1}{\sqrt{\lambda_k - \lambda_{k+1}}}.$

This work provides a refined analysis of the accelerated noisy power method, improving the upper bound on the allowable noise magnitude and establishing essentially optimal bounds in terms of the eigenvalue gap $\lambda_k - \lambda_{k+1}$ and the target accuracy $\epsilon$. The techniques developed in the paper also extend to the decentralized PCA setting, yielding communication complexity guarantees comparable to prior work while simultaneously achieving accelerated convergence.

**Compliance With Llm Reviewing Policy:**

Affirmed.

**Final Justification:**

I find this to be a strong paper, I maintain my positive score.

**Key Questions For Authors:**

1. Are there any advantages of using the noisy power method instead of using the accelerated noisy power method?
2. It would be great if you could give a more detailed proof sketch, the one that you currently have is about 3-4 paragraphs, which I think is too short for a more theoretically inclined paper.
3. Are there any theoretical guarantees on how one can get a good $\beta$ for the algorithm? (maybe you mentioned that and I missed it)

**Limitations:**

yes

**Strengths And Weaknesses:**

**Strengths**: The paper is well written and the authors do a good job of placing their work within the existing literature. The results appear to be essentially optimal for this algorithm, improving upon previous bounds. Moreover, the application to decentralized PCA is interesting and highlights the potential impact of the techniques developed in the paper.

**Weaknesses**: The proof sketch is quite short and limited in detail. As a result, it is somewhat difficult to fully evaluate the technical contribution of the work. Additionally, I am not very familiar with this line of research, and therefore I am unsure how much interest there is in guarantees that assume worst-case noise.

---

> ### Author Rebuttal · Authors · 2026-03-31
>
> We thank the reviewer for the constructive comments and positive feedback on our paper. We address the reviewer's questions as follows:
>
> ## Advantages of using the Noisy Power Method (NPM) instead of the Accelerated Noisy Power Method (ANPM)
>
> The main advantage of NPM over ANPM is the fact that no parameter needs to be specified before running it. In cases where $\beta > \lambda_k^2/4$, ANPM is not guaranteed to converge, and may fail to do so. NPM does not depend on user-specified parameters, and is guaranteed to converge as long as the noise conditions are satisfied. However, this weakness is alleviated by the tuning heuristic we propose for ANPM, which does not require specifying a parameter, and which in practice generally satisfies $\beta_t < \lambda_k^2/4$.
>
> Otherwise, as long as $\beta \leq \lambda_{k+1}^2/4$, the noise conditions we derive for ANPM are the same as those derived for NPM in [1], so that there are no losses in terms of final accuracy for ANPM in comparison to NPM.
>
> ## More detailed proof sketch
>
> We thank the reviewer for this suggestion, and we agree that the paper would gain from a more detailed proof sketch. If the paper gets accepted, we will add an additional paragraph in our proof sketch of Theorem 2.2. The paragraph we plan to add will explain more in detail the three-term recurrence on $\mathbf{H}_t$, more specifically how we make the sequence $\{\mathbf{G}_t\}$ appear in it, and how the recurrence on $\{\mathbf{G}_t\}$ allows us to tightly upper bound $||\mathbf{H}_t||_2$. We propose to focus on this part of the proof, as we deem it to be the key step that allows us to obtain better noise conditions than those of [2].
>
> ## Theoretical guarantees on how one can get a good $\beta$ for ANPM
>
> For ANPM, a "good" $\beta$ should be close to the optimal $\beta^* = \lambda_{k+1}^2/4$. However, estimating exactly $\lambda_{k+1}$ is generally a task at least as hard as estimating $\mathbf{U}\_k$. In general, without any prior knowledge of the structure of the matrix $\mathbf{A}$, there is no theoretically grounded way of estimating $\beta^*$. However, in many cases, the matrix $\mathbf{A}$ exhibits some structure, that may allow practitioners to have a good guess of what $\lambda_{k+1}$ is. For instance, a classical model in high-dimensional statistics is the spiked covariance model [3], whose limiting spectral distribution is a good approximation of those of covariance matrices in many practical settings. In that case, $\lambda_{k+1}$ could be considered to be the largest eigenvalue of the spectrum's "bulk", whose formula is known, and depends on the model's parameters.
>
> ## On the interest there is in guarantees that assume worst-case noise
>
> While we agree that in many applications, the noise is not assumed to be deterministic of bounded spectral norm, but rather random and centered, there are several settings in which bounded/adversarial noise assumptions are necessary in order to derive convergence results. This is the case of decentralized PCA, and it was the main motivation for our main theorem. Indeed, in decentralized PCA, the noise $\boldsymbol{\Xi}_t$ stems from the approximation error made during gossip averaging. Proposition D.2. gives an upper bound on $||\boldsymbol{\Xi}_t||_2$ under large enough communications $L$, which allows us to directly apply our main result to prove the convergence of ADePM in Theorem 3.3. Note that in that case, this upper bound on $||\boldsymbol{\Xi}_t||_2$ is the only way that we are able to control the noise, as the $\boldsymbol{\Xi}\_t$'s obtained through gossip are not stochastic mean-centered noises. As such, our main result, which assumes bounded/adversarial noise, is required for the analysis of ADePM.
>
> Beyond the setting of decentralized PCA, there are many other cases in which results assuming bounded/adversarial noise are necessary. This is the case for federated PCA with compression, in which each agent sends to a central server a compressed version of $\mathbf{Y}\_{i,t}$. In that case, the noise stems from the difference between the compressed version and the uncompressed version of $\mathbf{Y}\_{i,t}$. This compression noise is typically of bounded spectral norm, but is usually not random and mean-centered. Another example where the noise's spectral norm is upper bounded is the one of PCA in low-precision arithmetic.
>
> Finally, as explained in the paper, note that using concentration inequalities, one can obtain upper bounds on the spectral norm of random centered noise that hold with high probability, which can then be used to apply our main theorem.
>
> [1] Hardt, Price. The Noisy Power Method: A Meta Algorithm with Applications. NeurIPS 2014.
>
> [2] Xu. On the Accelerated Noise-Tolerant Power Method. AISTATS 2023.
>
> [3] Johnstone. On the distribution of the largest Principal Component. Ann. Statist. 2001.

---

> > ### Author Rebuttal · Reviewer_59aW · 2026-04-03
> >
> > The authors have addressed most of my concerns and answered my questions. I maintain my positive assessment of the work.

---

### Official Review · Reviewer_JXrL · 2026-03-11

**Soundness:** 4
**Presentation:** 4
**Significance:** 3
**Originality:** 3
**Overall Recommendation:** 5
**Confidence:** 4

**Summary:**

This paper provides an improved analysis of the Accelerated Noisy Power Method (ANPM) for computing the top-k eigenspace of a PSD matrix A when only approximate matrix-vector products x → Ax + Ξ are available. The key result (Theorem 2.2) shows that ANPM converges at the accelerated rate Õ(√(λ_k / (λ_k − λ_{k+1}))) — matching the optimal worst-case rate of Krylov subspace methods — under noise conditions that match those of the non-accelerated Noisy Power Method of Hardt & Price (2014). This is a substantial improvement over Xu (2023), whose analysis required the noise to be ε^μ-small, where μ = Ω̃(√(λ_k / (λ_k − λ_{k+1}))) — a condition that becomes vacuous for ill-conditioned problems. The authors further prove that their analysis is worst-case optimal: the convergence rate cannot be improved (Theorem 2.3), and the noise conditions cannot be relaxed (Theorems 2.4, 2.5). As a practical application, they derive ADePM, an accelerated algorithm for decentralized PCA that achieves the accelerated convergence rate with communication costs comparable to non-accelerated decentralized methods. Experiments on synthetic instances and two real-world decentralized tasks (Fed-Heart-Disease, Ego-Facebook spectral clustering) validate the theoretical findings.

**Compliance With Llm Reviewing Policy:**

Affirmed.

**Key Questions For Authors:**

- Can you comment on the prospects for extending the analysis to the wider-gap setting (p > k columns)? Is the H_t-based proof technique compatible with this extension, or would it require fundamentally different ideas?
- In the decentralized setting, the communication cost L depends on M = max_i ‖A_i‖₂, which measures client heterogeneity. How sensitive is ADePM to this quantity in practice? The experiments use naturally heterogeneous data, but a controlled study varying heterogeneity would be informative.

**Limitations:**

This is a strong theoretical contribution that resolves a clear and well-motivated open problem: achieving accelerated convergence for noisy PCA without imposing impractically restrictive noise conditions. The result is clean (the noise conditions match the non-accelerated baseline exactly), optimal (both the rate and the conditions are tight up to constants), and practically relevant (it enables the first accelerated decentralized PCA algorithm). The paper is well-written, the proof technique is elegant, and the experimental results, while limited in scale, validate the theory convincingly. The main limitations — restricted to the k-column setting, limited experimental scope, and lack of analysis for the adaptive heuristic — are minor relative to the strength of the core contribution. I recommend acceptance.

**Strengths And Weaknesses:**

## Strengths:

1. The main result is a clean and significant improvement. The gap between Hardt & Price (2014) and Xu (2023) was a genuine obstacle: the non-accelerated method required noise ε-small, while the accelerated method required noise
ε^μ-small with μ growing with the inverse spectral gap. Since the whole point of acceleration is to help with ill-conditioned problems (small eigengap), having noise conditions that become more restrictive precisely when the
gap is small rendered Xu (2023)'s result impractical. This paper closes the gap completely: acceleration is now "free" in the sense that it requires no stronger noise conditions than the non-accelerated baseline. This is the
right result, and it is satisfying that it can be proven.

2. The optimality results (Theorems 2.3–2.5) are a valuable complement to the upper bound. Theorem 2.3 confirms that the Õ(√(λ_k / (λ_k − λ_{k+1}))) rate is tight (matching Krylov lower bounds). Theorems 2.4 and 2.5 show that
the noise conditions on U_{−k} Ξ_t and U_k Ξ_t are individually tight up to constants — relaxing either one by a constant factor can prevent convergence. This three-part optimality (rate + both noise conditions) is unusually
complete for this type of analysis.

3. The proof technique is elegant. The key insight — analyzing the evolution of H_t = (U_{−k} X_t)(U_k X_t)^{−1} rather than X_t directly — avoids the coarse bounds on ‖R_t‖₂ that introduced the dependence on λ₁ in Xu (2023)'s
analysis. The connection to scaled Chebyshev polynomials through the three-term recurrence is natural and well-exploited.

4. The application to decentralized PCA (ADePM) is a compelling demonstration of practical relevance. Table 2 shows that ADePM achieves the accelerated iteration complexity while maintaining communication costs comparable to
DePM and DeEPCA. The authors correctly identify that the overly restrictive noise conditions of Xu (2023) were the bottleneck preventing accelerated decentralized PCA algorithms from being proposed — their improved analysis
directly removes this bottleneck. To the best of my knowledge, this is indeed the first decentralized PCA algorithm with provably accelerated convergence.

5. The experimental results, while primarily illustrative, are well-designed. The synthetic experiments (Figure 1) systematically vary β, Δ_k, and ξ to validate each aspect of the theory: the acceleration benefit, the dependence
on the eigengap, and the proportional scaling of final accuracy with noise magnitude. The adaptive β heuristic (Equation 5) performs comparably to the optimal β* in practice, which is important for usability. The decentralized
experiments on real data (Figure 2) confirm the speedup of ADePM over DePM and DeEPCA.

6. The paper is clearly and carefully written. The comparison table (Table 1) immediately conveys the improvement. The proof sketch in the main text provides good intuition, and the discussion of the transient vs. stationary
regimes in the experiments is insightful.

## Weaknesses:

1. The experimental evaluation, while clean, is limited in scope. The synthetic experiments are on relatively small instances (the dimensions and specific parameters are deferred to the appendix). The decentralized experiments
use only two datasets. For a paper whose primary contribution is theoretical, this is acceptable, but additional experiments on larger-scale problems — particularly in the private PCA or streaming PCA settings mentioned in the
introduction — would strengthen the practical narrative.

2. The paper does not address the wider-gap setting of Balcan et al. (2016) and Xu (2023), where the iterate X_t has p > k columns but only the top-k eigenspace is estimated. The authors acknowledge this in the conclusion as
future work, but since this setting can provide significant practical improvements in noise tolerance and convergence speed, its absence is a limitation of the current analysis.

3. The adaptive β heuristic (Equation 5) is presented without theoretical justification. While it works well empirically, the paper does not analyze whether it preserves the accelerated rate or the mild noise conditions. A brief
discussion of when this heuristic might fail would be useful.

4. The noise model is adversarial with bounded spectral norm. While this is the standard setting for this line of work, the paper could discuss more explicitly how the results translate to the stochastic noise setting (mentioned
briefly in Section 2.1), which is more natural for the streaming PCA application.

## Minor Comment:

- The observation in Section 4.1 that the dependence on Δ_k is "not exactly linear between 10⁻¹ and 10⁻²" and the conjecture about tail eigenvalues is an interesting remark that could inspire future work on gap-free or multi-gap analyses.

---

> ### Author Rebuttal · Authors · 2026-03-31
>
> We thank the reviewer for the constructive comments and positive feedback on our paper. We address the reviewer's questions as follows:
>
> ## Extending our analysis to the wider-gap setting
>
> Extending our results to the $p>k$ case as done in [1,2] is not immediate, however we believe that our proof technique could be used for this more general case. Indeed, the way the wider-gap convergence rates are obtained in [1,2] is through the analysis of the matrix $\tilde{\mathbf{H}}\_t = \mathbf{H}\_t [\mathbf{I}\_k , \mathbf{0}]^\top = (\mathbf{U}\_{-p}^\top \mathbf{X}\_t)(\mathbf{U}\_{p}^\top \mathbf{X}\_t)^{-1}[ \mathbf{I}\_k , \mathbf{0}]^\top$. From Lemma 2.3 in [1], the spectral norm of this matrix upper bounds $\sin\theta_k(\mathbf{U}\_k, \mathbf{X}\_t)$, so that showing that $||\tilde{\mathbf{H}}\_t||\_2$ converges at the rate $\tilde{O}(\sqrt{{\lambda_k}/(\lambda_k-\lambda_{p+1})})$ would give the result.
>
> To see how the $[\mathbf{I}\_k , \mathbf{0}]^\top$ factor leads to this improved rate, notice that in the noiseless case ($\boldsymbol{\Xi}\_t = \mathbf{0}$), $\mathbf{H}\_t = p_t(\boldsymbol{\Lambda}\_{-p}) \mathbf{H}\_0 p_t(\boldsymbol{\Lambda}\_p)^{-1}$, so that $\tilde{\mathbf{H}}\_t = p_t(\boldsymbol{\Lambda}\_{-p}) \mathbf{H}\_0 p_t(\boldsymbol{\Lambda}\_p)^{-1} [\mathbf{I}\_k , \mathbf{0}]^\top = p_t(\boldsymbol{\Lambda}\_{-p}) \mathbf{H}\_0 p_t(\boldsymbol{\Lambda}\_k)^{-1}$. If $\beta$ is well chosen (i.e. $\lambda_p > 2\sqrt{\beta} \geq \lambda_{p+1}$), we deduce the upper bound $||\tilde{\mathbf{H}}\_t||\_2 \leq \sqrt{\beta}^t/p_t(\lambda_k) ||\mathbf{H}\_0||\_2$, and the rate $\tilde{O}(\sqrt{{\lambda_k}/(\lambda_k-\lambda_{p+1})})$ follows if $\beta = \lambda_{p+1}^2/4$.
>
> The idea would be to use our expression of $\mathbf{H}_t$ in Lemma C.5 to upper bound $\|\tilde{\mathbf{H}}_t\|_2$. The main difficulty in doing so would be to derive a tight upper bound on $\|\mathbf{C}_t^{-1}  [ \mathbf{I}_k , \mathbf{0}]^\top\|_2$ (where $\mathbf{C}_t$ is defined in (21)). Without noise, it would be equal to exactly $1/p_t(\lambda_k)$, and we can expect that for small enough noises, it remains close to this value. Proving this would require a more precise analysis of the sequence $\{\mathbf{C}_t\}$ than what is done in Lemma C.8.
>
> ## Sensitivity of ADePM to $M = \max_i ||\mathbf{A}_i||_2$ in practice
>
> We propose an additional experiment, testing DePM, ADePM and DeEPCA for PCA on the digits dataset from scikit-learn, with $n=10$ agents, on two different instances. In the first one, the data is randomly spread between the agents, so that the local matrices $\mathbf{A}_i$ are homogeneous, with $M\approx230$. In the second one, agent $i$ builds the $\mathbf{A}_i$ using only the data points corresponding to digit $i$, so that the matrices are heterogeneous, and $M\approx725$. Due to a lack of space, we only provide the error $n^{-1}\sum_i \sin\theta_k(\mathbf{U}\_k,\mathbf{X}\_{i,T})$ reached by ADePM with $\beta=\beta^*$ (DePM and ADePM with $\beta = \beta_t$ reach similar levels) after $T=200$ iterations. For $L=20$ communication per iteration, the final error is $1.3\times10^{-4}$ in the homogeneous case, and $7.8\times10^{-4}$ in the heterogeneous case. For $L=40$, we get $2.6\times10^{-8}$ in the homogeneous case, and $1.7\times 10^{-7}$ in the heterogeneous case, which confirms that in practical settings, heterogeneity increases the communication complexity. If the paper is accepted, and the reviewer finds this experiment interesting, we will include it in the final version of the paper, in a similar form as the plots in Figure 2.
>
> As an additional comment, the condition on $L$ in Theorem 3.3 may suggest that at fixed $L,\lambda_k,$ and $\lambda_{k+1}$, $M/\varepsilon$ is constant, which is not observed in our experiment. This is because the bounds that make $M$ appear in our proof of Theorem 3.3 are loose, and tighter inequalities that involve the heterogeneity of the data and the graph's topology (as is done in e.g. [3] for decentralized optimization) may better capture the impact of heterogeneity on the communication cost.
>
> ## Additional experiments on larger-scale problems for private or streaming PCA
>
> We thank the reviewer for this suggestion. If our paper gets accepted, we would like to add an experiment simulating private PCA on the Amazon0302 dataset [4], which yields a sparse matrix of very large scale ($d>250000$), and in which we add noise $\boldsymbol{\Xi}_t$ whose entries are i.i.d. centered gaussians.
>
> [1] Balcan, Du, Wang, Yu. An Improved Gap-Dependency Analysis of the Noisy Power Method. COLT 2016.
>
> [2] Xu. On the Accelerated Noise-Tolerant Power Method. AISTATS 2023.
>
> [3] Le Bars, Bellet, Tommasi, Lavoie, Kermarrec. Refined Convergence and Topology Learning for Decentralized SGD with Heterogeneous Data. AISTATS 2023.
>
> [4] Leskovec, Adamic, Adamic. The Dynamics of Viral Marketing. ACM TWEB 2007.

---

> > ### Author Rebuttal · Reviewer_JXrL · 2026-04-03
> >
> > --

---

### Decision · Program_Chairs · 2026-04-30

**Decision:**

Accept (regular)

**Comment:**

This paper provides an improved analysis of the accelerated noisy power method, achieving the tighter dependency on spectral gap in the upper bounds. The authors also provide lower bounds to show the proposed method is optimal. Furthermore, the main results also can be extended to decentralized setting. All reviewers gave the positive evaluation on this paper, so that I recommend acceptance.

I have a minor question on the assumptions of Theorem 2, i.e., Eq (4) depends on ${\mathbf X}_t$, which is generated by ANPM iteration. Typically, we desire the assumption of the problem to be independent on the algorithm. I hope the authors could provide some discussion on this assumption.